# Learning long range dependencies through time reversal symmetry breaking

**Guillaume Pourcel**
University of Groningen *, INRIA †
g.a.pourcel@rug.nl

**Maxence Ernoult**
Google DeepMind
mernoult@google.com

## Abstract

Deep State Space Models (SSMs) reignite physics-grounded compute paradigms, as RNNs could natively be embodied into dynamical systems. This calls for dedicated learning algorithms obeying to core physical principles, with efficient techniques to simulate these systems and guide their design. We propose *Recurrent Hamiltonian Echo Learning* (RHEL), an algorithm which provably computes loss gradients as finite differences of physical trajectories of non-dissipative, *Hamiltonian systems*. In ML terms, RHEL only requires three "forward passes" irrespective of model size, without explicit Jacobian computation, nor incurring any variance in the gradient estimation. Motivated by the potential to implement our algorithm in non-digital physical systems, we first introduce RHEL in *continuous time* and demonstrate its formal equivalence with the continuous adjoint state method. To facilitate the simulation of Hamiltonian systems trained by RHEL, we propose a *discrete-time* version of RHEL which is equivalent to Backpropagation Through Time (BPTT) when applied to a class of recurrent modules which we call *Hamiltonian Recurrent Units* (HRUs). This setting allows us to demonstrate the scalability of RHEL by generalizing these results to hierarchies of HRUs, which we call *Hamiltonian SSMs* (HSSMs). We apply RHEL to train HSSMs with linear and nonlinear dynamics on a variety of time-series tasks ranging from mid-range to long-range classification and regression with sequence length reaching $\sim 50k$. We show that RHEL consistently matches the performance of BPTT across all models and tasks [3]. This work opens new doors for the design of scalable, energy-efficient physical systems endowed with self-learning capabilities for sequence modelling.

## 1   Introduction

The resurgence of Recurrent Neural Networks (RNNs) for sequence modeling [1], particularly when integrated with State Space Models (SSMs) [1–6], reopens the debate about the "hardware lottery". This raises a critical question with high stakes [7]: should future hardware development continue to optimize for the long-dominant GPU/TPU-Transformer-backprop paradigm [8], or should it explore alternative computational approaches alongside novel training algorithms?

This paper embraces a distinctive standpoint in the realm of AI hardware by not distinguishing algorithms from hardware [9] and posits that some SSMs could be physically realized in dynamical physical systems and, when endowed with bespoke credit assignment mechanisms, be turned into "self-learning" machines [10, 11] – see Section 5 for a detailed discussion. To this end, such algorithms should fulfill at least two requirements: i) use "forward passes" only (*i.e.* no backward

---

*Bernoulli Institute & CogniGron

†University of Lille, Inria, CNRS, Centrale Lille, UMR 9189 - CRIStAL

[3]Our code is available on: https://github.com/guillaumepourcel/rhel

passes), ii) do not use explicit state-Jacobian (*i.e.* Jacobian of the system's dynamics). The vast majority of existing algorithms endowed with these features revolve around forward-mode automatic differentiation (AD) [12]. The standard forward-mode AD algorithm for temporal processing is Recurrent Real-Time Learning [13] (RTRL). Due to its cubic memory complexity with respect to the number of neurons, RTRL can only be exactly implemented on architectures of limited size [14] and either requires low-rank approximations [15], or hardcoded [16] or metalearned [17] heuristics.

RTRL fails to satisfy criteria ii) because it explicitly uses the Jacobian to compute directional derivatives of the loss $L$ along each direction $v_i$ of the canonical basis of parameters $\theta$ as a supplementary computation during the forward pass. However, it can be reformulated as a zeroth-order procedure by approximating the directional derivatives via $\nabla_{\theta} L \cdot v = \lim_{\epsilon \to 0} \epsilon^{-1}(L(\theta + \epsilon v) - L(\theta))$. This recipe theoretically aligns with our three algorithmic requirements but incurs a prohibitive complexity cost by requiring separate forward passes for each perturbation along every parameter direction. Alternative methods attempt to circumvent this by sampling random directions from the parameter space [18–21] rather than exhaustively computing gradients along the entire canonical basis. However, these approaches suffer from either excessive variance [22] or high bias [23], limiting their effectiveness to small-scale applications [24] or fine-tuning tasks [25]. To avoid the pitfalls of RTRL and other forward-mode AD proxies, a better approach would be to instead emulate *backward-mode* AD, *forward* in time [26]. Yet, such techniques have only developed to emulate backward-mode *implicit differentiation* on energy-based models on static inputs [27] and have not been extended out of equilibrium to sequential data.

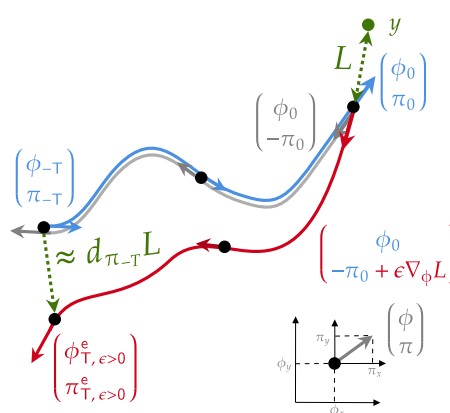

Figure 1: **HEB core mechanics [10].** After following a free forward trajectory (blue curve) and undergoing "rebound", the neurons exactly travel backward (grey curve). Instead, HEB prescribes nudging the "echo" trajectory (red curve) closer to $y$ before rebound, with the resulting position gap encoding the error gradient with respect to momentum (the left green arrow).

In this work, we propose an alternative approach that emulates *backward-mode* AD *forward in time*, inspired by the Hamiltonian Echo Backprop (HEB) algorithm [10], illustrated in Fig. 1 for a single neuron. Consider a system of neurons described by their position $\phi_i$ and momentum $\pi_i$ which do not dissipate energy. This means, for instance for coupled oscillators (Eq. (8)), that the initial energy provided to this system is preserved and transferred across oscillators from kinetic energy to elastic potential or *vice versa*. Such systems can be modelled by the *Hamiltonian formalism* (section 2.1). Let us say that we want to learn the initial conditions on this system such that it reaches some target $y$ after some time. HEB proceeds in three steps. For the first step, the system evolves freely, but does not reach $y$. For the second step, HEB perturbs the trajectory for a brief duration to drive the system slightly closer to $y$ per some metric $L$. Then in the last step, the neurons are "bounced" backwards, causing the system to evolve backward. If the perturbation was omitted, the system would *exactly* retrace its previous trajectory backward in time, a property which is called *time-reversal symmetry*. Because the perturbation *breaks* time-reversal symmetry, the system no longer retraces its initial trajectory exactly—the resulting deviation encodes the gradient of $L$ with respect to its initial state.

In spite of its significant implications for physical learning, HEB has multiple features which may hinder its broader investigation within the ML community: i) HEB is difficult to compare to standard ML algorithmic baselines as it was derived with dedicated theoretical physics tools and is entangled with fine-grained physics of the systems being trained; ii) HEB theory assumes that model parameters are also dynamical variables and that only their initial state is learnable, which is in stark contrast with standard RNN parametrization; iii) it is unclear how HEB would extend to SSMs, *i.e.* in *discrete time*, on sequential data with losses defined at all timesteps, on *hierarchical* recurrent units; iv) finally, HEB was only evaluated on small static problems (*e.g.* XOR, MNIST) and not proved at larger scale.

Taking inspiration from HEB and using standard ML tools, we propose *Recurrent Hamiltonian Echo Learning* (RHEL) as a simple, general and scalable forward-only proxy of backward-mode AD applying to a broad class of dissipative-free Hamiltonian models with the following key contributions:

- With the primary goal in mind to inform the design of self-learning physical systems and to facilitate its comparison to HEB, we first introduce RHEL in *continuous time*. We show that RHEL generalizes HEB to a broader class of models and problems and demonstrate its equivalence, at *every timestep*, with the *continuous adjoint state method* [28] in the limit of small trajectory perturbations (Theorem 3.1). We numerically highlight this property on a toy physical model (Eq. (8), Fig. 2).

- To efficiently simulate this algorithm, we extend RHEL to *discrete time* and demonstrate its equivalence with *Backpropagation Through Time* (BPTT) (Theorem 3.2) on *Hamiltonian Recurrent Units* (HRUs) – which are discrete-time, symplectic integrators of *separable* Hamiltonian dynamics. We further generalize this result to learning a *hierarchy* of HRUs, which we call *Hamiltonian State Space Models* (HSSMs, Fig. 2), and propose a *RHL chaining* procedure accordingly (Theorem 3.3).

- With this simulation toolbox in hand, we finally demonstrate the effectiveness of the proposed approach on HSSMs with linear [29] and nonlinear [30] recurrent units. We show that: i) gradients estimates produced by RHEL near perfectly match gradients computed by end-to-end AD (Fig. 4), ii) RHEL remains on par with AD in terms of resulting model performance across classification and regression long-range tasks (Tables 1–2).

## 2 Problem statement

### 2.1 Notations & model description.

**Notations.** Given a differentiable mapping $H : \mathbb{R}^d \to \mathbb{R}$, we denote $\nabla_{\boldsymbol{v}_1} H \in \mathbb{R}^{d_{\boldsymbol{v}_1} \times 1}$ and $\nabla^2_{\boldsymbol{v}_1, \boldsymbol{v}_2} H \in \mathbb{R}^{d_{\boldsymbol{v}_1} \times d_{\boldsymbol{v}_2}}$ the gradient and Hessian of $H$ respectively. When necessary, we use the Leibniz notation to denote as $\nabla_i H$ the gradient of $H$ with respect to the $i^{\text{th}}$ variable. When considering a differentiable mapping $\mathcal{M} : \mathbb{R}^d \to \mathbb{R}^n$, we denote its Jacobian matrix as $\partial \mathcal{M} \in \mathbb{R}^{n \times d}$. We also use the notation $d_{\boldsymbol{v}_1}$ to denote a *total* derivative with respect to some variable $\boldsymbol{v}_1$ which accounts for both direct and *indirect* effects of the variable $\boldsymbol{v}_1$ on the function being differentiated.

**Continuous Hamiltonian model.** Following [10], we model neurons as a vector $\boldsymbol{\Phi} \in \mathcal{S} \equiv \mathbb{R}^{d_{\boldsymbol{\Phi}}}$ comprising a position vector $\boldsymbol{\phi} \in \mathbb{R}^{\frac{d_{\boldsymbol{\Phi}}}{2}}$ and a momentum vector $\boldsymbol{\pi} \in \mathbb{R}^{\frac{d_{\boldsymbol{\Phi}}}{2}}$ such that $\boldsymbol{\Phi} := (\boldsymbol{\phi}^\top, \boldsymbol{\pi}^\top)^\top$. We define $\boldsymbol{\Sigma}_z \in \mathbb{R}^{d_{\boldsymbol{\Phi}} \times d_{\boldsymbol{\Phi}}}$ such that $\boldsymbol{\Sigma}_z \cdot \boldsymbol{\Phi} = (\boldsymbol{\phi}, -\boldsymbol{\pi})$ and $\boldsymbol{\Sigma}_x \in \mathbb{R}^{d_{\boldsymbol{\Phi}} \times d_{\boldsymbol{\Phi}}}$ such that $\boldsymbol{\Sigma}_x \cdot \boldsymbol{\Phi} = (\boldsymbol{\pi}, \boldsymbol{\phi})$. Denoting $\boldsymbol{\theta} \in \Theta \equiv \mathbb{R}^{d_{\boldsymbol{\theta}}}$ and $\boldsymbol{u} \in \mathcal{U} \equiv \mathbb{R}^{d_{\boldsymbol{u}}}$ the model parameters and inputs, we define the *Hamiltonian* associated to the model as a mapping $H : \mathcal{S} \times \Theta \times \mathcal{U} \to \mathbb{R}$. Taking $-T$ as the origin of time and starting from $\boldsymbol{\Phi}(-T) = \boldsymbol{x}$, we say that $\boldsymbol{\Phi}$ follows *Hamiltonian dynamics* under a sequence of inputs $t \to \boldsymbol{u}(t)$ and with some parameters $\boldsymbol{\theta}$ if it satisfies the ordinary differential equation (ODE):

$$\forall t \in [-T, 0] : \ \partial_t \boldsymbol{\Phi}(t) = \boldsymbol{J} \cdot \nabla_{\boldsymbol{\Phi}} H[\boldsymbol{\Phi}(t), \boldsymbol{\theta}, \boldsymbol{u}(t)], \quad \boldsymbol{J} := \begin{bmatrix} \boldsymbol{0} & \boldsymbol{I} \\ -\boldsymbol{I} & \boldsymbol{0} \end{bmatrix}. \tag{1}$$

**Assumptions.** The most fundamental requirement of the whole proposed approach for the Hamiltonian models at use is *time-reversal symmetry*. Heuristically, if we were recording the dynamics of a conversative system described by Eq. (1) and playing the resulting recording forward and backward, we could not distinguish them as both are *physically feasible* (Lemma A.3). A direct consequence of this fact is that if neurons are let to evolve for some time and then bounced back, *i.e.* their momenta are reversed, they will *exactly* travel back to their initial state (Corollary A.3). Namely (see Fig. 1):

$$\left( \boldsymbol{\Phi}(-T) \xrightarrow[\text{Eq. (1)}]{} \boldsymbol{\Phi}(0) \right) \Rightarrow \left( \boldsymbol{\Phi}^\star(0) := \boldsymbol{\Sigma}_z \cdot \boldsymbol{\Phi}(0) \xrightarrow[\text{Eq. (1)}]{} \boldsymbol{\Phi}^\star(-T) \right) \tag{2}$$

### 2.2 Learning as constrained optimization

**Problem formulation.** Our goal can be framed as a constrained optimization problem where we aim to find a set of model parameters $\boldsymbol{\theta}$ such that, under some input sequence $t \to u(t)$, the model trajectory approaches as much as possible some target trajectory, as measured by a *loss function $L$*, under the constraint that the model trajectory is *physically feasible* in the sense of satifying Eq. (1):

$$\min_{\boldsymbol{\theta}} L := \int_{-T}^0 dt \ell[\boldsymbol{\Phi}(t), t] \quad \text{s.t.} \quad \forall t \in [-T, 0] : \ \partial_t \boldsymbol{\Phi}(t) = \boldsymbol{J} \cdot \nabla_{\boldsymbol{\Phi}} H[\boldsymbol{\Phi}(t), \boldsymbol{\theta}, \boldsymbol{u}(t)], \tag{3}$$

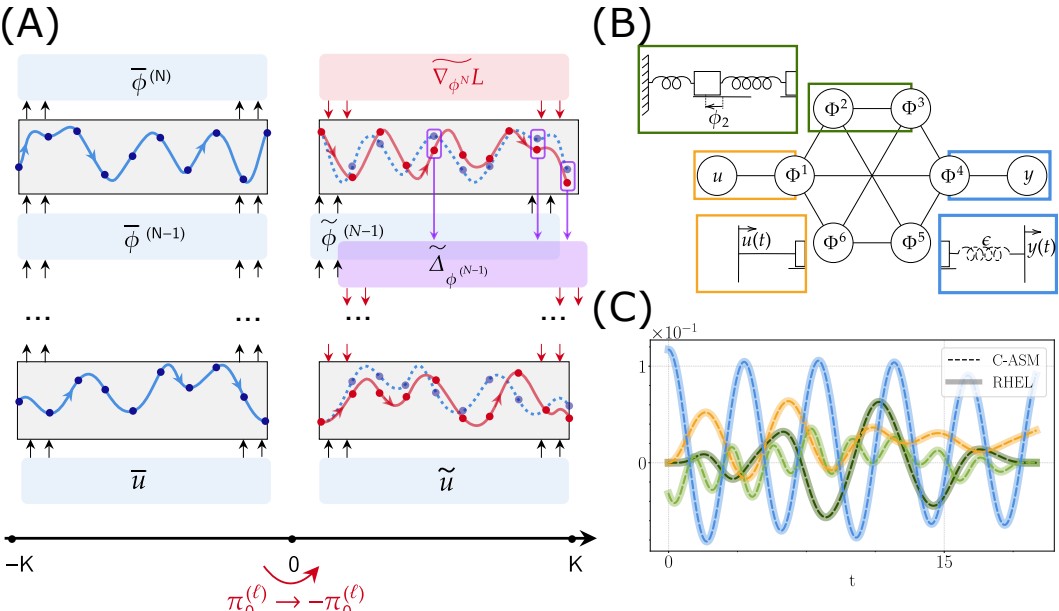

Figure 2: **(A): Learning Hamiltonian SSMs by RHEL.** The forward pass of a HSSM (left, Eq. (15)) reads as a composition of *Hamiltonian Recurrent Units* (HRUs, Eq. (9)). During the backward pass (right), the momenta of the neurons of the top-most HRU are flipped *and* nudged by the first error signal, yielding a perturbed trajectory (red curve). The contrast with the time-reversed trajectory (which would be obtained *without nudging*, dotted blue curve) yields an error signal that is passed backward (purple) to previous HRUs (Alg. 1). **(B)–(C): checking Theorem 3.1 on a toy model.** Six coupled harmonic oscillators with learnable parameters under input $u$ and target $y$ (Eq. (8)). Plots show gradients for a spring (dark green) and mass parameter (light green) comparing C-ASM ($t \to g_{\theta}(t)$, dotted) and RHEL ($t \to \Delta_{\theta}(t)$, solid), alongside sensitivities for oscillator positions $\phi^1$ (orange) and $\phi^4$ (blue) under both methods ($t \to \lambda(t)$, dotted; $t \to \Delta_{\Phi}(t)$, solid).

where $L$ reads as the sum of *cost functions* $t \to \ell(\cdot, t)$. The most common approach to solve Eq. (3) is by gradient descent or variants thereof such that the problem boils down to computing $d_{\theta}L$. Note that the problem defined by Eq. (3) is more general than the one solved by the seminal HEB work [10] as: i) the loss function $L$ is defined over the *whole* trajectory, ii) the Hamiltonian is *time-dependent* (through $t \to u(t)$) and parametrized by $\theta$ which is shared across the whole computational graph.

**Algorithmic baseline.**     One standard approach to compute $d_{\theta}L$ is the *continuous-adjoint state method* (ASM) [28], which can be simply regarded as the continuous counterpart of backward-mode AD. Given some *forward* trajectory $\{\Phi(t)\}_{t \in [-T,0]}$ spanned by Eq. (1), this method prescribes solving the following *backward* ODE:

$$\boldsymbol{\lambda}(0) = 0, \quad \partial_t \boldsymbol{\lambda}(t) = \nabla_1^2 H[\boldsymbol{\Phi}(-t), \boldsymbol{\theta}, \boldsymbol{u}(-t)] \cdot \boldsymbol{J}^\top \cdot \boldsymbol{\lambda}(t) + \nabla_1 \ell[\boldsymbol{\Phi}(-t), -t], \quad (4)$$

with the gradient of the loss with respect to the initial state of the neurons $x$ and the model parameters $\theta$ given by (Theorem A.1):

$$d_{\boldsymbol{x}}L = \boldsymbol{\lambda}(T), \quad d_{\boldsymbol{\theta}}L = \int_0^T g_{\boldsymbol{\theta}}(t)dt, \quad \text{with} \quad g_{\boldsymbol{\theta}}(t) := \nabla_{1,2}^2 H[\boldsymbol{\Phi}(-t), \boldsymbol{\theta}, \boldsymbol{u}(-t)] \cdot \boldsymbol{J}^\top \cdot \boldsymbol{\lambda}(t) \quad (5)$$

Note that in theory, the forward trajectory $\{\boldsymbol{\Phi}(t)\}_{t \in [-T,0]}$ can either be stored or recomputed backwards alongside $\boldsymbol{\lambda}$ yielding $\mathcal{O}(1)$ memory cost – a property which holds beyond Eq. (1) specifically [31]. However in practice, recomputing variables backward through a *discrete* computational graph is generally inexact and induces bias in the gradient estimation [32], unless a *reversible* discretization scheme is used. Fortunately, *symplectic* integrators associated with Hamiltonian flows are reversible [33], a property which has been leveraged to yield memory savings in neural networks [30]. Therefore, the continuous ASM *as well as* our proposed algorithm (Alg. 1) also naturally inherit this memory efficiency as a *model feature* rather than a feature of the training algorithms themselves.

# 3 Recurrent Hamiltonian Echo Learning (RHEL)

## 3.1 Definition in continuous time & equivalence with continuous ASM

We are now equipped to introduce *Recurrent Hamiltonian Echo Learning* (RHEL). In comparison to the continuous ASM, RHEL does not require solving a separate adjoint ODE akin to Eq. (4). Instead, RHEL simply prescribes running multiple additional times the forward trajectory Eq. (1) with three modifications: i) the neurons are first *conjugated*, *i.e.* $\boldsymbol{\Phi} \to \boldsymbol{\Phi}^\star$; ii) the inputs $t \to \boldsymbol{u}(-t)$ are processed *backwards*; iii) the trajectory of the neurons is slightly nudged, with a strength $\epsilon$, towards direction of decreasing loss values with cost functions $t \to \ell[\cdot, -t]$ processed *backwards* too. More precisely, let us assume a "free" forward trajectory Eq. (1) has been executed between $-T$ and $0$ yielding some neuron state $\boldsymbol{\Phi}(0)$. Then, we define the *echo* dynamics of the neurons $\boldsymbol{\Phi}^e$ through:

$$\boldsymbol{\Phi}^e(0) = \boldsymbol{\Phi}^\star(0), \quad \partial_t \boldsymbol{\Phi}^e(t, \epsilon) = \boldsymbol{J} \nabla_{\boldsymbol{\Phi}^e} H[\boldsymbol{\Phi}^e(t, \epsilon), \boldsymbol{\theta}, \boldsymbol{u}(-t)] - \epsilon \boldsymbol{J} \nabla_{\boldsymbol{\Phi}^e} \ell[\boldsymbol{\Phi}^e(t, \epsilon), -t] \quad \forall t \in [0, T] \tag{6}$$

A crucial observation is that when $\epsilon = 0$, the echo trajectory simply matches the forward trajectory in reverse: $\boldsymbol{\Phi}^e(t, \epsilon = 0) = \boldsymbol{\Phi}^\star(-t)$ for $t \in [0, T]$ (Lemma A.3), reflecting time-reversal symmetry. When $\epsilon \neq 0$, this symmetry is *broken*: the resulting trajectory difference $\boldsymbol{\Phi}^e(t, \epsilon) - \boldsymbol{\Phi}^e(t, \epsilon = 0)$ implicitly encodes for the error signals carried by the continuous ASM. We introduce this result more formally in Theorem 3.1 with the help of the following quantities capturing the essence of RHEL as a *forward-only*, difference-based gradient estimator:

$$\begin{cases} \Delta_{\boldsymbol{\theta}}^{\text{RHEL}}(t, \epsilon) &:= -\frac{1}{2\epsilon} \left( \nabla_2 H[\boldsymbol{\Phi}^e(t, \epsilon), \boldsymbol{\theta}, \boldsymbol{u}(-t)] - \nabla_2 H[\boldsymbol{\Phi}^e(t, -\epsilon), \boldsymbol{\theta}, \boldsymbol{u}(-t)] \right), \\ \Delta_{\boldsymbol{\Phi}}^{\text{RHEL}}(t, \epsilon) &:= \frac{1}{2\epsilon} \boldsymbol{\Sigma}_x \cdot \left( \boldsymbol{\Phi}^e(t, \epsilon) - \boldsymbol{\Phi}^e(t, -\epsilon) \right), \end{cases} \tag{7}$$

**Theorem 3.1** (Informal). *Under mild assumptions on the Hamiltonian function and with $t \to \lambda(t)$ and $t \to g_{\boldsymbol{\theta}}(t)$ defined by Eq. (4)–(5), the continuous ASM and RHEL are equivalent at all times in the sense that:*

$$\forall t \in [0, T], \quad \boldsymbol{\lambda}(t) = \lim_{\epsilon \to 0} \Delta_{\boldsymbol{\Phi}}^{\text{RHEL}}(t, \epsilon), \quad g_{\boldsymbol{\theta}}(t) = \lim_{\epsilon \to 0} \Delta_{\boldsymbol{\theta}}^{\text{RHEL}}(t, \epsilon)$$

*Proof sketch.* Defining $\Delta_{\boldsymbol{\Phi}}^{\text{RHEL}}(t) := \lim_{\epsilon \to 0} \Delta_{\boldsymbol{\Phi}}^{\text{RHEL}}(t, \epsilon)$ and noticing that $\Delta_{\boldsymbol{\Phi}}^{\text{RHEL}}(t) = \boldsymbol{\Sigma}_x \cdot \partial_\epsilon (\boldsymbol{\Phi}^e(t, \epsilon))|_{\epsilon=0}$, we can show by differentiating Eq. (6) around $\epsilon = 0$ and with some algebra that $t \to \Delta_{\boldsymbol{\Phi}}^{\text{RHEL}}(t)$ satisfies the same ODE as $t \to \boldsymbol{\lambda}(t)$ (Eq. (4) and have same initial conditions, therefore are equal at all times. The other two equalities of Theorem 3.1 can then be easily deduced. The full formal statement and proof are provided in Appendix A.1.3. $\square$

**Example.** We numerically verify Theorem 3.1 on a toy model of six coupled harmonic oscillators $i$ (Fig. 2) with masses $m_i$ and spring parameters $k_i, k_{ij}$. The system is driven by an external input force $u(t)$ applied to oscillator 1 and produces an output on oscillator 4, which is nudged toward a target trajectory $y(t)$ during the echo phase. The corresponding Hamiltonian reads:

$$H[\boldsymbol{\Phi}, \boldsymbol{\theta}, u] = \sum_{i=1}^{6} \frac{\pi_i^2}{2m_i} + \frac{1}{2} \sum_{i=1}^{6} k_i \phi_i^2 + \frac{1}{2} \sum_{i=1}^{6} \sum_{j>i}^{6} k_{ij}(\phi_j - \phi_i)^2 + u\phi_1, \tag{8}$$

with parameters $\boldsymbol{\theta} = \{m_i, k_i, k_{ij}\}_{i,j}$. This yields the following dynamics and associated RHEL gradient estimators (see App. A.4.1):

$$\begin{cases} \forall i \in [\![1, 6]\!], \quad \partial_t \phi_i = \pi_i/m_i, \quad \partial_t \pi_i = -k_i \phi_i + \sum_{j \neq i} k_{ij}(\phi_j - \phi_i) + \delta_{i1} u + \delta_{i4} \, \delta_e \, \epsilon(\phi_4 - y), \\ g_{k_{ij}}(t) = \lim_{\epsilon \to 0} \frac{-1}{2\epsilon} \left[ \left( \phi_j^e(t, \epsilon) - \phi_i^e(t, \epsilon) \right)^2 - \left( \phi_j^e(t, -\epsilon) - \phi_i^e(t, -\epsilon) \right)^2 \right], \end{cases}$$

where $u(t)$ denotes the external driving force applied to oscillator 1, $y(t)$ the target trajectory associated with the output oscillator 4, $\delta_{ij}$ the Kronecker delta, and $\delta_e$ an indicator equal to 1 during the echo phase and 0 otherwise.

## 3.2 Extension to discrete time & equivalence with BPTT

In this section, we propose a *discrete-time* version of RHEL by first defining a family of discrete models as integrators of the Hamiltonian flow which *exactly* preserve time-reversal symmetry, defining an associated surrogate problem and *then* solving it. This "discretize-then-optimize" approach ensures a better gradient estimation than directly discretizing RHEL theory in continuous time [32].

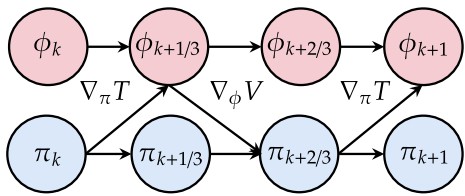

Figure 3: **Computational graph of** $\mathcal{M}_{H,\delta}$.

**Hamiltonian Recurrent Units (HRUs).** We now assume that the neurons $\mathbf{\Phi}$ obey the following discrete-time dynamics, starting from $\mathbf{\Phi}_{-K} = \boldsymbol{x}$ and:

$$\mathbf{\Phi}_{k+1} = \mathcal{M}_{H,\delta}[\mathbf{\Phi}_k, \boldsymbol{\theta}, \boldsymbol{u}_k] \quad \forall k = -K \cdots -1, \tag{9}$$

where $\mathcal{M}_{H,\delta}$ denotes the *integrator* associated with the Hamiltonian function $H$ with time discretization $\delta$. We classically pick $\mathcal{M}_{H,\delta}$ as the *Leapfrog* integrator [33] and restrict ourselves to *separable* Hamiltonians of the form $H[\mathbf{\Phi}] = T[\boldsymbol{\pi}] + V[\boldsymbol{\phi}]$ to yield a reversible *and* explicit integration scheme. One possible parametrization of the Leapfrog integrator in this case is as the composition of *three explicit Euler integrators*, which alternate between updates of the position $\boldsymbol{\phi}$ (first and third steps) and of the momentum (second step) – see Def. A.3 for a detailed description and Fig. 3 for the associated computational graph. As these two intermediate integrator time steps are needed to accurately define the RHEL learning rule in discrete time, we explicitly denote them as fractional time units:

$$\mathcal{M}_{H,\delta}: \quad \mathbf{\Phi}_k \longrightarrow \mathbf{\Phi}_{k+1/3} \longrightarrow \mathbf{\Phi}_{k+2/3} \longrightarrow \mathbf{\Phi}_{k+1} \tag{10}$$

We call such models *Hamiltonian Recurrent Units* (HRUs). Therefore by design, the time-reversal symmetry property defined in Eq. (2) extends in discrete time to HRUs (Corollary A.6).

**Surrogate learning problem.** Given this modelling choice, the continuous-time problem introduced in Eq. (3) naturally translates here in discrete time as:

$$\min_{\boldsymbol{\theta}} L := \sum_{k=-K+1}^{0} \ell[\mathbf{\Phi}_k, k] \quad \text{s.t.} \quad \mathbf{\Phi}_{k+1} = \mathcal{M}_{H,\delta}[\mathbf{\Phi}_k, \boldsymbol{\theta}, \boldsymbol{u}_k] \quad \forall k = -K \cdots -1 \tag{11}$$

**Algorithmic baseline.** Eq. (11) defines a classical RNN learning problem where *Backpropagation Through Time* (BPTT), *i.e.* the instantiation of backward-mode AD in this context, is a natural algorithmic baseline. BPTT simply amounts to apply the "chain rule" backward through the computational graph spanned by the forward pass of a HRU (Eq. (9)). Alternatively, it can also be regarded as the discrete counterpart of the continuous ASM previously introduced (Theorem A.3). For this reason, we re-use the same notations as above and define the following quantities associated to BPTT:

$$d_{\boldsymbol{\theta}} L = \sum_{k=0}^{K-1} g_{\boldsymbol{\theta}}(k), \quad g_{\boldsymbol{\theta}}(k) := d_{\boldsymbol{\theta}_k} L, \quad g_{\boldsymbol{u}}(k) := d_{\boldsymbol{u}_{-k}} L, \quad \boldsymbol{\lambda}_k := d_{\mathbf{\Phi}_{-k}} L, \tag{12}$$

where $g_{\boldsymbol{\theta}}(k)$, $g_{\boldsymbol{u}}(k)$ and $\boldsymbol{\lambda}_k$ denote the "sensitivity" of the loss $L$ to $\boldsymbol{\theta}$ at time step $k$, $\boldsymbol{u}_{-k}$ and $\mathbf{\Phi}_{-k}$ respectively – see App. A.2.2 for a more detailed definition and derivation of BPTT.

**RHEL in discrete time.** Finally, we extend RHEL in discrete-time by defining the echo dynamics on HRUs as:

$$\begin{cases} \mathbf{\Phi}_0^e(\epsilon) = \mathbf{\Phi}_0^\star + \epsilon \boldsymbol{\Sigma}_x \cdot \nabla_{\mathbf{\Phi}} \ell[\mathbf{\Phi}_0, 0], \\ \mathbf{\Phi}_{k+1}^e(\epsilon) = \mathcal{M}_{H,\delta}[\mathbf{\Phi}_k^e(\epsilon), \boldsymbol{\theta}, \boldsymbol{u}_{-(k+1)}] - \epsilon \boldsymbol{J} \cdot \nabla_{\mathbf{\Phi}^e} \ell[\mathbf{\Phi}_{k+1}^e, -(k+1)] \quad \forall k = 0, \cdots, K-1 \end{cases} \tag{13}$$

Denoting:

$$H^{1/2}[\mathbf{\Phi}_k^e(\epsilon), \boldsymbol{\theta}, \boldsymbol{u}_{-(k+1)}] := \frac{1}{2}\left( H[\mathbf{\Phi}_{k+1/3}^e(\epsilon), \boldsymbol{\theta}, \boldsymbol{u}_{-(k+1)}] + H[\mathbf{\Phi}_{k+2/3}^e(\epsilon), \boldsymbol{\theta}, \boldsymbol{u}_{-(k+1)}] \right), \tag{14}$$

we can define the discrete-time counterpart of Eq. (7) as:

$$\begin{cases} \Delta_{\boldsymbol{\theta}}^{\text{RHEL}}(k, \epsilon) := -\frac{\delta}{2\epsilon}\left( \nabla_2 H^{1/2}[\mathbf{\Phi}_k^e(\epsilon), \boldsymbol{\theta}, \boldsymbol{u}_{-(k+1)}] - \nabla_2 H^{1/2}[\mathbf{\Phi}_k^e(-\epsilon), \boldsymbol{\theta}, \boldsymbol{u}_{-(k+1)}] \right) \\ \Delta_{\boldsymbol{u}}^{\text{RHEL}}(k, \epsilon) := -\frac{\delta}{2\epsilon}\left( \nabla_3 H^{1/2}[\mathbf{\Phi}_k^e(\epsilon), \boldsymbol{\theta}, \boldsymbol{u}_{-(k+1)}] - \nabla_3 H^{1/2}[\mathbf{\Phi}_k^e(-\epsilon), \boldsymbol{\theta}, \boldsymbol{u}_{-(k+1)}] \right) \\ \Delta_{\mathbf{\Phi}}^{\text{RHEL}}(k, \epsilon) := \frac{1}{2\epsilon}\boldsymbol{\Sigma}_x \cdot \left( \mathbf{\Phi}_k^e(\epsilon) - \mathbf{\Phi}_k^e(-\epsilon) \right) \end{cases},$$

and consequently extend Theorem 3.1 in discrete time as well.

**Theorem 3.2** (Informal). *Under mild assumptions on the Hamiltonian function and with $(\lambda_k)_{k\in[0,K-1]}$, $(g_{\boldsymbol{\theta}}(k))_{k\in[0,K-1]}$ and $(g_{\boldsymbol{u}}(k))_{k\in[0,K-1]}$ defined by Eq. (12), BPTT and RHEL are equivalent at all times in the sense that $\forall k = 0, \cdots, K$:*

$$\boldsymbol{\lambda}_k = \lim_{\epsilon\to 0}\Delta_{\boldsymbol{\Phi}}^{\mathrm{RHEL}}(k,\epsilon), \quad g_{\boldsymbol{\theta}}(k) = \lim_{\epsilon\to 0}\Delta_{\boldsymbol{\theta}}^{\mathrm{RHEL}}(k,\epsilon), \quad g_{\boldsymbol{u}}(k) = \lim_{\epsilon\to 0}\Delta_{\boldsymbol{u}}^{\mathrm{RHEL}}(k,\epsilon).$$

*The full formal statement and proof are provided in Appendix A.2.3.*

### 3.3 Learning stacks of HRUs *via* RHL chaining

**Vectorized notations.** Given a discrete vector field $(V_k)_{k\in[\![-K,0]\!]}$, we denote $\overline{V} := (V_{-K}^{\top}, \cdots, V_0^{\top})^{\top} \in \mathbb{R}^{T\times d_V}$ and $\widetilde{V} := (V_0^{\top}, \cdots, V_{-K}^{\top})^{\top} \in \mathbb{R}^{T\times d_V}$ the forward and backward *trajectories* associated with $V$. We define the vectorized operator $\mathcal{M}_{H,\delta}$ such that the dynamics of a single HRU unit as defined in Eq. (9) and the the nudged dynamics prescribed by RHEL as given in Eq. (13) rewrite more compactly as:

$$\overline{\boldsymbol{\Phi}} = \mathcal{M}_{H,\delta}[\overline{\boldsymbol{\Phi}}, \boldsymbol{\theta}, \overline{\boldsymbol{u}}], \quad \overline{\boldsymbol{\Phi}}^e = \mathcal{M}_{H,\delta}[\overline{\boldsymbol{\Phi}}^e, \boldsymbol{\theta}, \widetilde{\boldsymbol{u}}] - \epsilon \boldsymbol{J} \cdot \nabla_{\overline{\boldsymbol{\Phi}}^e}\widetilde{L}[\overline{\boldsymbol{\Phi}}^e].$$

**Learning Hamiltonian State Space Models (HSSMs) as multi-level optimization.** We now consider *hierarchical* models reading as a composition of $N$ HRU units of the form:

$$\boldsymbol{\Phi}^{(0)} := \overline{\boldsymbol{u}}, \quad \forall \ell \in [\![0, N-1]\!] : \overline{\boldsymbol{\Phi}}^{(\ell+1)} = \mathcal{M}_{H^{(\ell)},\delta}^{(\ell)}[\overline{\boldsymbol{\Phi}}^{(\ell+1)}, \boldsymbol{\theta}^{(\ell)}, \overline{\boldsymbol{\Phi}}^{(\ell)}], \tag{15}$$

where the trajectory of the $(\ell)$-th HRU unit is fed as an input sequence into the $(\ell+1)$-th HRU unit (Fig. 2). We call such hierarchies of HRUs *Hamiltonian State Space Models* (HSSMs). The corresponding optimization problem reads:

$$\min_{\boldsymbol{\theta}} L := \sum_{k=-K+1}^{0} \ell[\boldsymbol{\Phi}_k^{(N)}, k] \quad \text{s.t.} \quad \overline{\boldsymbol{\Phi}}^{(\ell+1)} = \mathcal{M}_{H^{(\ell)},\delta}[\overline{\boldsymbol{\Phi}}^{(\ell+1)}, \boldsymbol{\theta}^{(\ell)}, \overline{\boldsymbol{\Phi}}^{(\ell)}] \quad \forall \ell \in [\![0, N-1]\!] \tag{16}$$

Alg. 1 prescribes an intuitive receipe to compute $d_{\boldsymbol{\theta}}L$ for the above optimization problem (Eq. (16)), chaining RHEL backward through HRUs. Namely, the echo dynamics of the top-most HRU read as Eq. (13) using the initial learning signal $\nabla_{\boldsymbol{\Phi}}L$ to nudge its trajectory. On top of estimating its parameter gradients, we also estimate its *input* gradients which are used to nudge the echo dynamics of the preceding HRU. This procedure is repeated until reaching the first HRU – see Fig. 2.

---

**Algorithm 1** Recurrent Hamiltonian Echo Learning (RHEL) on a single HRU

*Inputs*: $\boldsymbol{\Phi}_0$ (final state of the forward trajectory), $\Delta_{\boldsymbol{\Phi}}$ (incoming gradient), $\epsilon$ (nudging strength), $\delta$ (timestep)
*Outputs*: $\Delta\boldsymbol{\theta}$ (parameter gradient estimate), $\Delta_{\boldsymbol{u}}$ (input gradient estimate)

1: $\Delta\boldsymbol{\theta} \leftarrow \mathbf{0}_{d_{\boldsymbol{\theta}}}$
2: $\Delta_{\boldsymbol{u}} := (\Delta_{\boldsymbol{u}_{-K}}, \cdots, \Delta_{\boldsymbol{u}_0}) \leftarrow \mathbf{0}_{K\times d_{\boldsymbol{\Phi}}}$
3: $\boldsymbol{\Phi}_{\pm\epsilon}^e \leftarrow \boldsymbol{\Phi}_0^{\star} \pm \epsilon\boldsymbol{\Sigma}_x \cdot \Delta_{\boldsymbol{\Phi},0}$                        $\triangleright$ Compute $\boldsymbol{\Phi}_{\pm\epsilon}^e$ in parallel
4: **for** k in $0, \cdots, K$ **do**
5:      $\boldsymbol{\Phi}_{\pm\epsilon}^e \leftarrow \mathcal{M}_{H,\delta}[\boldsymbol{\Phi}_{\pm\epsilon}^e, \boldsymbol{\theta}, \boldsymbol{u}_{-(k+1)}] \pm \epsilon\boldsymbol{\Sigma}_x \cdot \Delta_{\boldsymbol{\Phi},-(k+1)}$
6:      $\Delta\boldsymbol{\theta} \leftarrow \Delta\boldsymbol{\theta} - \frac{\delta}{2\epsilon}\left(\nabla_2 H^{1/2}[\boldsymbol{\Phi}_{\epsilon}^e, \boldsymbol{\theta}, \boldsymbol{u}_{-(k+1)}] - \nabla_2 H^{1/2}[\boldsymbol{\Phi}_{-\epsilon}^e, \boldsymbol{\theta}, \boldsymbol{u}_{-(k+1)}]\right)$     $\triangleright$ Eq. (14)
7:      $\Delta_{\boldsymbol{u}_{-k}} \leftarrow -\frac{\delta}{2\epsilon}\left(\nabla_3 H^{1/2}[\boldsymbol{\Phi}_{\epsilon}^e, \boldsymbol{\theta}, \boldsymbol{u}_{-(k+1)}] - \nabla_3 H^{1/2}[\boldsymbol{\Phi}_{-\epsilon}^e, \boldsymbol{\theta}, \boldsymbol{u}_{-(k+1)}]\right)$
8: **end for**
9: **return** $\Delta\boldsymbol{\theta}, \Delta_{\boldsymbol{u}}$

---

**Theorem 3.3** (Informal). *Given $\overline{\boldsymbol{u}}$ an input sequence and a HSSM with layer-wise Hamiltonians $\{H^{(\ell)}\}_{k\in[1,N]}$, applying Alg. 1 recursively backward from the top-most HRU solves the multilevel optimization problem defined in Eq. (16) in the sense that:*

$$\forall \ell = 0, \cdots, N : \quad d_{\boldsymbol{\theta}^{(\ell)}}L = \lim_{\epsilon\to 0}\Delta\boldsymbol{\theta}^{(\ell)}(\epsilon)$$

*The full formal statement and proof are provided in Appendix A.4.*

# 4 Experiments

**Foreword.** We first introduce the two types HRUs at use, empirically assess the validity of Theorem 3.3 by statically comparing gradients computed by RHEL and BPTT on HSSMs made up of these two types of HRUs, then perform training experiments across classification and regression sequence tasks. Importantly, the goal of the proposed experiments is *not* to improve the SOTA performance on these tasks. Rather we want to show, on a *given* model satisfying the requirements of RHEL (*i.e.* being a HRU or a stack thereof), that RHEL maintains training performance *with respect to BPTT*.

**Linear HRU block.** Following recent work [29], we introduce the *linear* HRU block with the parametrization $\boldsymbol{\theta}^{(\ell)} = \{\boldsymbol{A}^{(\ell)}, \boldsymbol{B}^{(\ell)}, \boldsymbol{W}^{(\ell)}\}$ and Hamiltonian:

$$
\begin{cases}
H^{(\ell)}[\boldsymbol{\Phi}^{(\ell)}, \boldsymbol{\theta}^{(\ell)}, \boldsymbol{\Phi}^{(\ell-1)}] = \frac{1}{2}\|\boldsymbol{\pi}^{(\ell)}\|^2 + \frac{1}{2}\boldsymbol{\phi}^{(\ell)^\top} \cdot \boldsymbol{A}^{(\ell)} \cdot \boldsymbol{\phi}^{(\ell)} - \boldsymbol{\phi}^{(\ell)^\top} \cdot \boldsymbol{B}^{(\ell)} \cdot \boldsymbol{u}^{(\ell)}, \\
\boldsymbol{u}^{(\ell)} = F[\boldsymbol{\phi}^{(\ell-1)}, \boldsymbol{W}^{(\ell)}],
\end{cases}
\tag{17}
$$

where $\boldsymbol{A} \in \mathbb{R}^{(d_\Phi/2)\times(d_\Phi/2)}$ is assumed to be *diagonal* and to have positive entries, $\boldsymbol{B}^{(\ell)} \in \mathbb{R}^{(d_\Phi/2)\times d_u}$, $\alpha \in \mathbb{R}$, and $F$ is the nonlinear spatial building block parametrized by $\boldsymbol{W}^{(\ell)}$ (see App. A.4.3). Under these assumptions, it can be shown that such HRUs have a bounded spectrum and yield HSSMs which are universal approximators of continuous and causal operators between time-series [29].

**Nonlinear HRU block.** To demonstrate the validity of our approach beyond linear HRUs, we also introduce a *nonlinear* HRU block [30] with $\boldsymbol{\theta}^{(\ell)} = \{\boldsymbol{A}^{(\ell)}, \boldsymbol{W}^{(\ell)}, \boldsymbol{B}^{(\ell)}, \boldsymbol{b}^{(\ell)}, \alpha^{(\ell)}\}$ and:

$$
\begin{cases}
H^{(\ell)}[\boldsymbol{\Phi}^{(\ell)}, \boldsymbol{\theta}^{(\ell)}, \boldsymbol{\Phi}^{(\ell-1)}] = \frac{1}{2}\|\boldsymbol{\pi}^{(\ell)}\|^2 + \frac{\alpha^{(\ell)}}{2}\|\boldsymbol{\phi}^{(\ell)}\|^2 \\
\quad + \left(\boldsymbol{A}^{(\ell)}\right)^{-\top} \cdot \log\left(\cosh\left(\boldsymbol{A}^{(\ell)} \cdot \boldsymbol{\phi}^{(\ell)} + \boldsymbol{B}^{(\ell)} \cdot \boldsymbol{u}^{(\ell)} + \boldsymbol{b}^{(\ell)}\right)\right), \\
\boldsymbol{u}^{(\ell)} = F[\boldsymbol{\phi}^{(\ell-1)}, \boldsymbol{W}^{(\ell)}],
\end{cases}
\tag{18}
$$

where $\boldsymbol{A} \in \mathbb{R}^{(d_\Phi/2)\times(d_\Phi/2)}$ is also assumed to be *diagonal*, $\boldsymbol{B}^{(\ell)} \in \mathbb{R}^{(d_\Phi/2)\times d_u}$, $\boldsymbol{b}^{(\ell)} \in \mathbb{R}^{(d_\Phi/2)}$, $\alpha \in \mathbb{R}$, and $F$ is the nonlinear spatial building block parametrized by $\boldsymbol{W}^{(\ell)}$ (see App. A.4.5). It has been shown that these HRUs mitigate by design vanishing and exploding gradients [30]; we provide empirical validation in App. A.5.5..

## 4.1 Static gradient comparison

As a sanity check of Theorem 3.3 and preamble to training experiments, we first check that RHEL gradients are computed correctly. Given a HSSM model, we randomly sample a tuple $(\boldsymbol{u}_i, \boldsymbol{y}_i) \sim \mathcal{D}$ from the SCP1 dataset (see App. A.5.1 for details on this dataset) and pass $\boldsymbol{u}_i$ through the model. Given $\boldsymbol{y}$, we then run BPTT and RHEL through the model and compare them in terms of cosine similarity and norm ratio of the resulting layer-wise parameter gradients. We run this experiment on a HSSM made up of six linear HRU blocks (which we call "linear HSSM") and another HSSM comprising six nonlinear HRU blocks (resp. "nonlinear HSSM") and obtain Fig. 4. We observe that in terms of these two comparison metrics, RHEL and BPTT parameter gradients are near-perfectly aligned for all parameters and across all HRU blocks, for both the linear and nonlinear HSSMs.

## 4.2 Training experiments

**Classification.** To further check our theoretical guarantees, we now perform training experiments with BPTT and RHEL across six multivariate sequence classification datasets with various sequence lengths (from $\sim 400$ to $\sim 18k$) and number of classes, on linear and nonlinear HSSMs (see App. A.5.1 for details on these datasets) and display our results in Table 1. This series of tasks was recently introduced [34] as a subset of the University of East Anglia (UEA) datasets [35] with the longest sequences for increased difficulty and recently used to benchmark the linear HSSM previously introduced [29]. We observe that models trained by RHEL almost match, on average across the datasets, those trained by BPTT in terms of resulting performance on the test dataset.

**Regression & scalability.** Finally, to assess the applicability of RHEL beyond classification and scalability to longer sequences, we run training experiments on the PPG-DaLiA dataset, a multivariate

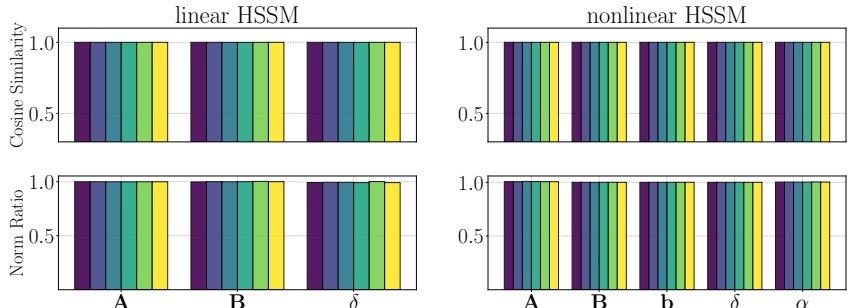

Figure 4: **Static comparison between RHEL and BPTT.** Given some $(\boldsymbol{u}_i, \boldsymbol{y}_i) \sim \mathcal{D}$, we perform BPTT and RHEL on six blocks-deep linear and nonlinear HSSMs. We measure, layer-wise (first layer in purple), the cosine similarity (top panels) and norm ratio (bottom pannels) between RHEL and BPTT parameter gradients of a linear HSSM (Eq. (17)) and a nonlinear HSSM, Eq. (18)).

| Tasks | | Worms | SCP1 | SCP2 | Ethanol | Heartbeat | Motor | |
|---|---|---|---|---|---|---|---|---|
| Seq. length | | 17,984 | 896 | 1,152 | 1,751 | 405 | 3,000 | |
| # classes | | 5 | 2 | 2 | 4 | 2 | 2 | **Avg** |
| Lin | BPTT | $78.3^{\pm 7.5}$ | $86.4^{\pm 1.8}$ | $63.9^{\pm 7.3}$ | $29.9^{\pm 0.6}$ | $73.9^{\pm 0.6}$ | $48.1^{\pm 5.7}$ | $\mathbf{63.4^{\pm 3.9}}$ |
| | RHEL | $75.0^{\pm 9.9}$ | $86.1^{\pm 2.9}$ | $61.4^{\pm 9.4}$ | $29.9^{\pm 0.6}$ | $73.5^{\pm 1.6}$ | $51.6^{\pm 5.0}$ | $\mathbf{62.9^{\pm 4.9}}$ |
| Nonlin | BPTT | $51.1^{\pm 7.2}$ | $86.8^{\pm 3.2}$ | $54.0^{\pm 4.9}$ | $29.9^{\pm 0.6}$ | $74.5^{\pm 2.4}$ | $56.5^{\pm 7.6}$ | $\mathbf{58.8^{\pm 4.3}}$ |
| | RHEL | $50.6^{\pm 6.7}$ | $85.6^{\pm 4.4}$ | $54.0^{\pm 2.0}$ | $29.9^{\pm 0.6}$ | $73.9^{\pm 4.3}$ | $53.0^{\pm 5.7}$ | $\mathbf{57.8^{\pm 4.0}}$ |

Table 1: Test mean accuracy (%, higher is better) across five different seeds ($\pm$ indicates standard deviation) using RHEL and BPTT, nonlinear and linear HSSMs, on six UEA time series classification datasets with various sequence length and number of classes.

time series regression dataset designed for heart rate prediction using data collected from a wrist-worn device [36]. With sequence length of $\sim 50k$, this task is considered to be a difficult "long-range" benchmark [29]. We display in Table 2 the results obtained when training linear and nonlinear HSSMs with RHEL and BPTT on this dataset. Here again, we observe that on average and with both models, the performance of RHEL matches that of BPTT.

## 5 Discussion

**Limitations.** We observe on Tables 1–2 that RHEL slightly underperforms BPTT, though this gap is statistically significant only for nonlinear HSSMs on regression tasks. This performance difference can be attributed to: i) the approximation bias introduced by finite nudging in our method, a known issue in related works [37], and ii) numerical precision that requires some careful selection of the nudging strength (see App. A.5.4).

**"Physical computing"?** Every logically irreversible computation, be it on a digital accelerator or any alternative substrate, is sustained by a physical process carried by wires and transistors which fundamentally generates heat [38]. Therefore the very notion of "physical computing" may sound pleonastic. However, digital hardware designs abstract core physics away to enable idealized boolean logic relying (among many other things) upon *statelessness, unidirectionality, determinism and synchro-*

| | | PPG-DaLiA |
|---|---|---|
| Seq. length | | 49,920 |
| Input/output dim. | | 6/1 |
| Lin | BPTT | $9.1^{\pm 1.1}$ |
| | RHEL | $9.5^{\pm 1.0}$ |
| Nonlin | BPTT | $7.8^{\pm 0.5}$ |
| | RHEL | $8.4^{\pm 0.5}$ |

Table 2: Test average mean-square error ($\times 10^{-2}$, lower is better) across five different seeds ($\pm$ indicates standard deviation) applying RHEL and BPTT to train nonlinear and linear HSSMs on the PPG-DaLiA dataset.

*nization* [39]. Compilation pipelines are then typically designed to be as hardware-agnostic and as general-purpose as possible [40]. In this generalistic computing paradigm, one of the most popular practices of "hardware-software" codesign is to write bespoke low-level kernels for specific workloads to mitigate their compute, memory costs and off-chip memory accesses [6, 41, 42], yet remaining largely physics-agnostic. Conversely, physical computing firstly targets *Application-Specific Integrated Circuits* (ASICs) rather than general-purpose machines [39]. Secondly, the aforementioned digital design constraints may be relaxed and the resulting *analog physics* leveraged. Supply voltages may be decreased and the resulting circuits become non-deterministic and be leveraged for stochastic algorithms [43]. Continuous-valued currents may be used for vector–matrix multiplication [44], matrix inversion [45], nearest neighbor search [46] within resistive systems. Constrained [47] or combinatorial [48] optimization problems may be embodied into a physical system and subsequently solved as the system settles to equilibrium– see more examples in recent review works [11, 39].

**Self-learning machines and Hamiltonian systems in ML.**    As a particular instantiation of physical computing, *self-learning machines* are physical embodiments of neural networks whose inference and gradient computation are, for instance, carried out by relaxing to equilibrium [49, 50]. In this regard, RHEL extends the design of scalable self-learning machines beyond *dissipative* systems emulating *implicit differentiation* [51–55]. In comparison to these algorithms, RHEL crucially avoids prohibitively long simulation times due to the use of lengthy root-finding algorithms, extends their core mechanics to the broader domain of sequence modelling and possibly larger-scale tasks. Finally, on top of its direct connection with HEB [10], RHEL belongs to a large body Hamiltonian-based ML algorithms dedicated to physics-informed neural networks [56–58], memory-efficient models [30], generative models [59], sampling [60–62] and optimization [63] algorithms.

**Forward-only learning.**    In mainstream ML, *zeroth-order optimization* (ZO) techniques are pursued for their memory efficiency compared to vanilla backprop with gradient checkpointing [64], which can be further enhanced with appropriate quantization schemes [65]. The relatively recent exploration of ZO techniques on Large Language Models (LLMs) finetuning [25] has spurred a revival of "forward-only" techniques for memory-cheap gradient computation in LLMs on a variety of applications. However, the quest for forward-only algorithms in physical computing is *not* primarily motivated by memory efficiency. In our context, "forward-only" means that both the forward and backward passes obey the *same physical principle*, for instance obeying to the same Hamiltonian dynamics as required by RHEL.

**Future work.**    The restriction of RHEL to non-dissipative systems limits model expressivity [29]. While inducing artificial dissipation through the use dissipative integrators [29] or by embedding a dissipative system within a larger conservative one [10] are convenient workarounds, there is ultimately a need for a temporal credit assignment algorithm tailored for *truly* dissipative systems. In Appendix A.3.1, we show that even when assuming that the time-reversed trajectory of a dissipative system is physically feasible, perturbations of this trajectory do *not* encode ASM error signals. Additionally and in the spirit of HEB, an idealized version of RHEL should be entirely "black-box" as it still requires explicit Hamiltonian parameter gradients to implement its learning rule (Alg. 1). This could possibly be achieved for instance using homeostatic control *via* control knobs acting on the model parameters [66] – we sketch inside Appendix A.3.2 what such a procedure would look like. Another exciting direction of work is to have RHEL operate *online* – it currently requires one forward pass and subsequent ones revisiting the inputs in reverse order. Finally, while HRU hidden units can be recomputed from their final state [30], it still requires storing the input sequence of the HRU so that memory gains are not evident within a stack of HRUs. Looking beyond analog physical systems, endowing RHEL with better memory efficiency and online mechanics could potentially yield a compelling alternative to BPTT on *digital* accelerators like GPUs [14].

**Conclusion.**    While we *simulated* Hamiltonian dynamics alongside RHEL on GPUs, the greatest potential of RHEL lies within *real* Hamiltonian systems, namely on-chip photonic circuits [67], superconducting circuits [68], or spintronic systems [69]. A possible physical implementation of a HSSM would map its feedforward components onto *digital* circuits (running backprop) and its recurrent components onto *analog* circuits (running RHEL) [54] – see Remark 8 in Appendix. We hope this work will incentivize the search for alternative analog hardware capable of training sequence models with much higher energy efficiency, as well as alternative algorithms for digital hardware.

## Acknowledgments and Disclosure of Funding

The authors warmly thank (in alphabetical order) Aditya Gilra, Albert Cohen, Debabrota Basu and James Laudon for their support of the project. ME thanks Vincent Roulet for his careful review which greatly helped improve the quality of this manuscript, as well as Amaury Hayat and Alexandre Krajenbrink for useful discussions. We also thank the anonymous reviewers for the very interesting research questions they raised and for greatly helping improve the clarity of the manuscript. Lastly, we thank the donors of the UEA data who indirectly contributed to facilitate the assessment of our algorithm on long-range tasks. GP acknowledges the CHIST-ERA grant for the CausalXRL project (CHIST-ERA-19-XAI-002) by L'Agence Nationale de la Recherche, France (grant reference ANR-21-CHR4-0007), and the ANR JCJC for the REPUBLIC project (ANR-22-CE23-0003-01).

## Author contributions

GP and ME collaborated closely on all aspects of this work. Both authors contributed to study design, with GP focusing primarily on experiments and code development, and ME focusing primarily on theoretical development and writing.

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

# A   Supplementary material

**Contents**

## A.1 Theoretical results in continuous time

**Summary.** In this section, we present all the results derived in *continuous* time. More precisely:

- We formally define our model (Def. A.1) and constrained optimization problem (Def. A.2) in *continuous time*.

- We state and prove our algorithm baseline, the *continuous adjoint state method* (ASM) for this constrained optimization problem (Theorem A.1). To ease the comparison of the continuous ASM with *Recurrent Hamiltonian Echo Learning* (RHEL) in continuous time, we state re-parametrized version of the continuous ASM where time is indexed *backwards* (Corollary A.1). Finally, we also state a variant of the continuous ASM when the loss function is only defined at the *final* timestep (Corollary A.2) to ease the comparison between the continuous ASM and *Hamiltonian Echo Backprop* (HEB, [10]).

- We introduce two technical results (Lemma A.1–A.2) which enable us to prove the *time-reversal invariance* property of our model (Lemma A.3). We then show that the direct consequence of this property is the *time-reversibily* of our model upon momentum flipping (Corollary A.3), the key mechanics which fundamentally underpins our algorithm. All these intermediate results allow us to finally introduce RHEL in continuous time and prove its equivalence with the continuous ASM (Theorem A.2).

- Lastly, we connect HEB [10] with the continuous ASM when the loss is defined only at the final time step (Theorem A.4). We highlight some key differences between HEB and RHEL.

### A.1.1 Definitions & assumptions

**Definition A.1** (Continuous Hamiltonian model). *Given $\boldsymbol{\theta} \in \mathbb{R}^{d_{\boldsymbol{\theta}}}$, $T \in \mathbb{R}_{+}^{\star}$ and an input sequence $t \to \boldsymbol{u}(t) \in \left(\mathbb{R}^{d_{\boldsymbol{u}}}\right)^{[-T,0]}$, the continuous Hamiltonian model prediction $t \to \boldsymbol{\Phi}(t) \in \left(\mathbb{R}^{d_{\boldsymbol{\Phi}}}\right)^{[-T,0]}$ is, by definition, implicitly given as the solution of the following ODE:*

$$\boldsymbol{\Phi}(-T) = \boldsymbol{x}, \quad \forall t \in [-T, 0] : \partial_t \boldsymbol{\Phi}(t) = \boldsymbol{J} \cdot \nabla_{\boldsymbol{\Phi}} H[\boldsymbol{\Phi}(t), \boldsymbol{\theta}, \boldsymbol{u}(t)], \quad \boldsymbol{J} := \begin{bmatrix} \boldsymbol{0} & \boldsymbol{I} \\ -\boldsymbol{I} & \boldsymbol{0} \end{bmatrix}.$$

*We assume that:*

1. *$H$ is time-reversal invariant:*

$$\forall \boldsymbol{\Phi} \in \mathbb{R}^{d_{\boldsymbol{\Phi}}}, \quad \forall \boldsymbol{\theta} \in \mathbb{R}^{d_{\boldsymbol{\theta}}}, \forall \boldsymbol{u} \in \mathbb{R}^{d_{\boldsymbol{u}}} : \quad H[\boldsymbol{\Phi}, \boldsymbol{\theta}, \boldsymbol{u}] = H[\boldsymbol{\Sigma}_z \cdot \boldsymbol{\Phi}, \boldsymbol{\theta}, \boldsymbol{u}], \quad \boldsymbol{\Sigma}_z := \begin{bmatrix} \boldsymbol{I} & \boldsymbol{0} \\ \boldsymbol{0} & -\boldsymbol{I} \end{bmatrix}$$

2. *$\boldsymbol{\Phi} \to H[\boldsymbol{\Phi}, \cdot, \cdot]$ is twice continuously differentiable,*
3. *$\boldsymbol{\theta} \to H[\cdot, \boldsymbol{\theta}, \cdot]$ is differentiable,*
4. *$\nabla_{1,2}^2 H$ exists and is continuous with respect to $t$,*
5. *$\boldsymbol{u} \to H[\cdot, \cdot, \boldsymbol{u}]$ is continuous,*
6. *$\boldsymbol{\Phi} \to \nabla_1 H[\boldsymbol{\Phi}, \cdot, \cdot]$ and $\boldsymbol{\Phi} \to \nabla_{2,1} H[\boldsymbol{\Phi}, \cdot, \cdot]$ are Lipschitz continuous.*

**Remark 1.** *While some of these assumptions will be used explicitly in our derivations, they are all needed to guarantee the existence of partial derivatives of $s$ as an* implicit function *of $\boldsymbol{x}$ and $\boldsymbol{\theta}$ through Eq. (1) and we refer to [28] for such claims given these assumptions.*

**Definition A.2** (Continuous constrained optimization optimization problem). *Given a continuous Hamiltonian model (Def. A.1), we consider the following constrained optimization problem:*

$$\min_{\boldsymbol{\theta}} L := \int_{-T}^{0} dt \ell[t, \boldsymbol{\Phi}(t), \boldsymbol{\theta}] \quad s.t. \quad \forall t \in [-T, 0] : \partial_t \boldsymbol{\Phi}(t) = \boldsymbol{J} \cdot \nabla_{\boldsymbol{\Phi}} H[\boldsymbol{\Phi}(t), \boldsymbol{\theta}, \boldsymbol{u}(t)],$$

*where we assume that:*

1. *$\ell$ is time-reversal invariant:*

$$\forall \boldsymbol{\Phi} \in \mathbb{R}^{d_{\boldsymbol{\Phi}}}, \quad \forall \boldsymbol{\theta} \in \mathbb{R}^{d_{\boldsymbol{\theta}}}, \forall t \in [-T, 0] : \quad \ell[t, \boldsymbol{\Phi}, \boldsymbol{\theta}] = \ell[t, \boldsymbol{\Sigma}_z \cdot \boldsymbol{\Phi}, \boldsymbol{\theta}]$$

2. *$t \to \ell[t, \cdot, \cdot]$ is continuous,*
3. *$\boldsymbol{\theta} \to \ell[\cdot, \cdot, \boldsymbol{\theta}]$ is differentiable,*
4. *$t \to \nabla_{\boldsymbol{\theta}} \ell[t, \cdot, \cdot]$ is continuous.*
5. *$\boldsymbol{\Phi} \to \ell[\cdot, \boldsymbol{\Phi}, \cdot]$ is twice differentiable,*

**Remark 2.** *Note that while we did not assume in the main part of this manuscript that $\ell$ depended on $\boldsymbol{\theta}$, we assume it in the appendix for the generality of our derivations.*

### A.1.2 Proof of the continuous adjoint state method (ASM)

**Theorem A.1** (Continuous adjoint state method ([28])). *Given assumptions A.1–A.2, the gradients of $L$ with respect to $\boldsymbol{\theta}$ and $\boldsymbol{x}$ are given by:*

$$d_{\boldsymbol{\theta}} L = \int_{-T}^{0} g_{\boldsymbol{\theta}}(t) dt, \quad d_x L = \boldsymbol{\lambda}(0)$$

*with $t \to g_{\boldsymbol{\theta}} \in \left(\mathbb{R}^{d_{\boldsymbol{\theta}}}\right)^{[-T, 0]}$ defined as:*

$$g_{\boldsymbol{\theta}}(t) := \nabla_{\boldsymbol{\theta}} \ell[t, \boldsymbol{\Phi}(t), \boldsymbol{\theta}] + \nabla^2_{\boldsymbol{\Phi}, \boldsymbol{\theta}} H[\boldsymbol{\Phi}(t), \boldsymbol{\theta}, \boldsymbol{u}(t)] \cdot \boldsymbol{J}^{\top} \cdot \boldsymbol{\lambda}(t) \quad \forall t \in [-T, 0]$$

*and $\boldsymbol{\lambda}$ solving for the **adjoint** ODE:*

$$\begin{cases} \boldsymbol{\lambda}(0) & = \boldsymbol{0} \\ \partial_t \boldsymbol{\lambda}(t) & = -\nabla^2_{\boldsymbol{\Phi}} H[\boldsymbol{\Phi}(t), \boldsymbol{\theta}, \boldsymbol{u}(t)] \cdot \boldsymbol{J}^{\top} \cdot \boldsymbol{\lambda}(t) - \nabla_{\boldsymbol{\Phi}} \ell[t, \boldsymbol{\Phi}(t), \boldsymbol{\theta}] \end{cases}$$

*Proof of Theorem A.1.* With slight adaptations, our proof mostly follows that of [32] and also *assume the existence of the partial derivatives of $\boldsymbol{\Phi} = \boldsymbol{\Phi}(t, \boldsymbol{x}, \boldsymbol{\theta})$ as an *implicit* function of $t, \boldsymbol{x}$ and $\boldsymbol{\theta}$ – we defer to [28] for the proof of this claim.

We start off defining the Lagrangian associated with the constraint optimization problem:

$$\mathcal{L}(\boldsymbol{\Phi}, \boldsymbol{\lambda}, \boldsymbol{\theta}, \boldsymbol{u}) := \int_{-T}^{0} dt \left( \ell[t, \boldsymbol{\Phi}(t, \boldsymbol{\theta}), \boldsymbol{\theta}] + \boldsymbol{\lambda}^{\top}(t) \cdot (\boldsymbol{J} \cdot \nabla_{\boldsymbol{\Phi}} H[\boldsymbol{\Phi}(t, \boldsymbol{\theta}), \boldsymbol{\theta}, \boldsymbol{u}(t)] - \partial_t \boldsymbol{\Phi}(t, \boldsymbol{\theta})) \right),$$

where $t \to \boldsymbol{\lambda}(t) \in \left(\mathbb{R}^{d_{\boldsymbol{\Phi}}}\right)^{[-T, 0]}$ denotes the Lagrangian multiplier associated to the constraint.

**Derivation of $d_{\boldsymbol{\theta}} L$.** For readability, we emphasize the dependence of $\boldsymbol{\Phi}$ on $t$ and $\boldsymbol{\theta}$, as we will leverage the existence of its partial derivatives with respect to these variables. Then, given Assumptions A.1, the total derivative of the Lagrangian with respect to $\boldsymbol{\theta}$ exists and reads:

$$d_{\boldsymbol{\theta}} \mathcal{L} = \int_{-T}^{0} dt \left( \partial_{\boldsymbol{\theta}} \boldsymbol{\Phi}(t, \boldsymbol{\theta})^{\top} \cdot \nabla_{\boldsymbol{\Phi}} \ell[t, \boldsymbol{\Phi}(t, \boldsymbol{\theta}), \boldsymbol{\theta}] + \nabla_{\boldsymbol{\theta}} \ell[t, \boldsymbol{\Phi}(t, \boldsymbol{\theta}), \boldsymbol{\theta}] \right)$$

$$+ \int_{-T}^{0} dt \left[ \partial_{\boldsymbol{\theta}} \boldsymbol{\Phi}(t, \boldsymbol{\theta})^{\top} \cdot \nabla^2_{\boldsymbol{\Phi}} H[\boldsymbol{\Phi}(t, \boldsymbol{\theta}), \boldsymbol{\theta}, \boldsymbol{u}(t)] \cdot \boldsymbol{J}^{\top} + \nabla^2_{\boldsymbol{\theta}, \boldsymbol{\Phi}} H[\boldsymbol{\Phi}(t, \boldsymbol{\theta}), \boldsymbol{\theta}, \boldsymbol{u}(t)]^{\top} \cdot \boldsymbol{J}^{\top} - \partial^2_{\boldsymbol{\theta}, t} \boldsymbol{\Phi}(t, \boldsymbol{\theta})^{\top} \right] \cdot \boldsymbol{\lambda}(t).$$

We can transform the last term of the integrand with $\partial_{\boldsymbol{\theta},t}\boldsymbol{\Phi}(t,\boldsymbol{\theta})$ by applying Schwartz's theorem and and integration by parts as:

$$-\int_{-T}^{0} dt\partial_{\boldsymbol{\theta},t}\boldsymbol{\Phi}(t,\boldsymbol{\theta})^{\top}\cdot\boldsymbol{\lambda}(t) = -\int_{-T}^{0} dt\partial_{t,\boldsymbol{\theta}}\boldsymbol{\Phi}(t,\boldsymbol{\theta})^{\top}\cdot\boldsymbol{\lambda}(t)$$

$$= \left[-\partial_{\boldsymbol{\theta}}\boldsymbol{\Phi}(t,\boldsymbol{\theta})^{\top}\cdot\boldsymbol{\lambda}(t)\right]_{-T}^{0} + \int_{-T}^{0} dt\partial_{\boldsymbol{\theta}}\boldsymbol{\Phi}(t,\boldsymbol{\theta})^{\top}\cdot\partial_t\boldsymbol{\lambda}(t)$$

$$= -\partial_{\boldsymbol{\theta}}\boldsymbol{\Phi}(0,\boldsymbol{\theta})^{\top}\cdot\boldsymbol{\lambda}(0) + \int_{-T}^{0} dt\partial_{\boldsymbol{\theta}}\boldsymbol{\Phi}(t,\boldsymbol{\theta})^{\top}\cdot\partial_t\boldsymbol{\lambda}(t)$$

where the contribution of the first term at $t=-T$ vanishes because:

$$\boldsymbol{\Phi}(-T,\boldsymbol{\theta}) = \boldsymbol{x} \Rightarrow \partial_{\boldsymbol{\theta}}\boldsymbol{\Phi}(0,\boldsymbol{\theta}) = 0$$

Plugging this back into $d_{\boldsymbol{\theta}}\mathcal{L}$ yields:

$$d_{\boldsymbol{\theta}}\mathcal{L} = -\partial_{\boldsymbol{\theta}}\boldsymbol{\Phi}(0,\boldsymbol{\theta})^{\top}\cdot\boldsymbol{\lambda}(0)$$
$$+ \int_{-T}^{0} dt\partial_{\boldsymbol{\theta}}\boldsymbol{\Phi}(t,\boldsymbol{x},\boldsymbol{\theta})^{\top}\cdot\left[\nabla_{\boldsymbol{\Phi}}\ell[t,\boldsymbol{\Phi}(t,\boldsymbol{\theta}),\boldsymbol{\theta}] + \nabla_{\boldsymbol{\Phi}}^2 H[\boldsymbol{\Phi}(t,\boldsymbol{x},\boldsymbol{\theta}),\boldsymbol{\theta}]\cdot\boldsymbol{J}^{\top}\cdot\boldsymbol{\lambda}(t) + \partial_t\boldsymbol{\lambda}(t)\right]$$
$$+ \int_{-T}^{0} dt\left(\nabla_{\boldsymbol{\theta}}\ell[t,\boldsymbol{\Phi}(t,\boldsymbol{\theta}),\boldsymbol{\theta}] + \nabla_{\boldsymbol{\Phi},\boldsymbol{\theta}}^2 H[\boldsymbol{\Phi}(t,\boldsymbol{x},\boldsymbol{\theta}),\boldsymbol{\theta}]\cdot\boldsymbol{J}^{\top}\cdot\boldsymbol{\lambda}(t)\right)$$

Denoting $\boldsymbol{\Phi}_*$ the solution of the (primal) ODE and by defining $\boldsymbol{\lambda}_*$ as the solution of the adjoint ODE:

$$\partial_t\boldsymbol{\lambda}_*(t) = -\nabla_{\boldsymbol{\Phi}}^2 H[\boldsymbol{\Phi}_*(t,\boldsymbol{\theta}),\boldsymbol{\theta},\boldsymbol{u}(t)]\cdot\boldsymbol{J}^{\top}\cdot\boldsymbol{\lambda}_*(t) - \nabla_{\boldsymbol{\Phi}}\ell[t,\boldsymbol{\Phi}_*(t,\boldsymbol{\theta}),\boldsymbol{\theta}], \quad \boldsymbol{\lambda}_*(0) = 0$$

we have:

$$d_{\boldsymbol{\theta}}L := d_{\boldsymbol{\theta}}\int_{-T}^{0} dt\ell[t,\boldsymbol{\Phi}(t,\boldsymbol{\theta}),\boldsymbol{\theta}]$$
$$= d_{\boldsymbol{\theta}}\mathcal{L}(\boldsymbol{\Phi}_*,\boldsymbol{\lambda}_*,\boldsymbol{\theta})$$
$$= \int_{-T}^{0} dt\left(\nabla_{\boldsymbol{\theta}}\ell[t,\boldsymbol{\Phi}_*(t,\boldsymbol{\theta}),\boldsymbol{\theta}] + \nabla_{\boldsymbol{\Phi},\boldsymbol{\theta}}^2 H[\boldsymbol{\Phi}_*(t,\boldsymbol{x},\boldsymbol{\theta}),\boldsymbol{\theta}]\cdot\boldsymbol{J}^{\top}\cdot\boldsymbol{\lambda}_*(t)\right)$$

**Derivation of $d_{\boldsymbol{x}}L$.** Similarly, we can prove that:

$$d_{\boldsymbol{x}}\mathcal{L} = \partial_{\boldsymbol{x}}\boldsymbol{\Phi}(0,\boldsymbol{x})^{\top}\cdot\left[\nabla\ell[\boldsymbol{\Phi}(0,\boldsymbol{x})] - \boldsymbol{\lambda}(0)\right]$$
$$+ \int_{-T}^{0} dt\partial_{\boldsymbol{x}}\boldsymbol{\Phi}(t,\boldsymbol{x})^{\top}\cdot\left[\nabla_{\boldsymbol{\Phi}}\ell[t,\boldsymbol{\Phi}(t,\boldsymbol{\theta}),\boldsymbol{\theta}] + \nabla_{\boldsymbol{\Phi}}^2 H[\boldsymbol{\Phi}(t,\boldsymbol{x}),\boldsymbol{\theta},\boldsymbol{u}(t)]\cdot\boldsymbol{J}^{\top}\cdot\boldsymbol{\lambda}(t) + \partial_t\boldsymbol{\lambda}(t)\right]$$
$$+ \boldsymbol{\lambda}(0)$$

Using the same $\boldsymbol{\Phi}_*$ and $\boldsymbol{\lambda}_*$, we get $d_{\boldsymbol{x}}L = \boldsymbol{\lambda}(0)$. $\qquad\square$

**Reparametrization of the continuous adjoint method.** To ease the comparison of the continuous adjoint state method with our algorithm, we slightly reparametrize the variables introduced in Theorem A.1.

**Corollary A.1.** *Under the same assumptions as Theorem A.1, the gradients of $L$ with respect to $\boldsymbol{\theta}$ and $\boldsymbol{x}$ are given by:*

$$d_{\boldsymbol{\theta}}L = \int_0^T g_{\boldsymbol{\theta}}(t)dt, \quad d_{\boldsymbol{x}}L = \boldsymbol{\lambda}(T)$$

*with $t\to g_{\boldsymbol{\theta}}\in\left(\mathbb{R}^{d_{\boldsymbol{\theta}}}\right)^{[0,T]}$ defined as:*

$$g_{\boldsymbol{\theta}}(t) := \nabla_{\boldsymbol{\theta}}\ell[-t,\boldsymbol{\Phi}(-t),\boldsymbol{\theta}] + \nabla_{\boldsymbol{\Phi},\boldsymbol{\theta}}^2 H[\boldsymbol{\Phi}(-t),\boldsymbol{\theta},\boldsymbol{u}(-t)]\cdot\boldsymbol{J}^{\top}\cdot\boldsymbol{\lambda}(t) \quad \forall t\in[0,T]$$

*and $\boldsymbol{\lambda}$ solving for the **adjoint** ODE:*

$$\begin{cases} \boldsymbol{\lambda}(0) &= \boldsymbol{0} \\ \partial_t\boldsymbol{\lambda}(t) &= \nabla_{\boldsymbol{\Phi}}^2 H[\boldsymbol{\Phi}(-t),\boldsymbol{\theta},\boldsymbol{u}(-t)]\cdot\boldsymbol{J}^{\top}\cdot\boldsymbol{\lambda}(t) + \nabla_{\boldsymbol{\Phi}}\ell[-t,\boldsymbol{\Phi}(-t),\boldsymbol{\theta}] \end{cases}$$

*Proof of Corollary A.1.* Immediately stems from Theorem A.1. $\qquad\square$

**Edge case: loss at the final timestep only.** The backward ODE can be differently parametrized when a loss is defined only at the last time step. We need this formulation for later convenience.

**Corollary A.2.** *Under the same assumptions as Theorem A.1, and assuming:*

$$\ell[t, \cdot, \cdot] = 0 \quad \forall t \in [0, T), \quad \ell[T, \cdot, \cdot] := \ell_T,$$

*the gradients of $\ell_T$ with respect to $\boldsymbol{\theta}$ and $\boldsymbol{x}$ are given by:*

$$d_{\boldsymbol{\theta}} \ell_T[\boldsymbol{\Phi}(0), \boldsymbol{\theta}] = \nabla_{\boldsymbol{\theta}} \ell_T[\boldsymbol{\Phi}(0), \boldsymbol{\theta}] + \int_0^T g_{\boldsymbol{\theta}}(t) dt, \quad d_x L = \boldsymbol{\lambda}(T)$$

*with $t \to g_{\boldsymbol{\theta}} \in \left(\mathbb{R}^{d_{\boldsymbol{\theta}}}\right)^{[0,T]}$ defined as:*

$$g_{\boldsymbol{\theta}}(t) := \nabla^2_{\boldsymbol{\Phi}, \boldsymbol{\theta}} H[\boldsymbol{\Phi}(-t), \boldsymbol{\theta}, \boldsymbol{u}(-t)] \cdot \boldsymbol{J}^\top \cdot \boldsymbol{\lambda}(t) \quad \forall t \in [0, T]$$

*and $\boldsymbol{\lambda}$ solving for the **adjoint** ODE:*

$$\begin{cases} \boldsymbol{\lambda}(0) &= \nabla_{\boldsymbol{\Phi}} \ell_T[\boldsymbol{\Phi}(0), \boldsymbol{\theta}] \\ \partial_t \boldsymbol{\lambda}(t) &= \nabla^2_{\boldsymbol{\Phi}} H[\boldsymbol{\Phi}(-t), \boldsymbol{\theta}, \boldsymbol{u}(-t)] \cdot \boldsymbol{J}^\top \cdot \boldsymbol{\lambda}(t) \end{cases}$$

*Proof of Corollary A.2.* Starting from the Lagrangian:

$$\mathcal{L}(\boldsymbol{\Phi}, \boldsymbol{\lambda}, \boldsymbol{\theta}, \boldsymbol{u}) := \ell_T[\boldsymbol{\Phi}(0, \boldsymbol{\theta}), \boldsymbol{\theta}] + \int_{-T}^0 dt \left(\boldsymbol{\lambda}^\top(t) \cdot (\boldsymbol{J} \cdot \nabla_{\boldsymbol{\Phi}} H[\boldsymbol{\Phi}(t, \boldsymbol{\theta}), \boldsymbol{\theta}, \boldsymbol{u}(t)] - \partial_t \boldsymbol{\Phi}(t, \boldsymbol{\theta}))\right),$$

the proof reads in the exact same fashion as that of Theorem A.1. $\qquad\square$

### A.1.3 Proof of Theorem 3.1

**Technical Lemmas.** We first introduce two technical Lemmas which will be needed for the derivation of our main result.

**Lemma A.1** (Block-wise Pauli matrices and associated properties). *Defining $\boldsymbol{\Sigma}_x, \boldsymbol{\Sigma}_y, \boldsymbol{\Sigma}_z \in \mathbb{R}^{d_\Phi \times d_\Phi}$ as:*

$$
\boldsymbol{\Sigma}_x := \begin{bmatrix} \boldsymbol{0} & \boldsymbol{I}_{d_\Phi/2} \\ \boldsymbol{I}_{d_\Phi/2} & \boldsymbol{0} \end{bmatrix}, \quad \boldsymbol{\Sigma}_y := \begin{bmatrix} \boldsymbol{0} & -i\boldsymbol{I}_{d_\Phi/2} \\ i\boldsymbol{I}_{d_\Phi/2} & \boldsymbol{0} \end{bmatrix}, \quad \boldsymbol{\Sigma}_z := \begin{bmatrix} \boldsymbol{I}_{d_\Phi/2} & \boldsymbol{0} \\ \boldsymbol{0} & -\boldsymbol{I}_{d_\Phi/2} \end{bmatrix}
$$

*where $i$ denotes the imaginary unit, the following equalities hold:*

1. $\boldsymbol{J} = i\boldsymbol{\Sigma}_y$
2. $\boldsymbol{\Sigma}_x^2 = \boldsymbol{\Sigma}_y^2 = \boldsymbol{\Sigma}_z^2 = \boldsymbol{I}_{d_\Phi/2}$,
3. $\boldsymbol{\Sigma}_x \cdot \boldsymbol{\Sigma}_y = i\boldsymbol{\Sigma}_z, \boldsymbol{\Sigma}_y \cdot \boldsymbol{\Sigma}_z = i\boldsymbol{\Sigma}_x, \boldsymbol{\Sigma}_z \cdot \boldsymbol{\Sigma}_x = i\boldsymbol{\Sigma}_y$,
4. $\boldsymbol{\Sigma}_i \cdot \boldsymbol{\Sigma}_j = -\boldsymbol{\Sigma}_j \cdot \boldsymbol{\Sigma}_i$ *for any $i \neq j \in \{x, y, z\}$.*

*Proof of Lemma A.1.* Because of the block-wise structure of $\boldsymbol{\Sigma}_x, \boldsymbol{\Sigma}_y, \boldsymbol{\Sigma}_z$, these equalities can be easily checked. $\qquad\square$

**Lemma A.2.** *Under the assumptions of Def. A.1, the following equalities hold for all $\boldsymbol{\Phi}, \boldsymbol{\theta}, \boldsymbol{u}$:*

$$
\nabla_{\boldsymbol{\Phi}} H[\boldsymbol{\Phi}, \boldsymbol{\theta}, \boldsymbol{u}] = \boldsymbol{\Sigma}_z \cdot \nabla_{\boldsymbol{\Phi}^\star} H[\boldsymbol{\Phi}^\star, \boldsymbol{\theta}, \boldsymbol{u}],
$$
$$
\nabla_{\boldsymbol{\Phi}}^2 H[\boldsymbol{\Phi}, \boldsymbol{\theta}, \boldsymbol{u}] = \boldsymbol{\Sigma}_z \cdot \nabla_{\boldsymbol{\Phi}^\star}^2 H[\boldsymbol{\Phi}^\star, \boldsymbol{\theta}, \boldsymbol{u}] \cdot \boldsymbol{\Sigma}_z,
$$
$$
\nabla_{\boldsymbol{\Phi}, \boldsymbol{\theta}} H[\boldsymbol{\Phi}, \boldsymbol{\theta}, \boldsymbol{u}] = \nabla_{\boldsymbol{\Phi}^\star, \boldsymbol{\theta}} H[\boldsymbol{\Phi}^\star, \boldsymbol{\theta}, \boldsymbol{u}] \cdot \boldsymbol{\Sigma}_z
$$

*Proof of Lemma A.2.* The above equalities can be simply obtained by differentiating through the time-reversal invariance hypothesis – which is possible because of the differentiability of $H$ with respect to $\boldsymbol{\Phi}$ and $\boldsymbol{\theta}$ – and using the chain rule. Namely, given some $\boldsymbol{\Phi}, \boldsymbol{\theta}, \boldsymbol{u}$:

$$
\begin{aligned}
\nabla_{\boldsymbol{\Phi}} H[\boldsymbol{\Phi}, \boldsymbol{\theta}, \boldsymbol{u}] &= \nabla_{\boldsymbol{\Phi}} \left( H[\boldsymbol{\Phi}^\star, \boldsymbol{\theta}, \boldsymbol{u}] \right) \\
&= \left( \partial_{\boldsymbol{\Phi}} \boldsymbol{\Phi}^\star \right)^\top \cdot \nabla_{\boldsymbol{\Phi}^\star} H[\boldsymbol{\Phi}^\star, \boldsymbol{\theta}, \boldsymbol{u}] \\
&= \boldsymbol{\Sigma}_z^\top \cdot \nabla_{\boldsymbol{\Phi}^\star} H[\boldsymbol{\Phi}^\star, \boldsymbol{\theta}, \boldsymbol{u}] = \boldsymbol{\Sigma}_z \cdot \nabla_{\boldsymbol{\Phi}^\star} H[\boldsymbol{\Phi}^\star, \boldsymbol{\theta}, \boldsymbol{u}],
\end{aligned}
$$

since $\boldsymbol{\Phi}^\star := \boldsymbol{\Sigma}_z \cdot \boldsymbol{\Phi}$. The other equalities are derived in the same way. $\qquad\square$

**Time-reversal invariance.** We highlight here how the assumption $H[\boldsymbol{\Phi}, \cdot, \cdot] = H[\boldsymbol{\Sigma}_z \cdot \boldsymbol{\Phi}, \cdot, \cdot]$ given inside Def. A.1 entails time-reversal invariance of the dynamics – up to time-reversal the input sequence.

**Lemma A.3** (Time-reversal invariance of the dynamics). *Under the assumptions given in Def. A.1, if $\boldsymbol{\Phi}$ the solution of the ODE:*

$$
\partial_t \boldsymbol{\Phi}(t) = \boldsymbol{J} \cdot \nabla_{\boldsymbol{\Phi}} H[\boldsymbol{\Phi}(t), \boldsymbol{\theta}, \boldsymbol{u}(t)],
$$

*the function $\widetilde{\boldsymbol{\Phi}}^\star : t \to \boldsymbol{\Sigma}_z \cdot \boldsymbol{\Phi}(-t) = [\boldsymbol{\phi}^\top(-t), -\boldsymbol{\pi}^\top(-t)]^\top$ is solution of the same ODE with the time-reversed input sequence $t \to \widetilde{\boldsymbol{u}}(t) := \boldsymbol{u}(-t)$.*

*Proof.* Let $t \in [-T, 0]$. We have:

$$
\begin{aligned}
\partial_t \widetilde{\boldsymbol{\Phi}}^\star(t) &= -\boldsymbol{\Sigma}_z \cdot \partial_t \boldsymbol{\Phi}(-t) && \text{(Def. of } \widetilde{\boldsymbol{\Phi}}^\star \text{ and chain rule)} \\
&= -\boldsymbol{\Sigma}_z \cdot \boldsymbol{J} \cdot \nabla_{\boldsymbol{\Phi}} H[\boldsymbol{\Phi}(-t), \boldsymbol{\theta}, \boldsymbol{u}(-t)] && \text{(by assumption)} \\
&= +\boldsymbol{J} \cdot \boldsymbol{\Sigma}_z \cdot \nabla_{\boldsymbol{\Phi}} H[\boldsymbol{\Phi}(-t), \boldsymbol{\theta}, \boldsymbol{u}(-t)] && \text{(Lemma A.1)} \\
&= \boldsymbol{J} \cdot \boldsymbol{\Sigma}_z^2 \cdot \nabla_{\boldsymbol{\Phi}^\star} H[\boldsymbol{\Phi}^\star(-t), \boldsymbol{\theta}, \boldsymbol{u}(-t)] && \text{(Lemma A.2)} \\
&= \boldsymbol{J} \cdot \nabla_{\boldsymbol{\Phi}^\star} H[\widetilde{\boldsymbol{\Phi}}^\star(t), \boldsymbol{\theta}, \boldsymbol{u}(-t)] && \text{(Lemma A.1)}
\end{aligned}
$$

$\qquad\square$

**Time reversibility.** A direct consequence of the time-reversal of the dynamics under consideration (Lemma A.3) is *time reversibility*: upon flipping the momentum of $\boldsymbol{\Phi}$ at time $t = 0$ ($\boldsymbol{\pi}(0) \leftarrow -\boldsymbol{\pi}(0)$) and presenting the input sequence in reversed order, the system evolves backward to its initial state.

**Corollary A.3** (Reversibility of the dynamics). *Under the same assumptions as Lemma A.3, we define* $\boldsymbol{\Phi}^e$ *as the solution of the ODE:*

$$\boldsymbol{\Phi}^e(0) = \boldsymbol{\Phi}^\star(0), \quad \partial_t \boldsymbol{\Phi}^e(t) = \boldsymbol{J} \cdot \nabla_{\boldsymbol{\Phi}^e} H[\boldsymbol{\Phi}^e(t), \boldsymbol{\theta}, \boldsymbol{u}(-t)] \quad \forall t \in [0, T].$$

*Then:*

$$\forall t \in [0, T]: \quad \boldsymbol{\Phi}^e(t) = \widetilde{\boldsymbol{\Phi}}^\star(t)$$

*Proof.* $\boldsymbol{\Phi}^e$ and $\widetilde{\boldsymbol{\Phi}}^\star$ satisfy: i) the same initial conditions ($\widetilde{\boldsymbol{\Phi}}^\star(0) = \boldsymbol{\Phi}^\star(0) = \boldsymbol{\Phi}^e(0)$), ii) the same ODE (Lemma A.3), therefore by unicity of the solution of the ODE, they are equal at all time over the domain of definition of $\boldsymbol{\Phi}$. $\qquad\square$

**Main result.** We are now ready to state and demonstrate our main result in continuous time.

**Theorem A.2** (Equivalence between RHEL and the continuous ASM). *Under the assumptions of Def. A.1 and Def. A.2, let* $\boldsymbol{\Phi}$ *be the solution of the ODE for* $t \in [-T, 0]$:

$$\boldsymbol{\Phi}(-T) = \boldsymbol{x}, \quad \partial_t \boldsymbol{\Phi}(t) = \boldsymbol{J} \cdot \nabla_{\boldsymbol{\Phi}} H[\boldsymbol{\Phi}(t), \boldsymbol{\theta}, \boldsymbol{u}(t)].$$

*Given* $\boldsymbol{\Phi}$, *let* $\boldsymbol{\Phi}^e$ *be defined as the solution of the other ODE for* $t \in [0, T]$:

$$\boldsymbol{\Phi}^e(0) = \boldsymbol{\Phi}^\star(0), \quad \partial_t \boldsymbol{\Phi}^e(t, \epsilon) = \boldsymbol{J} \nabla_{\boldsymbol{\Phi}^e} H[\boldsymbol{\Phi}^e(t, \epsilon), \boldsymbol{\theta}, \boldsymbol{u}(-t)] - \epsilon \boldsymbol{J} \nabla_{\boldsymbol{\Phi}^e} \ell[-t, \boldsymbol{\Phi}^e(t, \epsilon), \boldsymbol{\theta}].$$

*Defining:*

$$\Delta_{\boldsymbol{\theta}}^{\text{RHEL}}(t, \epsilon) := \nabla_{\boldsymbol{\theta}} \ell[-t, \boldsymbol{\Phi}^e(t, \epsilon), \boldsymbol{\theta}] + \frac{1}{2\epsilon} \left( \nabla_{\boldsymbol{\theta}} H[\boldsymbol{\Phi}^e(t, \epsilon), \boldsymbol{\theta}, \boldsymbol{u}(-t)] - \nabla_{\boldsymbol{\theta}} H[\boldsymbol{\Phi}^e(t, -\epsilon), \boldsymbol{\theta}, \boldsymbol{u}(-t)] \right),$$

$$\Delta_{\boldsymbol{\Phi}}^{\text{RHEL}}(t, \epsilon) := \frac{1}{2\epsilon} \boldsymbol{\Sigma}_x \cdot \left( \boldsymbol{\Phi}^e(t, \epsilon) - \boldsymbol{\Phi}^e(t, -\epsilon) \right),$$

*we have:*

$$\forall t \in [0, T], \quad \boldsymbol{\lambda}(t) = \lim_{\epsilon \to 0} \Delta_{\boldsymbol{\Phi}}^{\text{RHEL}}(t, \epsilon), \quad g_{\boldsymbol{\theta}}(t) = \lim_{\epsilon \to 0} \Delta_{\boldsymbol{\theta}}^{\text{RHEL}}(t, \epsilon)$$

*where* $\boldsymbol{\lambda}$ *and* $g_{\boldsymbol{\theta}}$ *are defined in Corollary A.1.*

*Proof.* Defining $\Delta_{\boldsymbol{\Phi}}^{\text{RHEL}}(t) := \lim_{\epsilon \to 0} \Delta_{\boldsymbol{\Phi}}^{\text{RHEL}}(t, \epsilon)$ and $\Delta_{\boldsymbol{\theta}}^{\text{RHEL}}(t) := \lim_{\epsilon \to 0} \Delta_{\boldsymbol{\theta}}^{\text{RHEL}}(t, \epsilon)$, note that:

$$\Delta_{\boldsymbol{\Phi}}^{\text{RHEL}}(t) = \boldsymbol{\Sigma}_x \cdot \partial_\epsilon \boldsymbol{\Phi}^e(t, \epsilon)|_{\epsilon=0}$$
$$\Delta_{\boldsymbol{\theta}}^{\text{RHEL}}(t) = \nabla_{\boldsymbol{\theta}} \ell[-t, \boldsymbol{\Phi}^e(t, 0), \boldsymbol{\theta}] + \partial_\epsilon \left( \nabla_{\boldsymbol{\theta}} H[\boldsymbol{\Phi}^e(t, \epsilon), \boldsymbol{\theta}, \boldsymbol{u}(-t)] \right)|_{\epsilon=0}$$

**Derivation of** $\boldsymbol{\lambda}(t) = \lim_{\epsilon \to 0} \Delta_{\boldsymbol{\Phi}}^{\text{RHEL}}(t, \epsilon)$. Given $t \in [0, T]$, differentiating the ODE satisfied by $\boldsymbol{\Phi}^e$ with respect to $\epsilon$ at $\epsilon = 0$ yields:

$$\partial_t (\partial_\epsilon \boldsymbol{\Phi}^e(t, \epsilon)|_{\epsilon=0}) = \partial_\epsilon (\partial_t \boldsymbol{\Phi}^e(t, \epsilon))|_{\epsilon=0} \qquad \text{(Schwartz Theorem)}$$
$$= \partial_\epsilon (\boldsymbol{J} \cdot \nabla_{\boldsymbol{\Phi}^e} H[\boldsymbol{\Phi}^e(t, \epsilon), \boldsymbol{\theta}, \boldsymbol{u}(t)] - \epsilon \boldsymbol{J} \nabla_{\boldsymbol{\Phi}^e} \ell[t, \boldsymbol{\Phi}^e(t, \epsilon), \boldsymbol{\theta}])|_{\epsilon=0}$$
$$= \boldsymbol{J} \cdot \nabla_{\boldsymbol{\Phi}^e}^2 H[\boldsymbol{\Phi}^e(t, 0)] \cdot \partial_\epsilon \boldsymbol{\Phi}^e(t, \epsilon)|_{\epsilon=0} - \boldsymbol{J} \nabla_{\boldsymbol{\Phi}^e} \ell[t, \boldsymbol{\Phi}^e(t, 0), \boldsymbol{\theta}].$$

By Lemma A.3:

$$\boldsymbol{\Phi}^e(t, 0) = \widetilde{\boldsymbol{\Phi}}^\star(t) \quad \forall t \in [0, T],$$

therefore:

$$\partial_t (\partial_\epsilon \boldsymbol{\Phi}^e(t, \epsilon)|_{\epsilon=0}) = \boldsymbol{J} \cdot \nabla_{\boldsymbol{\Phi}^\star}^2 H[\boldsymbol{\Phi}^\star(-t)] \cdot \partial_\epsilon \boldsymbol{\Phi}^e(t, \epsilon)|_{\epsilon=0} - \boldsymbol{J} \nabla_{\boldsymbol{\Phi}^\star} \ell[t, \boldsymbol{\Phi}^\star(-t), \boldsymbol{\theta}]$$
$$= \boldsymbol{J} \cdot \boldsymbol{\Sigma}_z \cdot \nabla_{\boldsymbol{\Phi}}^2 H[\boldsymbol{\Phi}(-t, 0)] \cdot \boldsymbol{\Sigma}_z \cdot \partial_\epsilon \boldsymbol{\Phi}^e(t, \epsilon)|_{\epsilon=0} - \boldsymbol{J} \cdot \boldsymbol{\Sigma}_z \cdot \nabla_{\boldsymbol{\Phi}} \ell[t, \boldsymbol{\Phi}(-t), \boldsymbol{\theta}] \qquad \text{(Lemma A.2)}$$

Additionally, note that we have by Lemma A.1:

$$\boldsymbol{J} \cdot \boldsymbol{\Sigma}_z = -\boldsymbol{\Sigma}_x, \quad \boldsymbol{\Sigma}_z = \boldsymbol{J} \cdot \boldsymbol{\Sigma}_x,$$

so that:

$$\partial_t(\partial_\epsilon \boldsymbol{\Phi}^e(t,\epsilon)|_{\epsilon=0}) = -\boldsymbol{\Sigma}_x \cdot \nabla^2_{\boldsymbol{\Phi}} H[\boldsymbol{\Phi}(-t,0)] \cdot (\boldsymbol{J} \cdot \boldsymbol{\Sigma}_x) \cdot \partial_\epsilon \boldsymbol{\Phi}^e(t,\epsilon)|_{\epsilon=0} + \boldsymbol{\Sigma}_x \cdot \nabla_{\boldsymbol{\Phi}} \ell[t, \boldsymbol{\Phi}(-t), \boldsymbol{\theta}] \quad (19)$$

Left multiplying Eq. (19) on both sides by $\boldsymbol{\Sigma}_x$ yields:

$$\begin{aligned}
\partial_t \left( \boldsymbol{\Sigma}_x \cdot \partial_\epsilon \boldsymbol{\Phi}^e(t,\epsilon)|_{\epsilon=0} \right) &= -\boldsymbol{\Sigma}_x^2 \cdot \nabla^2_{\boldsymbol{\Phi}} H[\boldsymbol{\Phi}(-t,0)] \cdot \boldsymbol{J} \cdot (\boldsymbol{\Sigma}_x \cdot \partial_\epsilon \boldsymbol{\Phi}^e(t,\epsilon)|_{\epsilon=0}) + \boldsymbol{\Sigma}_x^2 \cdot \nabla_{\boldsymbol{\Phi}} \ell[t, \boldsymbol{\Phi}(-t), \boldsymbol{\theta}] \\
&= -\nabla^2_{\boldsymbol{\Phi}} H[\boldsymbol{\Phi}(-t,0)] \cdot \boldsymbol{J} \cdot (\boldsymbol{\Sigma}_x \cdot \partial_\epsilon \boldsymbol{\Phi}^e(t,\epsilon)|_{\epsilon=0}) + \nabla_{\boldsymbol{\Phi}} \ell[t, \boldsymbol{\Phi}(-t), \boldsymbol{\theta}] \quad \text{(Lemma A.1)} \\
&= \nabla^2_{\boldsymbol{\Phi}} H[\boldsymbol{\Phi}(-t,0)] \cdot \boldsymbol{J}^\top \cdot (\boldsymbol{\Sigma}_x \cdot \partial_\epsilon \boldsymbol{\Phi}^e(t,\epsilon)|_{\epsilon=0}) + \nabla_{\boldsymbol{\Phi}} \ell[t, \boldsymbol{\Phi}(-t), \boldsymbol{\theta}] \quad (\boldsymbol{J}^\top = -\boldsymbol{J})
\end{aligned}$$

Finally, note that because $\boldsymbol{\Phi}^e(0) = \boldsymbol{\Phi}^\star$ does not depend on $\epsilon$, we have that:

$$\partial_t \left( \boldsymbol{\Sigma}_x \cdot \partial_\epsilon \boldsymbol{\Phi}^e(0,\epsilon)|_{\epsilon=0} \right) = 0,$$

so that all in all, $\Delta^{\mathrm{RHEL}}_{\boldsymbol{\Phi}}$ satisfies:

$$\begin{cases}
\Delta^{\mathrm{RHEL}}_{\boldsymbol{\Phi}}(0) &= \boldsymbol{0} \\
\partial_t \Delta^{\mathrm{RHEL}}_{\boldsymbol{\Phi}}(t) &= \nabla^2_{\boldsymbol{\Phi}} H[\boldsymbol{\Phi}(-t), \boldsymbol{\theta}, \boldsymbol{u}(-t)] \cdot \boldsymbol{J}^\top \cdot \Delta^{\mathrm{RHEL}}_{\boldsymbol{\Phi}}(t) + \nabla_{\boldsymbol{\Phi}} \ell[-t, \boldsymbol{\Phi}(-t), \boldsymbol{\theta}]
\end{cases}$$

Therefore $\Delta^{\mathrm{RHEL}}_{\boldsymbol{\Phi}}$ and $\boldsymbol{\lambda}$ (as defined in Corollary A.1) satisfy the same initial conditions and the same ODE, therefore they are equal at all times.

**Derivation of $g_{\boldsymbol{\theta}}(t) = \lim_{\epsilon \to 0} \Delta^{\mathrm{RHEL}}_{\boldsymbol{\theta}}(t,\epsilon)$.** Note that by Lemma A.3 and time-reversal invariance of $\ell$:

$$\begin{aligned}
\Delta^{\mathrm{RHEL}}_{\boldsymbol{\theta}}(t) &= \nabla_{\boldsymbol{\theta}} \ell[-t, \boldsymbol{\Phi}^\star(-t,0), \boldsymbol{\theta}] + \partial_\epsilon \left( \nabla_{\boldsymbol{\theta}} H[\boldsymbol{\Phi}^e(t,\epsilon), \boldsymbol{\theta}, \boldsymbol{u}(-t)] \right)|_{\epsilon=0} \\
&= \nabla_{\boldsymbol{\theta}} \ell[-t, \boldsymbol{\Phi}(-t,0), \boldsymbol{\theta}] + \partial_\epsilon \left( \nabla_{\boldsymbol{\theta}} H[\boldsymbol{\Phi}^e(t,\epsilon), \boldsymbol{\theta}, \boldsymbol{u}(-t)] \right)|_{\epsilon=0}
\end{aligned}$$

As the first term of $\Delta^{\mathrm{RHEL}}(t)$ and $g_{\boldsymbol{\theta}}(t)$ coincide, the remainder of the derivation focuses on the second term of $\Delta^{\mathrm{RHEL}}_{\boldsymbol{\theta}}(t)$. Given $t \in [0, T]$, we have:

$$\begin{aligned}
\nabla^2_{\boldsymbol{\Phi}, \boldsymbol{\theta}} H[\boldsymbol{\Phi}(-t), \boldsymbol{\theta}, \boldsymbol{u}(t)] \cdot \boldsymbol{J}^\top \cdot \boldsymbol{\lambda}(t) &= \nabla^2_{\boldsymbol{\Phi}^e, \boldsymbol{\theta}} H[\boldsymbol{\Phi}^e(t,0), \boldsymbol{\theta}, \boldsymbol{u}(t)] \cdot \boldsymbol{\Sigma}_z \cdot \boldsymbol{J}^\top \cdot \boldsymbol{\lambda}(t) & \text{(Lemma A.2)} \\
&= -\nabla^2_{\boldsymbol{\Phi}^e, \boldsymbol{\theta}} H[\boldsymbol{\Phi}^e(t,0), \boldsymbol{\theta}, \boldsymbol{u}(t)] \cdot \boldsymbol{\Sigma}_z \cdot i\boldsymbol{\Sigma}_y \cdot \boldsymbol{\lambda}(t) & (\boldsymbol{J} = i\boldsymbol{\Sigma}_y) \\
&= +\nabla^2_{\boldsymbol{\Phi}^e, \boldsymbol{\theta}} H[\boldsymbol{\Phi}^e(t,0), \boldsymbol{\theta}, \boldsymbol{u}(t)] \cdot i\boldsymbol{\Sigma}_y \cdot \boldsymbol{\Sigma}_z \cdot \boldsymbol{\lambda}(t) & \text{(Lemma A.1)} \\
&= -\nabla^2_{\boldsymbol{\Phi}^e, \boldsymbol{\theta}} H[\boldsymbol{\Phi}^e(t,0), \boldsymbol{\theta}, \boldsymbol{u}(t)] \cdot \boldsymbol{\Sigma}_x \cdot \boldsymbol{\lambda}(t) & \text{(Lemma A.1)} \\
&= -\nabla^2_{\boldsymbol{\Phi}^e, \boldsymbol{\theta}} H[\boldsymbol{\Phi}^e(t,0), \boldsymbol{\theta}, \boldsymbol{u}(t)] \cdot \boldsymbol{\Sigma}_x^2 \cdot \partial_\epsilon \boldsymbol{\Phi}^e(t,\epsilon)|_{\epsilon=0} & (\boldsymbol{\lambda} = \Delta^{\mathrm{RHEL}}_{\boldsymbol{\Phi}}) \\
&= -\nabla^2_{\boldsymbol{\Phi}^e, \boldsymbol{\theta}} H[\boldsymbol{\Phi}^e(t,0), \boldsymbol{\theta}, \boldsymbol{u}(t)] \cdot \partial_\epsilon \boldsymbol{\Phi}^e(t,\epsilon)|_{\epsilon=0} & \text{(Lemma A.1)} \\
&= -\partial_\epsilon \left( \nabla_{\boldsymbol{\theta}} H[\boldsymbol{\Phi}^e(t,\epsilon), \boldsymbol{\theta}, \boldsymbol{u}(t)] \right)|_{\epsilon=0},
\end{aligned}$$

which finishes to prove $g_{\boldsymbol{\theta}}(t) = \Delta^{\mathrm{RHEL}}_{\boldsymbol{\theta}}(t)$ for $t \in [0, T]$. $\qquad\square$

### A.1.4 Connection to Hamiltonian Echo Backpropagation (HEB)

**Remark 3.** *The above setup and implementation of RHEL is not exactly that of Hamiltonian Echo Backprop (HEB, [10]). In particular:*

- *the loss function in HEB is only defined at the final time step,*

- *the interaction with $\ell$ does not happen simultaneously with $H$,*

- *finally, we would like to recover the HEB formula giving the gradient estimate of the loss with respect to the initial state of the neurons.*

*In the following corollary, we make slight algorithmic adjustments to match the seminal HEB implementation as much as possible **while preserving the generality of the sequence modelling setting**, i.e. dependence of $H$ with $\boldsymbol{\theta}$ and $\boldsymbol{u}$. Note that in the seminal HEB work, $H$ does not depend on a static set of parameters $\boldsymbol{\theta}$ nor on an input sequence. $\boldsymbol{u}$.*

**Corollary A.4.** *Under the assumptions of Def. A.1 and Def. A.2, and assuming additionally:*

$$\ell[t, \cdot, \cdot] = 0 \quad \forall t \in [0, T), \quad \ell[T, \cdot, \cdot] := \ell_T,$$

*let $\boldsymbol{\Phi}$ be the solution of the ODE, for $t \in [-T, 0]$:*

$$\boldsymbol{\Phi}(-T) = \boldsymbol{x}, \quad \partial_t \boldsymbol{\Phi}(t) = \boldsymbol{J} \cdot \nabla_{\boldsymbol{\Phi}} H[\boldsymbol{\Phi}(t), \boldsymbol{\theta}, \boldsymbol{u}(t)],$$

*and the solution of another ODE, for $t \in [0, \epsilon]$:*

$$\partial_t \boldsymbol{\Phi}(t) = \boldsymbol{J} \nabla_{\boldsymbol{\Phi}} \ell_T[\boldsymbol{\Phi}(t), \boldsymbol{\theta}].$$

*Let $\boldsymbol{\Phi}^e$ be the solution of the following ODE, for $t \in [\epsilon, T]$:*

$$\boldsymbol{\Phi}^e(0, \epsilon) = \boldsymbol{\Phi}(\epsilon)^\star, \quad \partial_t \boldsymbol{\Phi}^e(t, \epsilon) = \boldsymbol{J} \nabla_{\boldsymbol{\Phi}^e} H[\boldsymbol{\Phi}^e(t, \epsilon), \boldsymbol{\theta}, \boldsymbol{u}(-t)],$$

*Defining:*

$$\Delta_{\boldsymbol{\theta}}^{\mathrm{RHEL}}(t, \epsilon) := \frac{1}{2\epsilon} \left( \nabla_{\boldsymbol{\theta}} H[\boldsymbol{\Phi}^e(t, \epsilon), \boldsymbol{\theta}, \boldsymbol{u}(-t)] - \nabla_{\boldsymbol{\theta}} H[\boldsymbol{\Phi}^e(t, -\epsilon), \boldsymbol{\theta}, \boldsymbol{u}(-t)] \right),$$

$$\Delta_{\boldsymbol{\Phi}}^{\mathrm{RHEL}}(t, \epsilon) := \frac{1}{2\epsilon} \boldsymbol{\Sigma}_x \cdot \left( \boldsymbol{\Phi}^e(t, \epsilon) - \boldsymbol{\Phi}^e(t, -\epsilon) \right),$$

*we have:*

$$\forall t \in [0, T], \quad \boldsymbol{\lambda}(t) = \lim_{\epsilon \to 0} \Delta_{\boldsymbol{\Phi}}^{\mathrm{RHEL}}(t, \epsilon), \quad g_{\boldsymbol{\theta}}(t) = \lim_{\epsilon \to 0} \Delta_{\boldsymbol{\theta}}^{\mathrm{RHEL}}(t, \epsilon)$$

*where $\boldsymbol{\lambda}$ and $g_{\boldsymbol{\theta}}$ are defined in Corollary A.2. In particular:*

$$-i\epsilon d_{\boldsymbol{x}^\star}^w \ell_T[\boldsymbol{\Phi}(0), \boldsymbol{\theta}] = \boldsymbol{\Phi}^e(T, \epsilon)^\star - \boldsymbol{\Phi}(-T) + \mathcal{O}(\epsilon^2)$$

*where $d_{\boldsymbol{x}^\star}^w \equiv d_{\boldsymbol{\Phi}}$ denotes the total Wirtinger derivative with respect to $\boldsymbol{x}^\star$ and $i$ the imaginary unit.*

*Proof of Corollary A.4.* The derivation is almost exactly similar to that of Theorem 3.1 with two key differences:

- the version of the continuous ASM against which this version of RHEL is compared is different (Corollary A.2),

- the interaction of $\boldsymbol{\Phi}$ with $\ell$ and $H$ do not happen simultaneously but on disjoint intervals.

We will simply show that the interaction with $\ell$ and conjugation $\boldsymbol{\Phi} \to \boldsymbol{\Phi}^\star$ yields the correct initial conditions and defer to the proof of Theorem 3.1 for the remainder. We will also use the same notations and denote $\Delta_{\boldsymbol{\Phi}}^{\mathrm{RHEL}}(t) := \lim_{\epsilon \to 0} \Delta_{\boldsymbol{\Phi}}^{\mathrm{RHEL}}(t, \epsilon)$, $\Delta_{\boldsymbol{\theta}}^{\mathrm{RHEL}}(t) := \lim_{\epsilon \to 0} \Delta_{\boldsymbol{\theta}}^{\mathrm{RHEL}}(t, \epsilon)$.

**Derivation of $\boldsymbol{\lambda}(t) = \lim_{\epsilon \to 0} \Delta_{\boldsymbol{\Phi}}^{\mathrm{RHEL}}(t, \epsilon)$.** Integrating the ODE satisfied by $\boldsymbol{\Phi}$ between $0$ and $T$ yields:

$$\boldsymbol{\Phi}(\epsilon) = \boldsymbol{\Phi}(0) + \int_0^\epsilon dt \boldsymbol{J} \cdot \nabla_{\boldsymbol{\Phi}} \ell[\boldsymbol{\Phi}(t), \boldsymbol{\theta}]$$

$$= \boldsymbol{\Phi}(0) + \epsilon \boldsymbol{J} \cdot \nabla_{\boldsymbol{\Phi}} \ell[\boldsymbol{\Phi}(0), \boldsymbol{\theta}] + \mathcal{O}(\epsilon^2)$$

Therefore, the initial state of $\boldsymbol{\Phi}^e$ can be written as:

$$\boldsymbol{\Phi}^e(\epsilon) = \boldsymbol{\Phi}^\star(0) + \epsilon \boldsymbol{\Sigma}_z \cdot \boldsymbol{J} \cdot \nabla_{\boldsymbol{\Phi}} \ell[\boldsymbol{\Phi}(0), \boldsymbol{\theta}] + \mathcal{O}(\epsilon^2)$$

$$= \boldsymbol{\Phi}^\star(0) + \epsilon \boldsymbol{\Sigma}_x \cdot \nabla_{\boldsymbol{\Phi}} \ell[\boldsymbol{\Phi}(0), \boldsymbol{\theta}] + \mathcal{O}(\epsilon^2) \qquad \text{(Lemma A.1)}.$$

By differentiating the last equality with respect to $\epsilon$ at $\epsilon = 0$, we obtain:

$$\Delta_{\boldsymbol{\Phi}}^{\mathrm{RHEL}}(0) = \nabla_{\boldsymbol{\Phi}} \ell[\boldsymbol{\Phi}(0), \boldsymbol{\theta}].$$

Proceeding exactly as in the proof of Theorem A.2, we obtain that:

$$\begin{cases} \Delta_{\boldsymbol{\Phi}}^{\mathrm{RHEL}}(0) & = \nabla_{\boldsymbol{\Phi}} \ell[\boldsymbol{\Phi}(0), \boldsymbol{\theta}], \\ \partial_t \Delta_{\boldsymbol{\Phi}}^{\mathrm{RHEL}}(t) & = \nabla_{\boldsymbol{\Phi}}^2 H[\boldsymbol{\Phi}(-t), \boldsymbol{\theta}, \boldsymbol{u}(-t)] \cdot \boldsymbol{J}^\top \cdot \Delta_{\boldsymbol{\Phi}}^{\mathrm{RHEL}}(t) \end{cases}$$

Therefore $\Delta_{\boldsymbol{\Phi}}^{\mathrm{RHEL}}$ and $\boldsymbol{\lambda}$ (as defined in Corollary A.2) satisfy the same initial conditions and the same ODE, therefore they are equal at all times.

**Derivation of $g_{\boldsymbol{\theta}}(t) = \lim_{\epsilon \to 0} \Delta_{\boldsymbol{\theta}}^{\mathrm{RHEL}}(t, \epsilon)$.**   See proof of Theorem A.2.

**Connection to HEB formula.**   In particular, we have:

$$\begin{aligned} d_{\boldsymbol{x}} \ell_T[\boldsymbol{\Phi}(0), \boldsymbol{\theta}] = \boldsymbol{\lambda}(T) = \boldsymbol{\Sigma}_x \cdot \partial_\epsilon \left( \boldsymbol{\Phi}(T, \epsilon) \right)|_{\epsilon=0} & \\ = \frac{1}{\epsilon} \boldsymbol{\Sigma}_x \left( \boldsymbol{\Phi}^e(T, \epsilon) - \boldsymbol{\Phi}^e(T, 0) \right) + \mathcal{O}(\epsilon) & \\ = \frac{1}{\epsilon} \boldsymbol{\Sigma}_x \left( \boldsymbol{\Phi}^e(T, \epsilon) - \boldsymbol{\Phi}^\star(-T) \right) + \mathcal{O}(\epsilon) & \quad \text{(Lemma A.3)} \\ = -\frac{i}{\epsilon} \boldsymbol{\Sigma}_y \cdot \boldsymbol{\Sigma}_z \cdot \left( \boldsymbol{\Phi}^e(T, \epsilon) - \boldsymbol{\Phi}^\star(-T) \right) + \mathcal{O}(\epsilon) & \quad \text{(Lemma A.1)} \\ = -\frac{i}{\epsilon} \boldsymbol{\Sigma}_y \cdot \left( \boldsymbol{\Phi}^e(T, \epsilon)^\star - \boldsymbol{\Phi}(-T) \right) + \mathcal{O}(\epsilon) & \end{aligned}$$

Left multiplying on both sides by $i\epsilon \boldsymbol{\Sigma}_y$ and noticing that $i d_{\boldsymbol{\Phi}^\star}^w \equiv -i \boldsymbol{\Sigma}_y \cdot d_{\boldsymbol{\Phi}}$, we finally obtain:

$$-i\epsilon d_{\boldsymbol{x}^\star}^w \ell_T[\boldsymbol{\Phi}(0), \boldsymbol{\theta}] = \boldsymbol{\Phi}^e(T, \epsilon)^\star - \boldsymbol{\Phi}(-T) + \mathcal{O}(\epsilon^2)$$

$\square$

## A.2 Theoretical results in discrete time

**Summary.**   In this section, we introduce all the results derived in *discrete* time. More precisely:

- We first formally define *Hamiltonian Recurrent Units* (HRUs, Definition A.3). HRUs can be regarded as the discrete-time counterpart of the continuous model introduced in the previous section (Definition A.1), namely as an *explicit* and *symplectic* integrator of the continuous Hamiltonian model which preserves the time-reversal invariance and time-reversibility properties in discrete time. We also introduce the constrained optimization problem naturally associated with HRUs (Definition A.4), which is the discrete time counterpart of the constrained continuous optimization problem introduced in the previous section (Definition A.2).

- We then formally define *Hamiltonian State Space Models* (HSSMs) as stacks of HRUs (Definition A.5) and the *multilevel* constrained optimization problem which is naturally associated to these models (Definition A.6).

- We state and prove our algorithmic baseline, *Backpropagation Through Time* (BPTT), through the lens of the *Lagrangian* formalism to establish a clear connection with the continuous ASM. We first introduce and derive BPTT in its general form (Theorem A.3) and then apply it more specifically to a HRU as defined in Definition A.3 (Corollary A.5).

- As we did in continuous time, we introduce a series of technical Lemmas needed to extend RHEL in discrete time. We first demonstrate the time-reversibility of HRUs on a single time step (Lemma A.4), which then enables us to extend the time reversibility property derived in continuous time (Corollary A.3) to *discrete* time (Corollary A.6). After introducing one last technical result (Lemma A.5), we then state and prove RHEL in discrete time when applied to HRUs (Corollary A.7). As the algorithm prescribed by Corollary A.7 includes solving an *implicit equation*, we finally introduce a slight practical (*i.e.* fully explicit) variant of RHEL in discrete time (Corollary A.8).

- Lastly, we show how to estimate gradients end-to-end in HSSMs by using *RHEL–chaining* (Theorem A.4). We also highlight that in practice, when using feedforward transformations across HRUs, the algorithm prescribed by Theorem A.4 implicitly requires to chain RHEL through HRUs and *automatic differentiation* through these feedforward transformations (Remark 8). This remark fundamentally underpins the actual algorithmic implementation of RHEL which was used throughout our experiments.

### A.2.1   Definitions & assumptions

**Definition A.3** (Hamiltonian Recurrent Unit). *Given $\boldsymbol{\theta} \in \mathbb{R}^{d_{\boldsymbol{\theta}}}$, $K \in \mathbb{N}^{\star}$ and an input sequence $(\boldsymbol{u}_k)_{k \in [-K, 0]} \in \left(\mathbb{R}^{d_{\boldsymbol{u}}}\right)^K$, the Hamiltonian Recurrent Unit (HRU) prediction is given by:*

$$\boldsymbol{\Phi}_{k+1} = \mathcal{M}_{H,\delta}[\boldsymbol{\Phi}_k, \boldsymbol{\theta}, \boldsymbol{u}_k] \quad \forall k = -K \cdots -1,$$

*with $H := T + V$ and:*

$$\mathcal{M}_{H,\delta} := \mathcal{M}_{T,\delta/2} \circ \mathcal{M}_{V,\delta} \circ \mathcal{M}_{T,\delta/2}, \quad \mathcal{M}_{T,\delta} := \boldsymbol{\Phi} + \delta \boldsymbol{J} \cdot \nabla_{\boldsymbol{\Phi}} T, \quad \mathcal{M}_{V,\delta} := \boldsymbol{\Phi} + \delta \boldsymbol{J} \cdot \nabla_{\boldsymbol{\Phi}} V$$

*We assume that:*

1. *$H$ is separable,* i.e. *$V$ and $T$ only depend on $\boldsymbol{\phi}$ and $\boldsymbol{\pi}$ respectively:*

$$V[\boldsymbol{\Phi}, \boldsymbol{\theta}, \boldsymbol{u}] = V[\boldsymbol{\phi}, \boldsymbol{\theta}, \boldsymbol{u}], \quad T[\boldsymbol{\Phi}, \boldsymbol{\theta}, \boldsymbol{u}] = T[\boldsymbol{\pi}, \boldsymbol{\theta}, \boldsymbol{u}]$$

2. *$T$ and $V$ are time-reversal invariant:*

$$\forall \boldsymbol{\Phi} \in \mathbb{R}^{d_{\boldsymbol{\Phi}}}, \, \forall \boldsymbol{\theta} \in \mathbb{R}^{d_{\boldsymbol{\theta}}}, \, \forall \boldsymbol{u} \in \mathbb{R}^{d_{\boldsymbol{u}}} : \quad \begin{cases} T[\boldsymbol{\Phi}, \boldsymbol{\theta}, \boldsymbol{u}] = T[\boldsymbol{\Sigma}_z \cdot \boldsymbol{\Phi}, \boldsymbol{\theta}, \boldsymbol{u}], \\ V[\boldsymbol{\Phi}, \boldsymbol{\theta}, \boldsymbol{u}] = V[\boldsymbol{\Sigma}_z \cdot \boldsymbol{\Phi}, \boldsymbol{\theta}, \boldsymbol{u}] \end{cases}$$

3. *$T$ and $V$ are twice differentiable with respect to $\boldsymbol{\Phi}$, $\boldsymbol{\theta}$ and $\boldsymbol{u}$.*

**Remark 4.** *Note that $\mathcal{M}_{H,\delta}$ is simply a Leapfrog integrator associated with $H$. We justify each of our design choices below:*

- **3 steps-parametrization.** *We write the Leapfrog integrator in a three-steps fashion to yield a reversible integrator.*

- **Separability of the Hamiltonian.** *In the case where $\phi$ and $\phi$ can be separated out in the Hamiltonian function, the Leapfrog integrator becomes explicit [33].*

- **$T$ and $V$ as functions of $\Phi$.** *Although $T$ and $V$ only depend on $\pi$ and $\phi$ respectively, we choose to define them as functions of $\Phi$ so that the proof of RHEL in the continuous case seamlessly translates to the discrete case.*

**Definition A.4** (Constrained optimization optimization problem in discrete time)**.** *Given a continuous Hamiltonian model (Def. A.1), we consider the following constrained optimization problem:*

$$\min_{\boldsymbol{\theta}} L := \sum_{k=-K+1}^{0} \ell[\boldsymbol{\Phi}_k, k] \quad s.t. \quad \boldsymbol{\Phi}_{k+1} = \mathcal{M}_{H,\delta}[\boldsymbol{\Phi}_k, \boldsymbol{\theta}, \boldsymbol{u}_k] \quad \forall k = -K \cdots -1$$

*where we assume that:*

1. *$\ell$ is time-reversal invariant:*

$$\forall \boldsymbol{\Phi} \in \mathbb{R}^{d_\Phi}, \ \forall \boldsymbol{\theta} \in \mathbb{R}^{d_\theta}, \ \forall k = -K, \cdots, 0 : \quad \ell_k[\boldsymbol{\Phi}, \boldsymbol{\theta}] = \ell_k[\boldsymbol{\Sigma}_z \cdot \boldsymbol{\Phi}, \boldsymbol{\theta}]$$

2. *$\ell$ is twice differentiable with respect to $\boldsymbol{\Phi}$ and $\boldsymbol{\theta}$.*

**Definition A.5** (Hamiltonian State Space Models)**.** *Given $(\boldsymbol{\theta}^{(1)}, \cdots, \boldsymbol{\theta}^{(N)}) \in (\mathbb{R}^{d_\theta})^N$, $K \in \mathbb{N}^\star$ and an input sequence $(\boldsymbol{u}_k)_{k \in [-K,0]} \in (\mathbb{R}^{d_u})^K$, a Hamiltonian State Space Model (HSSM) is defined as the composition of HRUs defined in Def. A.3 as:*

$$\boldsymbol{\Phi}^{(0)} := \overline{\boldsymbol{u}}, \quad \forall \ell \in [\![0, N-1]\!], \quad \forall k \in [\![-K, 0]\!] : \quad \boldsymbol{\Phi}_{k+1}^{(\ell+1)} = \mathcal{M}_{H^{(\ell)},\delta}^{(\ell)}[\boldsymbol{\Phi}_k^{(\ell+1)}, \boldsymbol{\theta}^{(\ell)}, \boldsymbol{\Phi}_k^{(\ell)}],$$

*or in a vectorized fashion as:*

$$\boldsymbol{\Phi}^{(0)} := \overline{\boldsymbol{u}}, \quad \forall \ell \in [\![0, N-1]\!] : \quad \overline{\boldsymbol{\Phi}}^{(\ell+1)} = \boldsymbol{\mathcal{M}}_{H^{(\ell)},\delta}^{(\ell)}[\overline{\boldsymbol{\Phi}}^{(\ell+1)}, \boldsymbol{\theta}^{(\ell)}, \overline{\boldsymbol{\Phi}}^{(\ell)}],$$

**Definition A.6** (Multilevel optimization problem in discrete time)**.** *Given a HSSM (Def. A.5), we consider the following constrained optimization problem:*

$$\min_{\boldsymbol{\theta}} L := \sum_{k=-K+1}^{0} \ell[\boldsymbol{\Phi}_k, k] \quad s.t. \quad \boldsymbol{\Phi}^{(0)} := \overline{\boldsymbol{u}},$$

$$\forall \ell \in [\![0, N-1]\!] : \quad \overline{\boldsymbol{\Phi}}^{(\ell+1)} = \boldsymbol{\mathcal{M}}_{H^{(\ell)},\delta}^{(\ell)}\left[\overline{\boldsymbol{\Phi}}^{(\ell+1)}, \boldsymbol{\theta}^{(\ell)}, \overline{\boldsymbol{\Phi}}^{(\ell)}\right]$$

*where we assume that $\ell$ satisfies the same assumptions as in Def. A.4.*

### A.2.2 Backpropagation Through Time (BPTT)

**General form.** We first state and prove Backpropagation Through Time (BPTT) for any integrator $\mathcal{M}_{H,\delta}$.

**Theorem A.3** (Backpropagation Through Time (BPTT)). *Given assumptions in Def. A.3–A.4, the gradients of the loss with respect to the parameters $\boldsymbol{\theta}$ and the inputs $\boldsymbol{u}_{-k}$ are given by:*

$$d_{\boldsymbol{\theta}}L = \sum_{k=0}^{K-1} g_{\boldsymbol{\theta}}(k), \quad d_{\boldsymbol{u}_{-(k+1)}}L = g_{\boldsymbol{u}}(k) \quad \forall k \in [\![0, K-1]\!],$$

*with:*

$$\begin{cases} g_{\boldsymbol{\theta}}(k) &= \nabla_2 \ell[\boldsymbol{\Phi}_{-k}, \boldsymbol{\theta}, -k] + \partial_2 \mathcal{M}_{H,\delta}[\boldsymbol{\Phi}_{-(k+1)}, \boldsymbol{\theta}, \boldsymbol{u}_{-(k+1)}]^\top \cdot \boldsymbol{\lambda}_k \\ g_{\boldsymbol{u}}(k) &= \partial_3 \mathcal{M}_{H,\delta}[\boldsymbol{\Phi}_{-(k+1)}, \boldsymbol{\theta}, \boldsymbol{u}_{-(k+1)}]^\top \cdot \boldsymbol{\lambda}_k, \end{cases}$$

*and where $(\boldsymbol{\lambda}_k)$ satisfy the following recursion relationship:*

$$\begin{cases} \boldsymbol{\lambda}_0 = \nabla_1 \ell[\boldsymbol{\Phi}_0, 0], \\ \boldsymbol{\lambda}_{k+1} = \partial_1 \mathcal{M}_{H,\delta}(\boldsymbol{\Phi}_{-(k+1)}, \boldsymbol{\theta}, \boldsymbol{u}_{-(k+1)})^\top \cdot \boldsymbol{\lambda}_k + \nabla_1 \ell[\boldsymbol{\Phi}_{-(k+1)}, -(k+1)] \quad \forall k = 0, \cdots, K-1 \end{cases}$$

*Proof of Theorem A.3.* BPTT can classically be derived through the application of the "chain rule" backward through the inference computational graph defined in Def. A.3. Another useful viewpoint though, which directly connects BPTT as the discrete counterpart of the continuous ASM and will be useful later in the appendix, is to derive it through *the method of Lagrangian multipliers*. Namely, the Lagrangian associated to the constrained optimization problem in Def. A.4 reads as:

$$\mathcal{L}(\boldsymbol{\Phi}, \boldsymbol{\lambda}, \boldsymbol{\theta}, \boldsymbol{u}) = \sum_{k=0}^{K-1} \ell[\boldsymbol{\Phi}_{-k}, \boldsymbol{\theta}, -k] + \boldsymbol{\lambda}_k^\top \cdot \left( \mathcal{M}_{H,\delta} \left[ \boldsymbol{\Phi}_{-(k+1)}, \boldsymbol{\theta}, \boldsymbol{u}_{-(k+1)} \right] - \boldsymbol{\Phi}_{-k} \right)$$

Extremizing $\mathcal{L}$ with respect to $\boldsymbol{\Phi}$ and $\boldsymbol{\lambda}$ yield $\boldsymbol{\Phi}_{k,*}$ and $\boldsymbol{\lambda}_{k,*}$:

$$\forall k = 0, \cdots, K-1 : \quad \partial_{\boldsymbol{\lambda}_k} \mathcal{L}(\boldsymbol{\Phi}_*, \boldsymbol{\lambda}_*, \boldsymbol{\theta}, \boldsymbol{u}) = \mathcal{M}_{H,\delta} \left[ \boldsymbol{\Phi}_{-(k+1),*}, \boldsymbol{\theta}, \boldsymbol{u}_{-k} \right] - \boldsymbol{\Phi}_{-k,*} = 0,$$

$$\partial_{\boldsymbol{\Phi}_0} \mathcal{L}(\boldsymbol{\Phi}_*, \boldsymbol{\lambda}_*, \boldsymbol{\theta}, \boldsymbol{u}) = \nabla_1 \ell[\boldsymbol{\Phi}_{0,*}, \boldsymbol{\theta}, 0] - \boldsymbol{\lambda}_{0,*} = 0$$

$$\forall k = 1, \cdots, K-1 : \quad \partial_{\boldsymbol{\Phi}_{-k}} \mathcal{L}(\boldsymbol{\Phi}_*, \boldsymbol{\lambda}_*, \boldsymbol{\theta}, \boldsymbol{u}) = \nabla_1 \ell[\boldsymbol{\Phi}_{-k}, \boldsymbol{\theta}, -k] + \partial_1 \mathcal{M}_{H,\delta}[\boldsymbol{\Phi}_{-k}, \boldsymbol{\theta}, \boldsymbol{u}_{-k}]^\top \cdot \boldsymbol{\lambda}_{k-1} - \boldsymbol{\lambda}_k = 0,$$

Finally, the total derivative of $L$ with respect to $\boldsymbol{\theta}$ reads as:

$$d_{\boldsymbol{\theta}}L = d_{\boldsymbol{\theta}} \mathcal{L}(\boldsymbol{\Phi}_*, \boldsymbol{\lambda}_*, \boldsymbol{\theta}, \boldsymbol{u})$$

$$= \partial_{\boldsymbol{\theta}} \mathcal{L}(\boldsymbol{\Phi}_*, \boldsymbol{\lambda}_*, \boldsymbol{\theta}, \boldsymbol{u}) + \partial_{\boldsymbol{\theta}} \boldsymbol{\Phi}_*^\top \cdot \underbrace{\partial_{\boldsymbol{\Phi}} \mathcal{L}(\boldsymbol{\Phi}_*, \boldsymbol{\lambda}_*, \boldsymbol{\theta}, \boldsymbol{u})}_{=0} + \partial_{\boldsymbol{\theta}} \boldsymbol{\lambda}_*^\top \cdot \underbrace{\partial_{\boldsymbol{\lambda}} \mathcal{L}(\boldsymbol{\Phi}_*, \boldsymbol{\lambda}_*, \boldsymbol{\theta}, \boldsymbol{u})}_{=0}$$

$$= \sum_{k=0}^{K-1} \nabla_2 \ell[\boldsymbol{\Phi}_{-k}, \boldsymbol{\theta}, -k] + \partial_2 \mathcal{M}_{H,\delta}[\boldsymbol{\Phi}_{-(k+1)}, \boldsymbol{\theta}, \boldsymbol{u}_{-(k+1)}]^\top \cdot \boldsymbol{\lambda}_k$$

The total derivative of $L$ with respect to $\boldsymbol{u}_{-k}$ is derived in the exact same fashion. $\qquad\square$

**Remark 5.** *Note that using the vectorized notations introduced in subsection 3.3, the Lagrangian of the constrained optimization problem defined in Def. A.4 re-writes:*

$$\mathcal{L} = \mathbf{1}^\top \cdot \ell[\widetilde{\boldsymbol{\Phi}}, \boldsymbol{\theta}] + \mathrm{Tr}\left[ \left( \mathcal{M}_{H,\delta}[\widetilde{\boldsymbol{\Phi}}, \boldsymbol{\theta}, \widetilde{\boldsymbol{u}}] - \widetilde{\boldsymbol{\Phi}} \right) \cdot \overline{\boldsymbol{\lambda}}^\top \right]$$

*with $\mathrm{Tr}$ denoting the trace matrix operator and:*

$$\mathbf{1} := \begin{bmatrix} 1 \\ \vdots \\ 1 \end{bmatrix} \in \mathbb{R}^{K \times 1}, \quad \ell[\widetilde{\boldsymbol{\Phi}}, \boldsymbol{\theta}] := \begin{bmatrix} \ell[\boldsymbol{\Phi}_0, \boldsymbol{\theta}, 0] \\ \vdots \\ \ell[\boldsymbol{\Phi}_{-(K-1)}, \boldsymbol{\theta}, -(K-1)] \end{bmatrix} \in \mathbb{R}^{K \times 1},$$

$$\mathcal{M}_{H,\delta}[\widetilde{\boldsymbol{\Phi}}, \boldsymbol{\theta}, \widetilde{\boldsymbol{u}}] := \begin{bmatrix} \mathcal{M}_{H,\delta}[\boldsymbol{\Phi}_0, \boldsymbol{\theta}, \boldsymbol{u}] \\ \vdots \\ \mathcal{M}_{H,\delta}[\boldsymbol{\Phi}_{-(K-1)}, \boldsymbol{\theta}, \boldsymbol{u}] \end{bmatrix} \in \mathbb{R}^{K \times d_{\boldsymbol{\Phi}}}$$

**Detailed BPTT.** For the needs of our derivation of RHEL in discrete time, we now introduce a finer-grained version of BPTT given model assumptions given in Def. A.3.

**Corollary A.5** (Detailed BPTT). *Given assumptions in Def. A.3–A.4, the gradients of the loss with respect to the parameters $\boldsymbol{\theta}$ and the inputs $\boldsymbol{u}_{-k}$ are given by:*

$$d_{\boldsymbol{\theta}}L = \sum_{k=0}^{K-1} g_{\boldsymbol{\theta}}(k), \quad d_{\boldsymbol{u}_{-k}}L = g_{\boldsymbol{u}}(k),$$

*with:*

$$g_{\boldsymbol{\theta}}(k) := \nabla_2 \ell[\boldsymbol{\Phi}_{-k}, \boldsymbol{\theta}, -k] + \frac{\delta}{2}\nabla_{1,2}^2 T[\boldsymbol{\Phi}_{-(k+1/3)}, \boldsymbol{\theta}, \boldsymbol{u}_{-(k+1)}] \cdot \boldsymbol{J}^\top \cdot \boldsymbol{\lambda}_k$$

$$+ \delta\nabla_{1,2}^2 V[\boldsymbol{\Phi}_{-(k+2/3)}, \boldsymbol{\theta}, \boldsymbol{u}_{-(k+1)}] \cdot \boldsymbol{J}^\top \cdot \boldsymbol{\lambda}_{k+1/3} + \frac{\delta}{2}\nabla_{1,2}^2 T[\boldsymbol{\Phi}_{-(k+1)}, \boldsymbol{\theta}, \boldsymbol{u}_{-(k+1)}] \cdot \boldsymbol{J}^\top \cdot \boldsymbol{\lambda}_{k+2/3},$$

$$g_{\boldsymbol{u}}(k) := \frac{\delta}{2}\nabla_{1,3}^2 T[\boldsymbol{\Phi}_{-(k+1/3)}, \boldsymbol{\theta}, \boldsymbol{u}_{-(k+1)}] \cdot \boldsymbol{J}^\top \cdot \boldsymbol{\lambda}_k$$

$$+ \delta\nabla_{1,3}^2 V[\boldsymbol{\Phi}_{-(k+2/3)}, \boldsymbol{\theta}, \boldsymbol{u}_{-(k+1)}] \cdot \boldsymbol{J}^\top \cdot \boldsymbol{\lambda}_{k+1/3} + \frac{\delta}{2}\nabla_{1,3}^2 T[\boldsymbol{\Phi}_{-(k+1)}, \boldsymbol{\theta}, \boldsymbol{u}_{-(k+1)}] \cdot \boldsymbol{J}^\top \cdot \boldsymbol{\lambda}_{k+2/3},$$

*and where $(\boldsymbol{\lambda}_k)$ satisfy the following recursion relationship, with $\boldsymbol{\lambda}_0 = \nabla_{\boldsymbol{\Phi}}\ell[\boldsymbol{\Phi}_0]$ and $\forall k \in [0, K-1]$:*

$$\begin{cases} \boldsymbol{\lambda}_{k+1/3} &= \boldsymbol{\lambda}_k + \frac{\delta}{2}\nabla_1^2 T[\boldsymbol{\Phi}_{-(k+1/3)}, \boldsymbol{\theta}, \boldsymbol{u}_{-(k+1)}] \cdot \boldsymbol{J}^\top \cdot \boldsymbol{\lambda}_k \\ \boldsymbol{\lambda}_{k+2/3} &= \boldsymbol{\lambda}_{k+1/3} + \delta\nabla_1^2 V[\boldsymbol{\Phi}_{-(k+2/3)}, \boldsymbol{\theta}, \boldsymbol{u}_{-(k+1)}] \cdot \boldsymbol{J}^\top \cdot \boldsymbol{\lambda}_{k+1/3} \\ \boldsymbol{\lambda}_{k+1} &= \boldsymbol{\lambda}_{k+2/3} + \frac{\delta}{2}\nabla_1^2 T[\boldsymbol{\Phi}_{-(k+1)}, \boldsymbol{\theta}, \boldsymbol{u}_{-(k+1)}] \cdot \boldsymbol{J}^\top \cdot \boldsymbol{\lambda}_{k+2/3} + \nabla_1 \ell[\boldsymbol{\Phi}_{-(k+1)}, \boldsymbol{\theta}] \end{cases}$$

*Proof of Corollary A.5.* Direct application of Theorem A.3 with the inference computational graph details defined inside Def. A.3. □

### A.2.3 Proof of Theorem 3.2

**Time reversibility in discrete time.** We first derive the discrete counterpart of Lemma A.3 as a technical pre-requisite for the extension of RHEL to the discrete-time setting.

**Lemma A.4** (Reversibility of $\mathcal{M}_{H,\delta}$). *Under the assumptions of Def. A.3–A.4:*

$$\forall \boldsymbol{\Phi}_k \in \mathbb{R}^{d_\Phi}, \; \forall \boldsymbol{\theta} \in \mathbb{R}^{d_\theta}, \; \forall \boldsymbol{u} \in \mathbb{R}^{d_u} : \quad \boldsymbol{\Phi}_{k+1} = \mathcal{M}_{H,\delta}[\boldsymbol{\Phi}_k, \boldsymbol{\theta}, \boldsymbol{u}] \Rightarrow \mathcal{M}_{H,\delta}[\boldsymbol{\Phi}_{k+1}^\star, \boldsymbol{\theta}, \boldsymbol{u}] = \boldsymbol{\Phi}_k^\star$$

*Proof of Lemma A.4.* Let $\boldsymbol{\Phi}_k, \boldsymbol{\Phi}_{k+1} \in \mathbb{R}^{d_\Phi}$ be such that:

$$\boldsymbol{\Phi}_{k+1} = \mathcal{M}_{H,\delta}[\boldsymbol{\Phi}_k, \boldsymbol{\theta}, \boldsymbol{u}]$$

which rewrites, given Def. A.3:

$$\mathcal{M}_{H,\delta} : \begin{cases} \phi_{k+1/3} & = \phi_k + \frac{\delta}{2}\nabla_{\boldsymbol{\pi}}T[\boldsymbol{\pi}_k, \boldsymbol{\theta}, \boldsymbol{u}] \\ \boldsymbol{\pi}_{k+1/3} & = \boldsymbol{\pi}_k \\ \phi_{k+2/3} & = \phi_{k+1/3} \\ \boldsymbol{\pi}_{k+2/3} & = \boldsymbol{\pi}_{k+1/3} - \delta\nabla_{\boldsymbol{\phi}}V[\phi_{k+1/3}, \boldsymbol{\theta}, \boldsymbol{u}] \\ \phi_{k+1} & = \phi_{k+2/3} + \frac{\delta}{2}\nabla_{\boldsymbol{\pi}}T[\boldsymbol{\pi}_{k+2/3}, \boldsymbol{\theta}, \boldsymbol{u}] \\ \boldsymbol{\pi}_{k+1} & = \boldsymbol{\pi}_{k+2/3} \end{cases} \tag{20}$$

It becomes apparent from Eq. (20) that $\mathcal{M}_{H,\delta}$ is invertible with respect to its first argument and that inverting $\mathcal{M}_{H,\delta}$ amounts to change $\delta$ to $-\delta$:

$$\mathcal{M}_{H,\delta}^{-1} : \begin{cases} \phi_{k+2/3} & = \phi_{k+1} - \frac{\delta}{2}\nabla_{\boldsymbol{\pi}}T[\boldsymbol{\pi}_{k+1}, \boldsymbol{\theta}, \boldsymbol{u}] \\ \boldsymbol{\pi}_{k+2/3} & = \boldsymbol{\pi}_{k+1} \\ \phi_{k+1/3} & = \phi_{k+2/3} \\ \boldsymbol{\pi}_{k+1/3} & = \boldsymbol{\pi}_{k+2/3} + \delta\nabla_{\boldsymbol{\phi}}V[\phi_{k+2/3}, \boldsymbol{\theta}, \boldsymbol{u}] \\ \phi_k & = \phi_{k+1/3} - \frac{\delta}{2}\nabla_{\boldsymbol{\pi}}T[\boldsymbol{\pi}_{k+1/3}, \boldsymbol{\theta}, \boldsymbol{u}] \\ \boldsymbol{\pi}_k & = \boldsymbol{\pi}_{k+1/3} \end{cases} \tag{21}$$

and therefore:

$$\boldsymbol{\Phi}_k = \mathcal{M}_{H,\delta}^{-1}[\boldsymbol{\Phi}_{k+1}, \boldsymbol{\theta}, \boldsymbol{u}] = \mathcal{M}_{H,-\delta}[\boldsymbol{\Phi}_{k+1}, \boldsymbol{\theta}, \boldsymbol{u}],$$

Denoting $\boldsymbol{\pi}^\star := -\boldsymbol{\pi}$, note that by time-reversal invariance hypothesis in Def. A.3 and Lemma A.2, we have that $\nabla_{\boldsymbol{\pi}}T[\boldsymbol{\pi}, \boldsymbol{\theta}, \boldsymbol{u}] = -\nabla_{\boldsymbol{\pi}^\star}T[\boldsymbol{\pi}^\star, \boldsymbol{\theta}, \boldsymbol{u}]$. Therefore, Eq. (21) rewrites as:

$$\begin{cases} \phi_{k+2/3} & = \phi_{k+1} + \frac{\delta}{2}\nabla_{\boldsymbol{\pi}^\star}T[\boldsymbol{\pi}_{k+1}^\star, \boldsymbol{\theta}, \boldsymbol{u}] \\ \boldsymbol{\pi}_{k+2/3}^\star & = \boldsymbol{\pi}_{k+1}^\star \\ \phi_{k+1/3} & = \phi_{k+2/3} \\ \boldsymbol{\pi}_{k+1/3}^\star & = \boldsymbol{\pi}_{k+2/3}^\star - \delta\nabla_{\boldsymbol{\phi}}V[\phi_{k+2/3}, \boldsymbol{\theta}, \boldsymbol{u}] \\ \phi_k & = \phi_{k+1/3} + \frac{\delta}{2}\nabla_{\boldsymbol{\pi}^\star}T[\boldsymbol{\pi}_{k+1/3}^\star, \boldsymbol{\theta}, \boldsymbol{u}] \\ \boldsymbol{\pi}_k^\star & = \boldsymbol{\pi}_{k+1/3}^\star \end{cases}, \tag{22}$$

where equations bearing on $\boldsymbol{\pi}$ have been multiplied on both sides by $-1$. Finally note that Eq. (22) simply rewrites as:

$$\mathcal{M}_{H,\delta}[\boldsymbol{\Phi}_{k+1}^\star, \boldsymbol{\theta}, \boldsymbol{u}] = \boldsymbol{\Phi}_k^\star$$

$\square$

**Corollary A.6.** *Under the assumptions of Def. A.3, if $\boldsymbol{\Phi}$ satisfies the following recursive equations:*

$$\boldsymbol{\Phi}_{-K} = \boldsymbol{x}, \quad \forall k = -K, \cdots, -1 : \quad \boldsymbol{\Phi}_{k+1} = \mathcal{M}_{H,\delta}[\boldsymbol{\Phi}_k, \boldsymbol{\theta}, \boldsymbol{u}_k]$$

*and $\boldsymbol{\Phi}^e$ is subsequently defined as:*

$$\boldsymbol{\Phi}_0^e = \boldsymbol{\Phi}_0^\star, \quad \forall t = 0, \cdots, K-1 : \quad \boldsymbol{\Phi}_{k+1}^e = \mathcal{M}_{H,\delta}[\boldsymbol{\Phi}_k^e, \boldsymbol{\theta}, \boldsymbol{u}_{-(k+1)}]$$

*Then:*

$$\forall t = 0, \cdots, K-1 : \quad \boldsymbol{\Phi}_k^e = \boldsymbol{\Phi}_{-k}^\star$$

*Proof of Corollary A.6.* This result is immediately obtained by iterating Lemma A.4 over the whole trajectory. $\square$

**A technical pre-requisite.** Finally, we need one last technical Lemma to handle subtleties pertaining to Jacobian evaluation which only occur in discrete time.

**Lemma A.5.** *Under the assumptions of Def. A.3, if we have, for some $\boldsymbol{\theta} \in \mathbb{R}^{d_\theta}$ and $\boldsymbol{u} \in \mathbb{R}^{d_u}$:*

$$\boldsymbol{\Phi}_{k+1} = \mathcal{M}_{H,\delta}[\boldsymbol{\Phi}_k, \boldsymbol{\theta}, \boldsymbol{u}],$$

*then:*

$$\begin{cases} T[\boldsymbol{\Phi}_{k+1/3}, \boldsymbol{\theta}, \boldsymbol{u}] = T[\boldsymbol{\Phi}_k, \boldsymbol{\theta}, \boldsymbol{u}], \\ V[\boldsymbol{\Phi}_{k+2/3}, \boldsymbol{\theta}, \boldsymbol{u}] = V[\boldsymbol{\Phi}_{k+1/3}, \boldsymbol{\theta}, \boldsymbol{u}] \\ T[\boldsymbol{\Phi}_{k+1}, \boldsymbol{\theta}, \boldsymbol{u}] = T[\boldsymbol{\Phi}_{k+2/3}, \boldsymbol{\theta}, \boldsymbol{u}] \end{cases}$$

*Proof.* This can be seen by simply writing $\mathcal{M}_{H,\delta}$ explicitly for $\phi$ and $\boldsymbol{\pi}$:

$$\mathcal{M}_{H,\delta} : \begin{cases} \phi_{k+1/3} &= \phi_k + \frac{\delta}{2}\nabla_{\boldsymbol{\pi}} T[\boldsymbol{\pi}_k, \boldsymbol{\theta}, \boldsymbol{u}] \\ \boldsymbol{\pi}_{k+1/3} &= \boldsymbol{\pi}_k \\ \phi_{k+2/3} &= \phi_{k+1/3} \\ \boldsymbol{\pi}_{k+2/3} &= \boldsymbol{\pi}_{k+1/3} - \delta\nabla_\phi V[\phi_{k+1/3}, \boldsymbol{\theta}, \boldsymbol{u}] \\ \phi_{k+1} &= \phi_{k+2/3} + \frac{\delta}{2}\nabla_{\boldsymbol{\pi}} T[\boldsymbol{\pi}_{k+2/3}, \boldsymbol{\theta}, \boldsymbol{u}] \\ \boldsymbol{\pi}_{k+1} &= \boldsymbol{\pi}_{k+2/3} \end{cases}$$

$\square$

**Discrete-time RHEL.** We are now ready to state the main result of this section.

**Corollary A.7.** *Under the assumptions of Def. A.3–A.4, let $(\boldsymbol{\Phi}_k)_k$ satisfy the recursive equation:*

$$\boldsymbol{\Phi}_{-K} = \boldsymbol{x}, \quad \boldsymbol{\Phi}_{k+1} = \mathcal{M}_{H,\delta}[\boldsymbol{\Phi}_k, \boldsymbol{\theta}, \boldsymbol{u}_k] \quad \forall k = -K, \cdots, -1,$$

*and let $\boldsymbol{\Phi}^e$ satisfy:*

$$\begin{cases} \boldsymbol{\Phi}_0^e(\epsilon) = \boldsymbol{\Phi}_0^\star + \epsilon\boldsymbol{\Sigma}_x \cdot \nabla_{\boldsymbol{\Phi}}\ell[\boldsymbol{\Phi}_0, \boldsymbol{\theta}, 0], \\ \forall k = 0, \cdots, K-1: \\ \quad \boldsymbol{\Phi}_{k+1/3}^e(\epsilon) = \mathcal{M}_{T,\delta/2}[\boldsymbol{\Phi}_k^e(\epsilon), \boldsymbol{\theta}, \boldsymbol{u}_{-(k+1)}] \\ \quad \boldsymbol{\Phi}_{k+2/3}^e(\epsilon) = \mathcal{M}_{V,\delta}[\boldsymbol{\Phi}_{k+1/3}^e(\epsilon), \boldsymbol{\theta}, \boldsymbol{u}_{-(k+1)}] \\ \quad \boldsymbol{\Phi}_{k+1}^e(\epsilon) = \mathcal{M}_{T,\delta/2}[\boldsymbol{\Phi}_{k+2/3}^e(\epsilon), \boldsymbol{\theta}, \boldsymbol{u}_{-(k+1)}] - \epsilon\boldsymbol{J} \cdot \nabla_{\boldsymbol{\Phi}^e}\ell[\boldsymbol{\Phi}_{k+1}^e, \boldsymbol{\theta}, -(k+1)], \end{cases}$$

*Then defining:*

$$H^{1/2}[\boldsymbol{\Phi}_k^e(\epsilon), \boldsymbol{\theta}, \boldsymbol{u}_{-(k+1)}] := \frac{1}{2}\left(H[\boldsymbol{\Phi}_{k+1/3}^e(\epsilon), \boldsymbol{\theta}, \boldsymbol{u}_{-(k+1)}] + H[\boldsymbol{\Phi}_{k+2/3}^e(\epsilon), \boldsymbol{\theta}, \boldsymbol{u}_{-(k+1)}]\right),$$

$$\Delta_{\boldsymbol{\theta}}^{\text{RHEL}}(k, \epsilon) := \nabla_2\ell[\boldsymbol{\Phi}_k^e(\epsilon), \boldsymbol{\theta}, -k]$$
$$- \frac{\delta}{2\epsilon}\left(\nabla_2 H^{1/2}[\boldsymbol{\Phi}_k^e(\epsilon), \boldsymbol{\theta}, \boldsymbol{u}_{-(k+1)}] - \nabla_2 H^{1/2}[\boldsymbol{\Phi}_k^e(-\epsilon), \boldsymbol{\theta}, \boldsymbol{u}_{-(k+1)}]\right),$$

$$\Delta_{\boldsymbol{u}}^{\text{RHEL}}(k, \epsilon) := -\frac{\delta}{2\epsilon}\left(\nabla_3 H^{1/2}[\boldsymbol{\Phi}_k^e(\epsilon), \boldsymbol{\theta}, \boldsymbol{u}_{-(k+1)}] - \nabla_3 H^{1/2}[\boldsymbol{\Phi}_k^e(-\epsilon), \boldsymbol{\theta}, \boldsymbol{u}_{-(k+1)}]\right),$$

$$\Delta_{\boldsymbol{\Phi}}^{\text{RHEL}}(k, \epsilon) := \frac{1}{2\epsilon}\boldsymbol{\Sigma}_x \cdot \left(\boldsymbol{\Phi}_k^e(\epsilon) - \boldsymbol{\Phi}_k^e(-\epsilon)\right),$$

*we have:*

$$\forall k = 0, \cdots, K-1: \quad \boldsymbol{\lambda}_k = \lim_{\epsilon \to 0}\Delta_{\boldsymbol{\theta}}^{\text{RHEL}}(k, \epsilon), \quad g_{\boldsymbol{\theta}}(k) = \lim_{\epsilon \to 0}\Delta_{\boldsymbol{\theta}}^{\text{RHEL}}(k, \epsilon),$$

$$g_{\boldsymbol{u}}(k) = \lim_{\epsilon \to 0}\Delta_{\boldsymbol{u}}^{\text{RHEL}}(k, \epsilon),$$

*where $(\boldsymbol{\lambda}_k)_{k \in [\![0,K]\!]}$, $(g_{\boldsymbol{\theta}}(k))_{k \in [\![0,K-1]\!]}$ and $(g_{\boldsymbol{u}}(k))_{k \in [\![0,K-1]\!]}$ are defined inside Corollary (A.5).*

*Proof of Corollary A.7.* Let $k \in [0, K-1]$. We define:

$$\Delta_{\boldsymbol{\Phi}}^{\mathrm{RHEL}}(k) := \lim_{\epsilon \to 0} \Delta_{\boldsymbol{\Phi}}^{\mathrm{RHEL}}(k, \epsilon) = \boldsymbol{\Sigma}_x \cdot \partial_\epsilon \boldsymbol{\Phi}(k, \epsilon)|_{\epsilon=0}$$

$$\Delta_{\boldsymbol{\theta}}^{\mathrm{RHEL}}(k) := \lim_{\epsilon \to 0} \Delta_{\boldsymbol{\theta}}^{\mathrm{RHEL}}(k, \epsilon) = \nabla_2 \ell[\boldsymbol{\Phi}_k^e(0), \boldsymbol{\theta}, -k] - \delta \partial_\epsilon \left( \nabla_2 H^{1/2}[\boldsymbol{\Phi}_k^e(\epsilon), \boldsymbol{\theta}, \boldsymbol{u}_{-(k+1)}] \right)|_{\epsilon=0}$$

$$\Delta_{\boldsymbol{u}}^{\mathrm{RHEL}}(k) := \lim_{\epsilon \to 0} \Delta_{\boldsymbol{u}}^{\mathrm{RHEL}}(k, \epsilon) = -\delta \partial_\epsilon \left( \nabla_3 H^{1/2}[\boldsymbol{\Phi}_k^e(\epsilon), \boldsymbol{\theta}, \boldsymbol{u}_{-(k+1)}] \right)|_{\epsilon=0}$$

**Derivation of $\boldsymbol{\lambda}_k = \lim_{\epsilon \to 0} \Delta_{\boldsymbol{\theta}}^{\mathrm{RHEL}}(k, \epsilon)$.** We proceed exactly as in Theorem A.2 with some subtle adaptations which we highlight. Differentiating the dynamics of $\boldsymbol{\Phi}^e$ between $k$ and $k + 1/3$ and proceeding as in the proof of Theorem A.2 using Lemma A.1, Lemma A.2 and Corollary A.6 (as the discrete counterpart of Lemma A.3 which was used for Theorem A.2), we obtain:

$$\Delta_{\boldsymbol{\Phi}}^{\mathrm{RHEL}}(k + 1/3) = \Delta_{\boldsymbol{\Phi}}^{\mathrm{RHEL}}(k) + \frac{\delta}{2} \nabla_{\boldsymbol{\Phi}}^2 T[\boldsymbol{\Phi}_{-k}, \boldsymbol{\theta}, \boldsymbol{u}_{-(k+1)}] \cdot \boldsymbol{J}^\top \cdot \Delta_{\boldsymbol{\Phi}}^{\mathrm{RHEL}}(k)$$

However note that this does not correctly match the dynamics satisfied by $\boldsymbol{\lambda}$ inside Corollary A.5 between $k$ and $k + 1/3$: the Hessian $\nabla_{\boldsymbol{\Phi}}^2 T$ should instead be evaluated at $\boldsymbol{\Phi}_{-(k+1/3)}$. Fortunately, using Lemma A.5:

$$\nabla_{\boldsymbol{\Phi}}^2 T[\boldsymbol{\Phi}_{-k}, \boldsymbol{\theta}, \boldsymbol{u}_{-(k+1)}] = \nabla_{\boldsymbol{\Phi}}^2 T[\boldsymbol{\Phi}_{-(k+1/3)}, \boldsymbol{\theta}, \boldsymbol{u}_{-(k+1)}]$$

Therefore we get:

$$\Delta_{\boldsymbol{\Phi}}^{\mathrm{RHEL}}(k + 1/3) = \Delta_{\boldsymbol{\Phi}}^{\mathrm{RHEL}}(k) + \frac{\delta}{2} \nabla_{\boldsymbol{\Phi}}^2 T[\boldsymbol{\Phi}_{-(k+1/3)}, \boldsymbol{\theta}, \boldsymbol{u}_{-(k+1)}] \cdot \boldsymbol{J}^\top \cdot \Delta_{\boldsymbol{\Phi}}^{\mathrm{RHEL}}(k)$$

Proceeding the same way on $\boldsymbol{\Phi}_{k+2/3}^e$ and $\boldsymbol{\Phi}_{k+1}^e$, we get altogether:

$$\begin{cases} \Delta_{\boldsymbol{\Phi}}^{\mathrm{RHEL}}(k + 1/3) = \Delta_{\boldsymbol{\Phi}}^{\mathrm{RHEL}}(k) + \frac{\delta}{2} \nabla_{\boldsymbol{\Phi}}^2 T[\boldsymbol{\Phi}_{-(k+1/3)}, \boldsymbol{\theta}, \boldsymbol{u}_{-(k+1)}] \cdot \boldsymbol{J}^\top \cdot \Delta_{\boldsymbol{\Phi}}^{\mathrm{RHEL}}(k) \\ \Delta_{\boldsymbol{\Phi}}^{\mathrm{RHEL}}(k + 2/3) = \Delta_{\boldsymbol{\Phi}}^{\mathrm{RHEL}}(k + 1/3) + \delta \nabla_{\boldsymbol{\Phi}}^2 V[\boldsymbol{\Phi}_{-(k+2/3)}, \boldsymbol{\theta}, \boldsymbol{u}_{-(k+1)}] \cdot \boldsymbol{J}^\top \cdot \Delta_{\boldsymbol{\Phi}}^{\mathrm{RHEL}}(k + 1/3) \\ \Delta_{\boldsymbol{\Phi}}^{\mathrm{RHEL}}(k + 1) = \Delta_{\boldsymbol{\Phi}}^{\mathrm{RHEL}}(k + 2/3) + \frac{\delta}{2} \nabla_{\boldsymbol{\Phi}}^2 T[\boldsymbol{\Phi}_{-(k+1)}, \boldsymbol{\theta}, \boldsymbol{u}_{-(k+1)}] \cdot \boldsymbol{J}^\top \cdot \Delta_{\boldsymbol{\Phi}}^{\mathrm{RHEL}}(k + 2/3) \\ \qquad + \nabla_{\boldsymbol{\Phi}} \ell[\boldsymbol{\Phi}_{-(k+1)}, \boldsymbol{\theta}, -(k+1)] \end{cases}$$

$\Delta_{\boldsymbol{\Phi}}^{\mathrm{RHEL}}$ satisfying the same equations as $(\boldsymbol{\lambda}_k)_k$ given by BPTT, together with same initial conditions:

$$\Delta_{\boldsymbol{\Phi}}^{\mathrm{RHEL}}(0) = \nabla_{\boldsymbol{\Phi}} \ell[\boldsymbol{\Phi}_0, \boldsymbol{\theta}, 0]$$

yields the desired equality.

**Derivation of $g_{\boldsymbol{\theta}}(k) = \lim_{\epsilon \to 0} \Delta_{\boldsymbol{\theta}}^{\mathrm{RHEL}}(k, \epsilon)$.** Proceeding in the same way as in the derivation of Theorem A.2, starting from the expression of $g_{\boldsymbol{\theta}}(k)$ derived in Corollary A.5, using $\Delta_{\boldsymbol{\Phi}}^{\mathrm{RHEL}}(k) = \boldsymbol{\lambda}_k$ and paying attention to evaluating Jacobian at the right places using Lemma A.5, we obtain $\forall k = 0, \cdots, K - 1$:

$$g_{\boldsymbol{\theta}}(k) = \nabla_2 \ell[\boldsymbol{\Phi}_k^e(0), \boldsymbol{\theta}, -k]$$
$$- \frac{\delta}{2} \partial_\epsilon \left\{ \nabla_{\boldsymbol{\theta}} T[\boldsymbol{\Phi}_k^e(\epsilon), \boldsymbol{\theta}, \boldsymbol{u}_{-(k+1)}] + 2\nabla_{\boldsymbol{\theta}} V[\boldsymbol{\Phi}_{k+1/3}^e(\epsilon), \boldsymbol{\theta}, \boldsymbol{u}_{-(k+1)}] + \nabla_{\boldsymbol{\theta}} T[\boldsymbol{\Phi}_{k+2/3}^e(\epsilon), \boldsymbol{\theta}, \boldsymbol{u}_{-(k+1)}] \right\} \Big|_{\epsilon=0}.$$

There again, using Lemma A.5:

$$\nabla_{\boldsymbol{\theta}} T[\boldsymbol{\Phi}_k^e(\epsilon), \boldsymbol{\theta}, \boldsymbol{u}_{-(k+1)}] = \nabla_{\boldsymbol{\theta}} T[\boldsymbol{\Phi}_{k+1/3}^e(\epsilon), \boldsymbol{\theta}, \boldsymbol{u}_{-(k+1)}], \quad \nabla_{\boldsymbol{\theta}} V[\boldsymbol{\Phi}_{k+1/3}^e(\epsilon), \boldsymbol{\theta}, \boldsymbol{u}_{-(k+1)}] = \nabla_{\boldsymbol{\theta}} V[\boldsymbol{\Phi}_{k+2/3}^e(\epsilon), \boldsymbol{\theta}, \boldsymbol{u}_{-(k+1)}],$$

therefore:

$$g_{\boldsymbol{\theta}}(k) = \nabla_2 \ell[\boldsymbol{\Phi}_k^e(0), \boldsymbol{\theta}, -k] - \frac{\delta}{2} \partial_\epsilon \left\{ \nabla_{\boldsymbol{\theta}} T[\boldsymbol{\Phi}_{k+1/3}^e(\epsilon), \boldsymbol{\theta}, \boldsymbol{u}_{-(k+1)}] + \nabla_{\boldsymbol{\theta}} V[\boldsymbol{\Phi}_{k+1/3}^e(\epsilon), \boldsymbol{\theta}, \boldsymbol{u}_{-(k+1)}] \right.$$
$$\left. + \nabla_{\boldsymbol{\theta}} V[\boldsymbol{\Phi}_{k+2/3}^e(\epsilon), \boldsymbol{\theta}, \boldsymbol{u}_{-(k+1)}] + \nabla_{\boldsymbol{\theta}} T[\boldsymbol{\Phi}_{k+2/3}^e(\epsilon), \boldsymbol{\theta}, \boldsymbol{u}_{-(k+1)}] \right\} \Big|_{\epsilon=0}.$$
$$= \nabla_2 \ell[\boldsymbol{\Phi}_k^e(0), \boldsymbol{\theta}, -k] - \frac{\delta}{2} \partial_\epsilon \left( H[\boldsymbol{\Phi}_{k+1/3}^e(\epsilon), \boldsymbol{\theta}, \boldsymbol{u}_{-(k+1)}] + H[\boldsymbol{\Phi}_{k+2/3}^e(\epsilon), \boldsymbol{\theta}, \boldsymbol{u}_{-(k+1)}] \right) \Big|_{\epsilon=0}$$
$$= \nabla_2 \ell[\boldsymbol{\Phi}_k^e(0), \boldsymbol{\theta}, -k] - \delta \partial_\epsilon \left( H^{1/2}[\boldsymbol{\Phi}_k^e(\epsilon), \boldsymbol{\theta}, \boldsymbol{u}_{-(k+1)}] \right) \Big|_{\epsilon=0} = \lim_{\epsilon \to 0} \Delta_{\boldsymbol{\theta}}^{\mathrm{RHEL}}(k, \epsilon)$$

**Derivation of** $g_{\boldsymbol{u}}(k) = \lim_{\epsilon \to 0} \Delta_{\boldsymbol{u}}^{\mathrm{RHEL}}(k, \epsilon)$. Strictly identical to the above paragraph. □

**Remark 6.** *Note that the echo dynamics prescribed by Corollary A.7 are* ***implicit****:*

$$\begin{cases} \boldsymbol{\Phi}_0^e(\epsilon) = \boldsymbol{\Phi}_0^\star + \epsilon\boldsymbol{\Sigma}_x \cdot \nabla_{\boldsymbol{\Phi}}\ell[\boldsymbol{\Phi}_0, \boldsymbol{\theta}, 0], \\ \forall k = 0, \cdots, K-1: \\ \quad \boldsymbol{\Phi}_{k+1/3}^e(\epsilon) = \mathcal{M}_{T,\delta/2}[\boldsymbol{\Phi}_k^e(\epsilon), \boldsymbol{\theta}, \boldsymbol{u}_{-(k+1)}] \\ \quad \boldsymbol{\Phi}_{k+2/3}^e(\epsilon) = \mathcal{M}_{V,\delta}[\boldsymbol{\Phi}_{k+1/3}^e(\epsilon), \boldsymbol{\theta}, \boldsymbol{u}_{-(k+1)}] \\ \quad \textcolor{red}{\boldsymbol{\Phi}_{k+1}^e}(\epsilon) = \mathcal{M}_{T,\delta/2}[\boldsymbol{\Phi}_{k+2/3}^e(\epsilon), \boldsymbol{\theta}, \boldsymbol{u}_{-(k+1)}] - \epsilon\boldsymbol{J} \cdot \nabla_{\boldsymbol{\Phi}^e}\ell[\textcolor{red}{\boldsymbol{\Phi}_{k+1}^e}, \boldsymbol{\theta}, -(k+1)], \end{cases}$$

*where* $\boldsymbol{\Phi}_{k+1}^e$ *appears, as highlighted in red, on both sides of the last step. A straightforward way to make the echo dynamics explicit while preserving the theoretical guarantees of Corollary A.7 is to* linearize *the nudging signal, namely using instead the following set of equations:*

$$\begin{cases} \boldsymbol{\Phi}_0^e(\epsilon) = \boldsymbol{\Phi}_0^\star + \epsilon\boldsymbol{\Sigma}_x \cdot \nabla_{\boldsymbol{\Phi}}\ell[\boldsymbol{\Phi}_0, \boldsymbol{\theta}, 0], \\ \forall k = 0, \cdots, K-1: \\ \quad \boldsymbol{\Phi}_{k+1/3}^e(\epsilon) = \mathcal{M}_{T,\delta/2}[\boldsymbol{\Phi}_k^e(\epsilon), \boldsymbol{\theta}, \boldsymbol{u}_{-(k+1)}] \\ \quad \boldsymbol{\Phi}_{k+2/3}^e(\epsilon) = \mathcal{M}_{V,\delta}[\boldsymbol{\Phi}_{k+1/3}^e(\epsilon), \boldsymbol{\theta}, \boldsymbol{u}_{-(k+1)}] \\ \quad \boldsymbol{\Phi}_{k+1}^e(\epsilon) = \mathcal{M}_{T,\delta/2}[\boldsymbol{\Phi}_{k+2/3}^e(\epsilon), \boldsymbol{\theta}, \boldsymbol{u}_{-(k+1)}] + \epsilon\boldsymbol{\Sigma}_x \cdot \nabla_{\boldsymbol{\Phi}^e}\ell[\boldsymbol{\Phi}_{-(k+1)}, \boldsymbol{\theta}, -(k+1)], \end{cases}$$

*Note that the* $\epsilon\boldsymbol{J}$ *of the original implicit equation becomes* $-\epsilon\boldsymbol{\Sigma}_x$ *in its linearized counterpart.*

This remark leads us to a slight variant of Corollary A.7.

**Corollary A.8.** *Under the assumptions of Def. A.3–A.4, let* $(\boldsymbol{\Phi}_k)_k$ *satisfy the recursive equation:*

$$\boldsymbol{\Phi}_{-K} = \boldsymbol{x}, \quad \boldsymbol{\Phi}_{k+1} = \mathcal{M}_{H,\delta}[\boldsymbol{\Phi}_k, \boldsymbol{\theta}, \boldsymbol{u}_k] \quad \forall k = -K, \cdots, -1,$$

*and let* $\boldsymbol{\Phi}^e$ *satisfy:*

$$\begin{cases} \boldsymbol{\Phi}_0^e(\epsilon) = \boldsymbol{\Phi}_0^\star + \epsilon\boldsymbol{\Sigma}_x \cdot \boldsymbol{y}_0, \\ \forall k = 0, \cdots, K-1: \\ \quad \boldsymbol{\Phi}_{k+1/3}^e(\epsilon) = \mathcal{M}_{T,\delta/2}[\boldsymbol{\Phi}_k^e(\epsilon), \boldsymbol{\theta}, \boldsymbol{u}_{-(k+1)}] \\ \quad \boldsymbol{\Phi}_{k+2/3}^e(\epsilon) = \mathcal{M}_{V,\delta}[\boldsymbol{\Phi}_{k+1/3}^e(\epsilon), \boldsymbol{\theta}, \boldsymbol{u}_{-(k+1)}] \\ \quad \boldsymbol{\Phi}_{k+1}^e(\epsilon) = \mathcal{M}_{T,\delta/2}[\boldsymbol{\Phi}_{k+2/3}^e(\epsilon), \boldsymbol{\theta}, \boldsymbol{u}_{-(k+1)}] + \epsilon\boldsymbol{\Sigma}_x \cdot \boldsymbol{y}_{-(k+1)}, \end{cases}$$

*where* $\boldsymbol{y} \in \mathbb{R}^{K \times d_{\boldsymbol{\Phi}}}$ *does not depend on* $\boldsymbol{\Phi}$*. Then the same conclusions as Corollary A.7 hold, with* $(\boldsymbol{\lambda}_k)_{k \in [\![0, K-1]\!]}$ *satisfying* $\boldsymbol{\lambda}_0 = \boldsymbol{y}_0$ *and* $\forall k \in [0, K-1]$:

$$\begin{cases} \boldsymbol{\lambda}_{k+1/3} = \boldsymbol{\lambda}_k + \frac{\delta}{2}\nabla_1^2 T[\boldsymbol{\Phi}_{-(k+1/3)}, \boldsymbol{\theta}, \boldsymbol{u}_{-(k+1)}] \cdot \boldsymbol{J}^\top \cdot \boldsymbol{\lambda}_k \\ \boldsymbol{\lambda}_{k+2/3} = \boldsymbol{\lambda}_{k+1/3} + \delta\nabla_1^2 V[\boldsymbol{\Phi}_{-(k+2/3)}, \boldsymbol{\theta}, \boldsymbol{u}_{-(k+1)}] \cdot \boldsymbol{J}^\top \cdot \boldsymbol{\lambda}_{k+1/3} \\ \boldsymbol{\lambda}_{k+1} = \boldsymbol{\lambda}_{k+2/3} + \frac{\delta}{2}\nabla_1^2 T[\boldsymbol{\Phi}_{-(k+1)}, \boldsymbol{\theta}, \boldsymbol{u}_{-(k+1)}] \cdot \boldsymbol{J}^\top \cdot \boldsymbol{\lambda}_{k+2/3} + \boldsymbol{y}_{-(k+1)} \end{cases}$$

*Proof Corollary A.8.* Identical to the proof of Corollary A.7. □

### A.2.4 Proof of Theorem 3.3

**Theorem A.4.** *Assuming a HSSM (Def. A.5) and the optimization problem depicted in Def. A.6, we have:*

$$\forall \ell = 0, \cdots, N-1: \quad d_{\boldsymbol{\theta}^{(\ell)}} L = \lim_{\epsilon \to 0} \Delta \boldsymbol{\theta}^{(\ell)}(\epsilon),$$

*where $\Delta \boldsymbol{\theta}^{(\ell)}(\epsilon)$ can be recursively computed backwards, starting from the **top-most block** as:*

$$\overline{\boldsymbol{\Phi}}^{e,(N)}(\epsilon) = \mathcal{M}_{H^{(N-1)},\delta}[\overline{\boldsymbol{\Phi}}^{e,(N)}(\epsilon), \boldsymbol{\theta}^{(N-1)}, \widetilde{\boldsymbol{\Phi}}^{(N-1)}] + \epsilon \boldsymbol{\Sigma}_x \cdot \nabla_{\boldsymbol{\Phi}^{(N)}} L$$

$$\Delta \boldsymbol{\theta}^{(N-1)}(\epsilon) = \sum_{k=0}^{K-1} \nabla_2 \ell[\boldsymbol{\Phi}_k^e, \boldsymbol{\theta}^{(L-1)}, -k]$$

$$- \frac{\delta}{2\epsilon} \sum_{k=0}^{K-1} \left( \nabla_2 H^{1/2}[\boldsymbol{\Phi}_k^{e,(L)}(\epsilon), \boldsymbol{\theta}^{(N-1)}, \boldsymbol{\Phi}_{-k}^{(N-1)}] - \nabla_2 H^{1/2}[\boldsymbol{\Phi}_k^{e,(N)}(-\epsilon), \boldsymbol{\theta}^{(N-1)}, \boldsymbol{\Phi}_{-k}^{(N-1)}] \right),$$

$$\overline{\boldsymbol{\Delta}}_{\boldsymbol{\Phi}^{(N-1)}}(\epsilon) = -\frac{\delta}{2\epsilon} \left( \nabla_3 H^{1/2}[\overline{\boldsymbol{\Phi}}^{e,(N)}(\epsilon), \boldsymbol{\theta}^{(N-1)}, \widetilde{\boldsymbol{\Phi}}^{(N-1)}] - \nabla_3 H^{1/2}[\overline{\boldsymbol{\Phi}}^{e,(L)}(-\epsilon), \boldsymbol{\theta}^{(N-1)}, \widetilde{\boldsymbol{\Phi}}^{(N-1)}] \right)$$

*and subsequently for **upstream blocks**, i.e. $\forall \ell = N-2, \cdots, 0$:*

$$\overline{\boldsymbol{\Phi}}^{e,(\ell+1)}(\epsilon) = \mathcal{M}_{H^{(\ell)},\delta}[\overline{\boldsymbol{\Phi}}^{e,(\ell+1)}(\epsilon), \boldsymbol{\theta}^{(\ell)}, \widetilde{\boldsymbol{\Phi}}^{(\ell)}] + \epsilon \boldsymbol{\Sigma}_x \cdot \widetilde{\boldsymbol{\Delta}}_{\boldsymbol{\Phi}^{(\ell+1)}}(\epsilon)$$

$$\Delta \boldsymbol{\theta}^{(\ell)}(\epsilon) = -\frac{\delta}{2\epsilon} \sum_{k=0}^{K-1} \left( \nabla_2 H^{1/2}[\boldsymbol{\Phi}_k^{e,(\ell+1)}(\epsilon), \boldsymbol{\theta}^{(\ell)}, \boldsymbol{\Phi}_{-k}^{(\ell)}] - \nabla_2 H^{1/2}[\boldsymbol{\Phi}_k^{e,(\ell+1)}(-\epsilon), \boldsymbol{\theta}^{(\ell)}, \boldsymbol{\Phi}_{-k}^{(\ell)}] \right)$$

$$\boldsymbol{\Delta}_{\boldsymbol{\Phi}^{(\ell)}}(\epsilon) = -\frac{\delta}{2\epsilon} \left( \nabla_3 H^{1/2}[\overline{\boldsymbol{\Phi}}^{e,(\ell+1)}(\epsilon), \boldsymbol{\theta}^{(\ell)}, \widetilde{\boldsymbol{\Phi}}^{(\ell)}] - \nabla_3 H^{1/2}[\overline{\boldsymbol{\Phi}}^{e,(\ell+1)}(-\epsilon), \boldsymbol{\theta}^{(\ell)}, \widetilde{\boldsymbol{\Phi}}^{(\ell)}] \right)$$

*Proof.* Re-using the vectorized notations introduced in subsection 3.3 and used in Remark 5, the Lagrangian associated to the optimization problem depicted in Def. A.6 reads:

$$\mathcal{L} = \mathbf{1}^\top \cdot \ell[\widetilde{\boldsymbol{\Phi}}^{(L)}, \boldsymbol{\theta}^{(L-1)}] + \sum_{\ell=0}^{L-1} \mathrm{Tr}\left[ \left( \mathcal{M}_{H^{(\ell)},\delta}[\widetilde{\boldsymbol{\Phi}}^{(\ell+1)}, \boldsymbol{\theta}^{(\ell)}, \widetilde{\boldsymbol{u}}^{(\ell)}] - \widetilde{\boldsymbol{\Phi}}^{(\ell+1)} \right) \cdot \left( \overline{\boldsymbol{\lambda}}^{(\ell+1)} \right)^\top \right]$$

$$= \sum_{k=0}^{K-1} \ell[\boldsymbol{\Phi}_{-k}^{(L)}, \boldsymbol{\theta}^{(L)}, -k] + \sum_{\ell=0}^{L-1} \left( \boldsymbol{\lambda}_k^{(\ell+1)} \right)^\top \cdot \left( \mathcal{M}_{H^{(\ell)},\delta}[\boldsymbol{\Phi}_{-(k+1)}^{(\ell+1)}, \boldsymbol{\theta}^{(\ell)}, \boldsymbol{\Phi}_{-(k+1)}^{(\ell)}] - \boldsymbol{\Phi}_{-k}^{(\ell+1)} \right)$$

We proceed by induction on the block index starting from $\ell = L$.

**Initialization ($\ell = L$).** Let $\boldsymbol{\Phi}_k^{(N)}$ and $\boldsymbol{\lambda}_k^{(N)}$ for $k \in [\![0, K-1]\!]$ be the critical points of $\mathcal{L}$. By Theorem A.3:

$$d_{\boldsymbol{\theta}^{(N-1)}} L = \sum_{k=0}^{K-1} \nabla_2 \ell \left[ \boldsymbol{\Phi}_{-k}^{(N)}, \boldsymbol{\theta}^{(N-1)}, -k \right] + \partial_2 \mathcal{M}_{H^{(N-1)},\delta} \left[ \boldsymbol{\Phi}_{-(k+1)}^{(N)}, \boldsymbol{\theta}^{(L-1)}, \boldsymbol{\Phi}_{-(k+1)}^{(N-1)} \right]^\top \cdot \boldsymbol{\lambda}_k^{(N)},$$

with $(\boldsymbol{\lambda}_k^{(N)})_{k \in [\![0,K-1]\!]}$ satisfying the following recursion relationship:

$$\begin{cases} \boldsymbol{\lambda}_0^{(N)} = \nabla_1 \ell[\boldsymbol{\Phi}_0^{(N)}, \boldsymbol{\theta}^{(N-1)}, 0], \\ \forall k = 0, \cdots, K-1: \\ \quad \boldsymbol{\lambda}_{k+1}^{(N)} = \partial_1 \mathcal{M}_{H^{(N-1)},\delta} \left[ \boldsymbol{\Phi}_{-(k+1)}^{(N)}, \boldsymbol{\theta}^{(N-1)}, \boldsymbol{\Phi}_{-(k+1)}^{(N-1)} \right]^\top \cdot \boldsymbol{\lambda}_k^{(N)} + \nabla_1 \ell \left[ \boldsymbol{\Phi}_{-(k+1)}^{(N)}, \boldsymbol{\theta}, -(k+1) \right] \end{cases}$$

Given the definition of the dynamics of $\boldsymbol{\Phi}^{e,(N)}$ *by hypothesis*, we can directly apply Corollary A.7 to obtain:

$$d_{\boldsymbol{\theta}^{(N-1)}} L = \sum_{k=0}^{K-1} \nabla_2 \ell \left[ \boldsymbol{\Phi}_{-k}^{(N)}, \boldsymbol{\theta}^{(N-1)}, -k \right] - \delta \, \partial_\epsilon \left( \nabla_2 H^{(N-1),1/2} \left[ \boldsymbol{\Phi}_k^{e,(N)}(\epsilon), \boldsymbol{\theta}^{(L-1)}, \boldsymbol{\Phi}_{-(k+1)}^{(N-1)} \right] \right) \Big|_{\epsilon=0},$$

$$d_{\boldsymbol{\Phi}_{k-1}^{(N-1)}} L = \partial_3 \mathcal{M}_{H^{(N-1)},\delta}[\boldsymbol{\Phi}_{-(k+1)}^{(N)}, \boldsymbol{\theta}^{(N-1)}, \boldsymbol{\Phi}_{-(k+1)}^{(N-1)}]^\top \cdot \boldsymbol{\lambda}_k^{(N)}$$

$$= -\delta \, \partial_\epsilon \left( \nabla_3 H^{(N-1),1/2} \left[ \boldsymbol{\Phi}_k^{e,(N)}(\epsilon), \boldsymbol{\theta}^{(L-1)}, \boldsymbol{\Phi}_{-(k+1)}^{(N-1)} \right] \right) \Big|_{\epsilon=0}$$

**Induction** $(\ell + 1 \to \ell)$. Let us assume that the desired property is satisfied at layer $\ell + 1$. We denote again $\boldsymbol{\Phi}_k^{(\ell)}$ and $\boldsymbol{\lambda}_k^{(\ell)}$ for $k \in [\![0, K-1]\!]$ the critical point of $\mathcal{L}$. We have, for $k \in [\![0, K-1]\!]$:

$$
\begin{cases}
\boldsymbol{\lambda}_0^{(\ell)} = \partial_3 \mathcal{M}_{H^{(\ell)}, \delta} \left[ \boldsymbol{\Phi}_0^{(\ell+1)}, \boldsymbol{\theta}^{(\ell)}, \boldsymbol{\Phi}_0^{(\ell)} \right]^\top \cdot \boldsymbol{\lambda}_0^{(\ell+1)} \\
\forall k = 0, \cdots, K-1: \\
\quad \boldsymbol{\lambda}_{k+1}^{(\ell)} = \partial_1 \mathcal{M}_{H^{(\ell-1)}, \delta} \left[ \boldsymbol{\Phi}_{-(k+1)}^{(\ell)}, \boldsymbol{\theta}^{(\ell-1)}, \boldsymbol{\Phi}_{-(k+1)}^{(\ell-1)} \right]^\top \cdot \boldsymbol{\lambda}_k^{(\ell)} + \partial_3 \mathcal{M}_{H^{(\ell)}, \delta} \left[ \boldsymbol{\Phi}_{-(k+1)}^{(\ell+1)}, \boldsymbol{\theta}^{(\ell)}, \boldsymbol{\Phi}_{-(k+1)}^{(\ell)} \right]^\top \cdot \boldsymbol{\lambda}_k^{(\ell+1)}
\end{cases}
$$

Using the induction hypothesis at layer $(\ell + 1)$:

$$
\partial_3 \mathcal{M}_{H^{(\ell)}, \delta} \left[ \boldsymbol{\Phi}_{-(k+1)}^{(\ell+1)}, \boldsymbol{\theta}^{(\ell)}, \boldsymbol{\Phi}_{-(k+1)}^{(\ell)} \right]^\top \cdot \boldsymbol{\lambda}_k^{(\ell+1)} = -\delta \, \partial_\epsilon \left( \nabla_3 H^{(\ell), 1/2} \left[ \boldsymbol{\Phi}_k^{e, (\ell+1)}(\epsilon), \boldsymbol{\theta}^{(\ell)}, \boldsymbol{\Phi}_{-(k+1)}^{(\ell)} \right] \right) \Big|_{\epsilon=0}
$$
$$
= \lim_{\epsilon \to 0} \Delta_{\boldsymbol{\Phi}_{(k+1)}^{(\ell)}} (\epsilon)
$$

Therefore on the one hand, denoting $\Delta_{\boldsymbol{\Phi}^{(\ell)}} := \lim_{\epsilon \to 0} \Delta_{\boldsymbol{\Phi}^{(\ell)}}(\epsilon) \in \mathbb{R}^{K \times d_\Phi}$, the dynamics on $\boldsymbol{\lambda}$ rewrite:

$$
\begin{cases}
\boldsymbol{\lambda}_0^{(\ell)} = \Delta_{\boldsymbol{\Phi}_0^{(\ell)}} \\
\forall k = 0, \cdots, K-1: \\
\quad \boldsymbol{\lambda}_{k+1}^{(\ell)} = \partial_1 \mathcal{M}_{H^{(\ell-1)}, \delta} \left[ \boldsymbol{\Phi}_{-(k+1)}^{(\ell)}, \boldsymbol{\theta}^{(\ell-1)}, \boldsymbol{\Phi}_{-(k+1)}^{(\ell-1)} \right]^\top \cdot \boldsymbol{\lambda}_k^{(\ell)} + \Delta_{\boldsymbol{\Phi}_{(k+1)}^{(\ell)}}
\end{cases}
$$

On the other hand, the dynamics of $\boldsymbol{\Phi}^{e, (\ell)}$ read *by hypothesis*:

$$
\begin{cases}
\boldsymbol{\Phi}_0^{(\ell)} = \left( \boldsymbol{\Phi}_0^{(\ell)} \right)^\star + \epsilon \boldsymbol{\Sigma}_x \cdot \Delta_{\boldsymbol{\Phi}_0^{(\ell)}}(\epsilon) \\
\forall k = 0, \cdots, K-1: \\
\quad \boldsymbol{\Phi}_{k+1}^{e, (\ell)} = \mathcal{M}_{H^{(\ell-1)}, \delta} \left[ \boldsymbol{\Phi}_{k+1}^{e, (\ell)}, \boldsymbol{\theta}^{(\ell-1)}, \boldsymbol{\Phi}_{-(k+1)}^{(\ell-1)} \right] + \epsilon \boldsymbol{\Sigma}_x \cdot \Delta_{\boldsymbol{\Phi}_{(k+1)}^{(\ell)}}
\end{cases}
$$

using Corollary A.8 with $y = \Delta_{\boldsymbol{\Phi}^{(\ell)}}$, we conclude that:

$$
d_{\boldsymbol{\theta}^{(\ell)}} L = \lim_{\epsilon \to 0} \Delta \boldsymbol{\theta}^{(\ell)}(\epsilon)
$$

$\square$

**Remark 7.** *Theorem A.4, and more generally our definition of HSSMs (Def. A.5) assume that the connectivity pattern of the HRU units is a* linear chain. *Note that while we chose this hypothesis for the sake of clarity of our results and their derivations, Theorem A.4 could be seamlessly extended to any* Directed Acyclic Graph *(DAG) of HRUs. This allows, for instance as a simple and realistic case, to use skip connections across HRUs within HSSMs.*

**Remark 8.** *Note that RHEL chaining as prescribed by Theorem A.4 **implicitly chains RHEL and automatic differentiation**. Indeed, if H explicitly parametrizes feedforward mappings across HRUs as:*

$$
H^{(\ell)} \left[ \boldsymbol{\Phi}_k^{e, (\ell+1)}(\epsilon), \boldsymbol{\theta}^{(\ell)}, \boldsymbol{\Phi}_{-(k+1)}^{(\ell)} \right] = H^{(\ell)} \left[ \boldsymbol{\Phi}_k^{e, (\ell+1)}(\epsilon), \boldsymbol{\theta}_\alpha^{(\ell)}, F \left[ \boldsymbol{\Phi}_{-(k+1)}^{(\ell)}, \boldsymbol{\theta}_\beta^{(\ell)} \right] \right], \quad (23)
$$

*then, denoting $\boldsymbol{u}^{(\ell)} := F \left[ \boldsymbol{\Phi}_{-(k+1)}^{(\ell)}, \boldsymbol{\theta}_\beta^{(\ell)} \right]$, we have:*

$$
\partial_\epsilon \left( \nabla_3 H^{(\ell), 1/2} \left[ \boldsymbol{\Phi}_k^{e, (\ell+1)}(\epsilon), \boldsymbol{\theta}^{(\ell)}, \boldsymbol{\Phi}_{-(k+1)}^{(\ell)} \right] \right) \Big|_{\epsilon=0}
$$
$$
= \partial_1 F \left[ \boldsymbol{\Phi}_{-(k+1)}^{(\ell)}, \boldsymbol{\theta}_\beta^{(\ell)} \right]^\top \cdot \partial_\epsilon \left( \nabla_3 H^{(\ell), 1/2} \left[ \boldsymbol{\Phi}_k^{e, (\ell+1)}(\epsilon), \boldsymbol{\theta}_\alpha^{(\ell)}, \boldsymbol{u}^\ell \right] \right) \Big|_{\epsilon=0}
$$
$$
\approx \partial_1 F \left[ \boldsymbol{\Phi}_{-(k+1)}^{(\ell)}, \boldsymbol{\theta}_\beta^{(\ell)} \right]^\top \cdot \frac{1}{2\epsilon} \left( \nabla_3 H^{(\ell), 1/2} \left[ \boldsymbol{\Phi}_k^{e, (\ell+1)}(\epsilon), \boldsymbol{\theta}_\alpha^{(\ell)}, \boldsymbol{u}^\ell \right] - \nabla_3 H^{(\ell), 1/2} \left[ \boldsymbol{\Phi}_k^{e, (\ell+1)}(-\epsilon), \boldsymbol{\theta}_\alpha^{(\ell)}, \boldsymbol{u}^\ell \right] \right)
$$

*The red part is done by automatic differentiation and the blue part by RHEL. This underpins the implementation of RHEL chaining we used in our own code.*

### A.3 Looking ahead

#### A.3.1 Does RHEL break with dissipative dynamics?

**Foreword.** The applicability of RHEL is fundamentally limited by its restriction to conservative systems. Whenever dissipation is introduced, the system is no longer time-reversal invariant so that RHEL does not readily apply. One interesting question though is whether RHEL could be extended to the case where the time-reversed trajectory is *physically feasible*. Namely: the energy which is lost during the forward trajectory could be *exactly* pumped back into the system during the echo trajectory. Although the system may no longer be time-reversal invariant, we can ask ourselves whether perturbations of the time-reversed trajectory carry relevant error signals.

**Dissipative Hamiltonian model.** For this purpose, we introduce dissipation inside Hamiltonian dynamics as [70]:
$$\boldsymbol{\Phi}(-T) = \boldsymbol{x}, \quad \forall t \in [-T, 0] : \ \partial_t \boldsymbol{\Phi}(t) = (\boldsymbol{J} - \boldsymbol{R}) \cdot \nabla_{\boldsymbol{\Phi}} H[\boldsymbol{\Phi}(t), \boldsymbol{\theta}, \boldsymbol{u}(t)],$$

with $\boldsymbol{R}$ symmetric and positive definite. We set $\boldsymbol{R} = \boldsymbol{I}$ for simplicity – the loss of generality of this analysis is not an issue here as we state a negative result below.

**Time-reversed dynamics.** With this model, the corresponding time-reversed dynamics read:
$$
\begin{aligned}
\partial_t \widetilde{\boldsymbol{\Phi}}^{\star}(t) &= -\boldsymbol{\Sigma}_z \cdot \partial_t \boldsymbol{\Phi}(-t) \\
&= \boldsymbol{J} \nabla_{\boldsymbol{\Phi}^{\star}} H[\widetilde{\boldsymbol{\Phi}}^{\star}(t), \boldsymbol{\theta}, \boldsymbol{u}(-t)] + \boldsymbol{\Sigma}_z \cdot \boldsymbol{\Sigma}_z \cdot \nabla_{\boldsymbol{\Phi}^{\star}} H[\widetilde{\boldsymbol{\Phi}}^{\star}(t), \boldsymbol{\theta}, \boldsymbol{u}(-t)] \\
&= \boldsymbol{J} \nabla_{\boldsymbol{\Phi}^{\star}} H[\widetilde{\boldsymbol{\Phi}}^{\star}(t), \boldsymbol{\theta}, \boldsymbol{u}(-t)] + \nabla_{\boldsymbol{\Phi}^{\star}} H[\widetilde{\boldsymbol{\Phi}}^{\star}(t), \boldsymbol{\theta}, \boldsymbol{u}(-t)]
\end{aligned}
\tag{24}
$$

Therefore as expected: i) time-reversal invariance *no long holds*, ii) the sign of the dissipation in the forward trajectory is switched in the time-reversed trajectory.

**ASM with dissipative dynamics.** For this model, the ASM gradient estimate at time $t$ reads $\forall t \in [0, T]$:
$$g_{\boldsymbol{\theta}}(t) := \nabla_{\boldsymbol{\theta}} \ell[-t, \boldsymbol{\Phi}(-t), \boldsymbol{\theta}] + \nabla^2_{\boldsymbol{\Phi}, \boldsymbol{\theta}} H[\boldsymbol{\Phi}(-t), \boldsymbol{\theta}, \boldsymbol{u}(-t)] \cdot \boldsymbol{J}^{\top} \cdot \boldsymbol{\lambda}(t) - \nabla^2_{\boldsymbol{\Phi}, \boldsymbol{\theta}} H[\boldsymbol{\Phi}(-t), \boldsymbol{\theta}, \boldsymbol{u}(-t)] \cdot \boldsymbol{\lambda}(t)$$

with $\boldsymbol{\lambda}$ solving for the **adjoint** ODE:
$$
\begin{cases}
\boldsymbol{\lambda}(0) &= \boldsymbol{0} \\
\partial_t \boldsymbol{\lambda}(t) &= \nabla^2_{\boldsymbol{\Phi}} H[\boldsymbol{\Phi}(-t), \boldsymbol{\theta}, \boldsymbol{u}(-t)] \cdot \boldsymbol{J}^{\top} \cdot \boldsymbol{\lambda}(t) - {\color{blue}\nabla^2_{\boldsymbol{\Phi}} H[\boldsymbol{\Phi}(-t), \boldsymbol{\theta}, \boldsymbol{u}(-t)] \cdot \boldsymbol{\lambda}(t)} + \nabla_{\boldsymbol{\Phi}} \ell[-t, \boldsymbol{\Phi}(-t), \boldsymbol{\theta}]
\end{cases}
\tag{25}
$$

We highlight in blue the extra term in the adjoint dynamics which is induced by the dissipation.

**RHEL procedure with dissipative dynamics.** Following the intuition conveyed earlier, we now define the echo dynamics as perturbations of the time-reversed trajectory (Eq. (24)):
$$
\begin{cases}
\boldsymbol{\Phi}^e(0) &= \boldsymbol{\Phi}^{\star}(0) \\
\forall t \in [0, T] : \\
\partial_t \boldsymbol{\Phi}^e(t, \epsilon) &= \boldsymbol{J} \nabla_{\boldsymbol{\Phi}^e} H[\boldsymbol{\Phi}^e(t, \epsilon), \boldsymbol{\theta}, \boldsymbol{u}(-t)] + \nabla_{\boldsymbol{\Phi}^e} H[\boldsymbol{\Phi}^e(t, \epsilon), \boldsymbol{\theta}, \boldsymbol{u}(-t)] - \epsilon \boldsymbol{J} \nabla_{\boldsymbol{\Phi}^e} \ell[\boldsymbol{\Phi}^e(t, \epsilon), -t]
\end{cases}
$$
Proceeding exactly as we did for the derivation of Theorem A.2, we differentiate the above echo dynamics with respect to $\epsilon$ at $\epsilon = 0$, yielding:
$$
\begin{aligned}
\partial_t (\partial_\epsilon \boldsymbol{\Phi}^e(t, \epsilon)|_{\epsilon=0}) = &-\boldsymbol{\Sigma}_x \cdot \nabla^2_{\boldsymbol{\Phi}} H[\boldsymbol{\Phi}(-t, 0)] \cdot (\boldsymbol{J} \cdot \boldsymbol{\Sigma}_x) \cdot \partial_\epsilon \boldsymbol{\Phi}^e(t, \epsilon)|_{\epsilon=0} + \boldsymbol{\Sigma}_z \cdot \nabla^2_{\boldsymbol{\Phi}} H[\boldsymbol{\Phi}(-t, 0)] \cdot \boldsymbol{\Sigma}_z \cdot \partial_\epsilon \boldsymbol{\Phi}^e(t, \epsilon)|_{\epsilon=0} \\
&+ \boldsymbol{\Sigma}_x \cdot \nabla_{\boldsymbol{\Phi}} \ell[t, \boldsymbol{\Phi}(-t), \boldsymbol{\theta}]
\end{aligned}
$$
Multiplying by $\boldsymbol{\Sigma}_x$ on both sides and reusing the notation $\Delta^{\text{RHEL}}_{\boldsymbol{\Phi}}(t) := \boldsymbol{\Sigma}_x \cdot \partial_\epsilon \boldsymbol{\Phi}^e(t, \epsilon)|_{\epsilon=0}$, we have that:
$$
\begin{cases}
\Delta^{\text{RHEL}}_{\boldsymbol{\Phi}}(0) &= \boldsymbol{0} \\
\partial_t \Delta^{\text{RHEL}}_{\boldsymbol{\Phi}}(t) &= \nabla^2_{\boldsymbol{\Phi}} H[\boldsymbol{\Phi}(-t, 0)] \cdot \boldsymbol{J}^{\top} \cdot \Delta^{\text{RHEL}}_{\boldsymbol{\Phi}}(t) - {\color{red}\boldsymbol{J} \cdot \nabla^2_{\boldsymbol{\Phi}} H[\boldsymbol{\Phi}(-t, 0)] \cdot \boldsymbol{J} \cdot \Delta^{\text{RHEL}}_{\boldsymbol{\Phi}}(t)} + \nabla_{\boldsymbol{\Phi}} \ell[t, \boldsymbol{\Phi}(-t), \boldsymbol{\theta}]
\end{cases}
\tag{26}
$$

where we have highlighted in red the extra term induced by dissipation.

Comparing Eq. (25) and Eq. (26), we see that the red term inside the perturbed echo dynamics and the blue term inside the ASM dynamics differ. Therefore, it cannot be concluded that $\boldsymbol{\lambda}$ and $\Delta^{\text{RHEL}}_{\boldsymbol{\Phi}}$ are equal at all times, as it is the case for time-reversal invariant systems.

**Conclusion.** Using our methodology based on the equivalence with the continuous ASM, the echo dynamics cannot be seamlessly adapted to the case where the system is subject to dissipation, even when "pumping energy" back into the system". However, this does *not* mean that developing forward-only, perturbation-based methods for temporal credit assignment is impossible in general when systems exhibit dissipation. In a study released shortly after our submission [71], it was shown that a class of linear dissipative systems could be described with a Lagrangian formalism (which is closely related to our Hamiltonian formalism) and gradients estimated as finite differences of two dissipative trajectories, provided that the system is subject to periodic boundary conditions.

### A.3.2 A black-box version of RHEL?

**Foreword.** RHEL requires knowledge of the analytical expression of the Hamiltonian to implement its learning rule (Alg. 1). Based on the *Agnostic Equilibrium Propagation* [66] algorithm, we briefly sketch the most straightforward application of weight-centric homeostatic control to RHEL which, under some hypothesis on the model definition, yields an algorithm which may not require exact knowledge of the Hamiltonian. We caution the reader that this proposal is still brittle and would require much more thinking about the right modeling choices, the theoretical guarantees of the resulting algorithm and experimental validation thereof, which goes beyond the scope of this paper.

**Model description.** We propose *joint* dynamics of the neurons and weights as:

$$\boldsymbol{\Phi}(-T) = \boldsymbol{x}, \quad \forall t \in [-T, 0] : \; \partial_t \boldsymbol{\Phi}(t) = \boldsymbol{J}_\phi \cdot \nabla_{\boldsymbol{\Phi}} H[\boldsymbol{\Phi}(t), \boldsymbol{\Theta}(t), \boldsymbol{u}(t)],$$

$$\partial_t \boldsymbol{\Theta}(t) = \boldsymbol{J}_\theta \cdot \nabla_{\boldsymbol{\Theta}} \left[ H[\boldsymbol{\Phi}(t), \boldsymbol{\Theta}(t), \boldsymbol{u}(t)] + \frac{1}{2} \|\boldsymbol{c}(t) - \boldsymbol{\theta}(t)\|^2 \right],$$

where $\boldsymbol{c}$ denotes a control variable of the same dimensionality as $\boldsymbol{\theta}$ – namely: one control variable per weight – and $\boldsymbol{\Theta} := (\boldsymbol{\theta}^\top, \boldsymbol{\pi}_\theta^\top)^\top$ denote the concatenation of the position *and* momentum vector of the parameters. Note that this modelling choice is similar to that of the seminal HEB work where both the neurons *and* weights are dynamical variables [10].

**A candidate procedure.** Based on the above model proposal, a plausible procedure could read as follow:

- **First phase.** Throughout the first forward trajectory, we adjust the control variable $\boldsymbol{c}$ dynamically such that $\boldsymbol{\theta}$ remains stationary at its initial value $\boldsymbol{\theta}(0) := \boldsymbol{\theta}_0$:

  $$\forall t \in [-T, 0] : \; \partial_t \boldsymbol{\Phi}(t) = \boldsymbol{J}_\phi \cdot \nabla_{\boldsymbol{\Phi}} H[\boldsymbol{\Phi}(t), \boldsymbol{\Theta}(t), \boldsymbol{u}(t)],$$

  $$\partial_t \boldsymbol{\Theta}(t) = \boldsymbol{J}_\theta \cdot \nabla_{\boldsymbol{\theta}} \left[ H[\boldsymbol{\Phi}(t), \boldsymbol{\Theta}(t), \boldsymbol{u}(t)] + \frac{1}{2} \|\boldsymbol{c}(t) - \boldsymbol{\theta}(t)\|^2 \right] \approx 0$$

  $$\Rightarrow \forall t \in [-T, 0] : \; \boldsymbol{c}(t) \approx \boldsymbol{\theta}_0 + \nabla_{\boldsymbol{\theta}} H[\boldsymbol{\Phi}(t), \boldsymbol{\Theta}(t), \boldsymbol{u}(t)], \quad \nabla_{\boldsymbol{\pi}_\theta} H[\boldsymbol{\Phi}(t), \boldsymbol{\Theta}(t), \boldsymbol{u}(t)] \approx 0$$

  We denote $\{\boldsymbol{c}_0(t)\}_{t \in [-T, 0]}$ the resulting control trajectory.

- **Second phase.** During the echo dynamics, we apply the control $\boldsymbol{c}_0$ *in reverse*:

  $$\forall t \in [0, T] : \; \partial_t \boldsymbol{\Phi}^e(t) = \boldsymbol{J}_\phi \cdot \nabla_{\boldsymbol{\Phi}^e} H[\boldsymbol{\Phi}^e(t), \boldsymbol{\Theta}^e(t), \boldsymbol{u}(-t)] - \epsilon \boldsymbol{J} \nabla_{\boldsymbol{\Phi}^e} \ell[-t, \boldsymbol{\Phi}^e(t)],$$

  $$\partial_t \boldsymbol{\Theta}^e(t) = \boldsymbol{J}_\theta \cdot \nabla_{\boldsymbol{\Theta}} \left[ H[\boldsymbol{\Phi}^e(t), \boldsymbol{\Theta}^e(t), \boldsymbol{u}(-t)] + \frac{1}{2} \|\boldsymbol{c}_0(-t) - \boldsymbol{\theta}^e(t)\|^2 \right]$$

  The echo dynamics of the positions and momenta of $\Theta$ read:

  $$\begin{cases} \partial_t \boldsymbol{\theta}^e(t) & = \nabla_{\boldsymbol{\pi}_\theta^e} H[\boldsymbol{\Phi}^e(t), \boldsymbol{\Theta}^e(t), \boldsymbol{u}(-t)] \\ \partial_t \boldsymbol{\pi}_\theta^e(t) & = -\nabla_{\boldsymbol{\theta}} H[\boldsymbol{\Phi}^e(t), \boldsymbol{\Theta}^e(t), \boldsymbol{u}(-t)] - \boldsymbol{\theta}^e(t) + \boldsymbol{c}_0(-t) \end{cases} \tag{27}$$

  Note that because we apply the control in reverse and the parameter dynamics are Hamiltonian, the parameter dynamics are also *time-reversal invariant*. This means that if $\epsilon = 0$, executing Eq. (27) will maintain $\theta$ at $\theta_0$. Now assuming that $\epsilon > 0$, note that re-using the expression of $\boldsymbol{c}_0$ obtained above, the echo dynamics of $\boldsymbol{\pi}_\theta$ re-write:

$$\partial_t \boldsymbol{\pi}_\theta^e(t) \approx -\epsilon \left( \frac{\nabla_{\boldsymbol{\theta}} H[\boldsymbol{\Phi}^e(t), \boldsymbol{\Theta}^e(t), \boldsymbol{u}(-t)] - \nabla_{\boldsymbol{\theta}} H[\boldsymbol{\Phi}(-t), \boldsymbol{\Theta}_0, \boldsymbol{u}(-t)]}{\epsilon} \right) + \boldsymbol{\theta}_0 - \boldsymbol{\theta}^e(t)$$

$$= -\epsilon \Delta_{\boldsymbol{\theta}}^{\mathrm{aRHEL}}(t, \epsilon) - \nabla_{\boldsymbol{\theta}^e} \frac{1}{2} \|\boldsymbol{\theta}^e - \boldsymbol{\theta}_0\|^2, \tag{28}$$

where $\Delta_{\boldsymbol{\theta}}^{\mathrm{aRHEL}}(t, \epsilon) := \epsilon^{-1}(\nabla_{\boldsymbol{\theta}} H[\boldsymbol{\Phi}^e(t), \boldsymbol{\Theta}^e(t), \boldsymbol{u}(-t)] - \nabla_{\boldsymbol{\theta}} H[\boldsymbol{\Phi}(-t), \boldsymbol{\Theta}_0, \boldsymbol{u}(-t)])$. Note that if $\boldsymbol{\Theta}^e$ was "close enough" to $\boldsymbol{\Theta}_0$, then we would have $\Delta_{\boldsymbol{\theta}}^{\mathrm{aRHEL}}(t, \epsilon) \approx \Delta_{\boldsymbol{\theta}}^{\mathrm{RHEL}}(t, \epsilon)$. Additionally, note that playing the control $\boldsymbol{c}_0$ in reverse explicitly biases $\boldsymbol{\theta}^e$ towards $\boldsymbol{\theta}^0$ through an extra elastic contribution which appears in Eq. (28).

**Third phase.** At end of the previous procedure, Eq. (28) highlights that we could impart a momentum kick proportional to the gradient of the cost function, namely:

$$\boldsymbol{\pi}_{\boldsymbol{\theta}}^e(T) \approx -\epsilon \int_0^T dt \Delta_{\boldsymbol{\theta}}^{\mathrm{RHEL}}(t, \epsilon)$$

but what we really want to do is to update the parameters $\theta$. This pattern also appears in the original HEB algorithm. As done in [10], we would need a *decay step* which converts the momentum shift into a *position shift* in order to yield:

$$\boldsymbol{\theta}^e(T) \approx \boldsymbol{\theta}^e(-T) - \epsilon \int_0^T dt \Delta_{\boldsymbol{\theta}}^{\mathrm{RHEL}}(t, \epsilon) \approx \boldsymbol{\theta}^e(-T) - d_{\boldsymbol{\theta}(-T)} L$$

## A.4 Models and algorithms details

**Summary.** In this section, we provide details about our models and algorithms. More precisely:

- In section A.4.1, we first describe our *toy model* used inside Fig. 2 in terms of its Hamiltonian, resulting continuous-time dynamics and RHEL gradient estimators as prescribed by Theorem 3.1.

- In section A.4.2, we provide details about the HSSMs which we used in our experiments. We describe in greater details HSSMs made up of *linear* HRU blocks (section A.4.3). We describe their Hamiltonian, their resulting dynamics, the associated gradient estimators prescribed by RHEL and explain how *parallel scan* can be used on these models, especially when applying RHEL. Similarly, we describe HSSMs made up of *nonlinear* HRU blocks (section A.4.5).

- In section A.4.6, we show how the time discretization $\delta$ can itself be trained by absorbing it into the definition of the Hamiltonian function.

- Finally, in the light of Remark 8, we highlight in section A.4.7 how our implementation hybridizes *Automatic Differentiation* (AD) and RHEL using Algorithms 3–4.

### A.4.1 Toy model

In this section, we provide the gradient estimators for the parameters of the toy model. The toy model is a simple network of six mechanically coupled oscillators. Each oscillator is described by a state $\boldsymbol{\Phi^i} = (\phi^i, \pi^i)$ where $\phi^i \in \mathbb{R}$ is the position of the oscillator and $\pi^i \in \mathbb{R}$ is its momentum. It has a mass parameter $m_i$ and spring parameter $k_i$. Any pair of oscillators $(i, j)$ is coupled via the spring parameter $k_{ij}$. The input to the models is a time-varying external force $\boldsymbol{u}(t) \in \mathbb{R}$ coupled to oscillator 1. During the echo passes, the nudging force is modelled by a spring coupling with parameter $\epsilon \in \mathbb{R}$ to an external force $y(t) \in \mathbb{R}$. The Hamiltonian of the system is given by:

$$H[\boldsymbol{\Phi}, \boldsymbol{\theta}, \mathbf{u}] = \sum_i \frac{(\pi^i)^2}{2m_i} + \frac{1}{2}\sum_i k_i(\phi^i)^2 + \frac{1}{2}\sum_i \sum_{j>i} k_{ij}(\phi^j - \phi^i)^2 + \mathbf{u}\,\phi^1 \qquad (29)$$

Which gives the following equations of motion:

$$\begin{cases} \dot{\phi}^i = \frac{\pi^i}{m_i} \quad \text{for all } i \in \{1, 6\} \\ \dot{\pi}^i = -k_i\phi^i + \sum_{j, j \neq i} k_{ij}(\phi^j - \phi^i), \quad i \in \{2, 3, 5, 6\} \\ \dot{\pi}^1 = -k_1\phi^1 + \sum_{j, j \neq 1} k_{1j}(\phi^j - \phi^1) + \boldsymbol{u} \\ \dot{\pi}^4 = -k_4\phi^4 + \sum_{j, j \neq 4} k_{4j}(\phi^j - \phi^4) - \delta_e\epsilon(\boldsymbol{\phi}^4 - y) \end{cases}$$

where $\delta_e$ is the indicator function of the echo pass, it's equal to 1 during the echo pass and 0 otherwise.

**RHEL gradient estimators of the model parameters.** For the mass $m_i$, we have:

$$\begin{aligned} \Delta_{m_i}^{RHEL} &= -\frac{1}{2\epsilon}\left(\nabla_{m_i} H\left[\boldsymbol{\Phi}^e(t, \epsilon), \boldsymbol{\theta}, \boldsymbol{u}\right] - \nabla_{m_i} H\left[\boldsymbol{\Phi}^e(t, -\epsilon), \boldsymbol{\theta}, \boldsymbol{u}\right]\right) \\ &= \frac{1}{2\epsilon}\left(\frac{\left(\pi^i(t, \epsilon)\right)^2}{2m_i^2} - \frac{\left(\pi^i(t, -\epsilon)\right)^2}{2m_i^2}\right) \end{aligned} \qquad (30)$$

For the spring parameters $k_i$, we haves:

$$\begin{aligned} \Delta_{k_i}^{RHEL} &= -\frac{1}{2\epsilon}\left(\nabla_{k_i} H[\boldsymbol{\Phi}^e(t, \epsilon), \boldsymbol{\theta}, \boldsymbol{u}] - \nabla_{k_i} H[\boldsymbol{\Phi}^e(t, -\epsilon), \boldsymbol{\theta}, \boldsymbol{u}]\right) \\ &= -\frac{1}{2\epsilon}\left(\left(\phi^i(t, \epsilon)\right)^2 - \left(\phi^i(t, -\epsilon)\right)^2\right) \end{aligned} \qquad (31)$$

$$\qquad (32)$$

For the coupling parameters $k_{ij}$, we have:

$$\Delta_{k_{ij}}^{RHEL} = -\frac{1}{2\epsilon} \left( \nabla_{k_{ij}} H[\boldsymbol{\Phi}^e(t,\epsilon), \boldsymbol{\theta}, \boldsymbol{u}] - \nabla_{k_{ij}} H[\boldsymbol{\Phi}^e(t,-\epsilon), \boldsymbol{\theta}, \boldsymbol{u}] \right)$$
$$= -\frac{1}{2\epsilon}\left( \left(\phi^j(t,\epsilon) - \phi^i(t,\epsilon)\right)^2 - \left(\phi^j(t,-\epsilon) - \phi^i(t,-\epsilon)\right)^2 \right) \tag{33}$$

**RHEL gradient estimators of the state sensitivities**    For the state position sensitivities, we have:

$$\Delta_{\phi^i}^{RHEL} = -\frac{1}{2\epsilon} \boldsymbol{\Sigma}_x \left( \boldsymbol{\Phi}^e(t,\epsilon) - \boldsymbol{\Phi}^e(t,-\epsilon) \right)$$
$$= -\frac{1}{2\epsilon} \left( \pi^i(t,\epsilon) - \pi^i(t,-\epsilon) \right) \tag{34}$$

For the state momentum sensitivities, we have:

$$\Delta_{\pi^i}^{RHEL} = -\frac{1}{2\epsilon} \boldsymbol{\Sigma}_x \left( \boldsymbol{\Phi}^e(t,\epsilon) - \boldsymbol{\Phi}^e(t,-\epsilon) \right)$$
$$= -\frac{1}{2\epsilon} \left( \phi^i(t,\epsilon) - \phi^i(t,-\epsilon) \right) \tag{35}$$

### A.4.2    HSSM architecture

In this section, we outline the detailed architecture of a full multi-layer HSSM architecture. For the inference, we re-use the same stacking architecture of recurrent blocks and feedforward elements as the LinOSS model [29]. The model starts by encoding an input sequence $\overline{\boldsymbol{u}} \in \mathbb{R}^{d_{\boldsymbol{u}} \times K}$ via an affine transformation. The transformed sequence then progresses through multiple HSSM blocks, linear (see Appendix A.4.3), or nonlinear (see Appendix A.4.5), directly followed by nonlinear transformations. These transformations include the Gaussian error linear unit (GELU) [72] and the Gated Linear Unit (GLU) [73], defined as $\text{GLU}(\mathbf{x}) = \text{sigmoid}(\mathbf{W}_1\mathbf{x}) \circ \mathbf{W}_2\mathbf{x}$ where $\mathbf{W}_{1,2}$ represent trainable weight matrices, accompanied by a residual connection. The sequence output from the final block undergoes a second affine transformation to produce the model output.

The full linear and nonlinear HSSM is further presented in Algorithm 2. For clarity, when applying operations to sequence elements denoted with an overline (e.g., $\overline{\boldsymbol{u}}$), these operations are implicitly broadcast across the time dimension. Specifically, for any function $f$ applied to $\overline{\boldsymbol{u}}$, we have $f(\overline{\boldsymbol{u}})_t = f(\boldsymbol{u}_t)$ for all time steps $t \in \{1, 2, ..., K\}$.

For the inference of the linear HSSMs, we keep the same recurrent block as LinOSS with a slight change in the integrator (see A.4.3). For the nonlinear HSSM, we replace the recurrent block by a UniCORRN recurrent block [30] for which we use the same integrator as for the linear HSSM (see A.4.5).

---

**Algorithm 2** HSSM model

---

1: **Input:** Input sequence $\overline{\boldsymbol{u}}$, model type **type** $\in \{$linear, nonlinear$\}$
2: **Output:** HSSM output sequence $\overline{\boldsymbol{u}}$
3: $\overline{\boldsymbol{u}}^{(0)} \leftarrow \boldsymbol{W}_{enc}\overline{\boldsymbol{u}} + \boldsymbol{b}_{enc}$
4: **for** $\ell = 1, \ldots, N$ **do**
5:     **if type** $=$ linear **then**
6:         $\overline{\boldsymbol{\Phi}}^{(\ell)}, \_, \_ \leftarrow \text{LINEARHRU}(\overline{\boldsymbol{u}}^{(\ell-1)}, 0)$                    ▷ Via parallel scan
7:     **else**
8:         $\overline{\boldsymbol{\Phi}}^{(\ell)}, \_, \_ \leftarrow \text{NONLINEARHRU}(\overline{\boldsymbol{u}}^{(\ell-1)}, 0)$                    ▷ Sequentially
9:     **end if**
10:     $\overline{\boldsymbol{x}}^{(\ell)} \leftarrow \boldsymbol{C}\overline{\boldsymbol{\phi}}^{(\ell)} + \boldsymbol{D}\overline{\boldsymbol{u}}^{\ell-1}$
11:     $\overline{\boldsymbol{x_g}}^{(\ell)} \leftarrow \text{GELU}(\overline{\boldsymbol{x}}^{(\ell)})$
12:     $\overline{\boldsymbol{u}}^{(\ell)} \leftarrow \text{GLU}(\overline{\boldsymbol{x_g}}^{(\ell)} + \overline{\boldsymbol{u}}^{(\ell-1)})$
13: **end for**
14: $\overline{\boldsymbol{o}} \leftarrow \boldsymbol{W}_{dec}\overline{\boldsymbol{u}}^{(N)} + \boldsymbol{b}_{dec}$

---

### A.4.3 Linear HRU Block

**Hamiltonian of the recurrence.** The linear HRU block is the composition of a nonlinear spatial transformation and a linear recurrent transformation (see Eq. 17). Here we provide more details about the linear recurrence that is computed with the RHEL gradient estimator. The linear recurrence is defined by the following Hamiltonian:

$$H[\boldsymbol{\Phi}, \boldsymbol{\theta}, \boldsymbol{u}] = T[\boldsymbol{\pi}, \boldsymbol{\theta}, \boldsymbol{u}] + V[\boldsymbol{\phi}, \boldsymbol{\theta}, \boldsymbol{u}]$$
$$= \frac{1}{2}\|\boldsymbol{\pi}\|^2 + \left(\frac{1}{2}\boldsymbol{\phi}^\top \boldsymbol{A}\boldsymbol{\phi} - \boldsymbol{\phi}^\top \boldsymbol{B}\boldsymbol{u}\right) \tag{36}$$

**Dynamics.** The dynamics of the linear HRU block are defined by the following equations:

$$\begin{cases} \dot{\boldsymbol{\phi}} = \boldsymbol{\pi} \\ \dot{\boldsymbol{\pi}} = -\boldsymbol{A}\boldsymbol{\phi} + \boldsymbol{B}\boldsymbol{u} \end{cases} \tag{37}$$

Which, after time-discretization with the integrator defined in A.2.1, gives the following equations:

$$\begin{cases} \boldsymbol{\phi}_{k+1/3} = \boldsymbol{\phi}_k + \dfrac{\delta}{2}\nabla_{\boldsymbol{\pi}} T[\boldsymbol{\pi}_k, \boldsymbol{\theta}, \boldsymbol{u}_k] \\ \qquad\quad = \boldsymbol{\phi}_k + \dfrac{\delta}{2}\boldsymbol{\pi}_k \\ \boldsymbol{\pi}_{k+1/3} = \boldsymbol{\pi}_k \\ \boldsymbol{\phi}_{k+2/3} = \boldsymbol{\phi}_{k+1/3} \\ \boldsymbol{\pi}_{k+2/3} = \boldsymbol{\pi}_{k+1/3} + \delta\nabla_{\boldsymbol{\phi}} V[\boldsymbol{\phi}_{k+1/3}, \boldsymbol{\theta}, \boldsymbol{u}_k] \\ \qquad\quad = \boldsymbol{\pi}_{k+1/3} - \delta\boldsymbol{A}\boldsymbol{\phi}_{k+1/3} + \delta\boldsymbol{B}\boldsymbol{u}_k \\ \boldsymbol{\phi}_{k+1} = \boldsymbol{\phi}_{k+2/3} + \dfrac{\delta}{2}\nabla_{\boldsymbol{\pi}} T[\boldsymbol{\pi}_{k+2/3}, \boldsymbol{\theta}, \boldsymbol{u}_k] \\ \qquad\quad = \boldsymbol{\phi}_{k+2/3} + \dfrac{\delta}{2}\boldsymbol{\pi}_{k+2/3} \\ \boldsymbol{\pi}_{k+1} = \boldsymbol{\pi}_{k+2/3} \end{cases} \tag{38}$$

with the initial condition $\boldsymbol{\Phi}_{-K} = (\boldsymbol{\phi}_{-K}^\top, \boldsymbol{\pi}_{-K}^\top)^\top = \boldsymbol{x}$.

For the echo passes, the initial condition are $\boldsymbol{\Phi}_0^e = \boldsymbol{\Phi}_0^\star \pm \epsilon \boldsymbol{\Sigma}_x \cdot \Delta_{\boldsymbol{\Phi}} \ell[\boldsymbol{\Phi}_0, 0]$. The dynamics equations follow below, with modifications from equation 38 highlighted in blue::

$$\begin{cases} \boldsymbol{\phi}_{k+1/3}^e = \boldsymbol{\phi}_k^e + \dfrac{\delta}{2}\boldsymbol{\pi}_k^e \\ \boldsymbol{\pi}_{k+1/3}^e = \boldsymbol{\pi}_k^e \\ \boldsymbol{\phi}_{k+2/3}^e = \boldsymbol{\phi}_{k+1/3}^e \\ \boldsymbol{\pi}_{k+2/3}^e = \boldsymbol{\pi}_{k+1/3}^e - \delta\boldsymbol{A}\boldsymbol{\phi}_{k+1/3}^e + \delta\boldsymbol{B}\boldsymbol{u}_{-(k+1)} \\ \boldsymbol{\phi}_{k+1}^e = \boldsymbol{\phi}_{k+2/3}^e + \dfrac{\delta}{2}\boldsymbol{\pi}_{k+2/3}^e \textcolor{blue}{+ \epsilon\nabla_{\boldsymbol{\pi}}\ell[\boldsymbol{\Phi}_{-(k+1)}]} \\ \boldsymbol{\pi}_{k+1}^e = \boldsymbol{\pi}_{k+2/3}^e \textcolor{blue}{+ \epsilon\nabla_{\boldsymbol{\phi}}\ell[\boldsymbol{\Phi}_{-(k+1)}]} \end{cases} \tag{39}$$

**RHEL gradient estimators.** The gradient estimators of the parameters of the linear HRU are:

$$
\begin{aligned}
\Delta_{\boldsymbol{A}}^{RHEL}(k,\epsilon) &= -\frac{\delta}{2\epsilon}\left(\nabla_{\boldsymbol{A}}H^{1/2}[\boldsymbol{\Phi}_k^e(\epsilon),\boldsymbol{\theta},\boldsymbol{u}_{-(k+1)}] - \nabla_{\boldsymbol{A}}H^{1/2}[\boldsymbol{\Phi}_k^e(-\epsilon),\boldsymbol{\theta},\boldsymbol{u}_{-(k+1)}]\right)\\
&= -\frac{\delta}{4\epsilon}\left[\left(\boldsymbol{\phi}_{k+1/3}^e(\epsilon)^\top\boldsymbol{\phi}_{k+1/3}^e(\epsilon) + \boldsymbol{\phi}_{k+2/3}^e(\epsilon)^\top\boldsymbol{\phi}_{k+2/3}^e(\epsilon)\right)\right.\\
&\qquad\left. - \left(\boldsymbol{\phi}_{k+1/3}^e(-\epsilon)^\top\boldsymbol{\phi}_{k+1/3}^e(-\epsilon) + \boldsymbol{\phi}_{k+2/3}^e(-\epsilon)^\top\boldsymbol{\phi}_{k+2/3}^e(-\epsilon)\right)\right]\\
&= -\frac{\delta}{2\epsilon}\left[\left(\boldsymbol{\phi}_{k+1/3}^e(\epsilon)\right)^\top\left(\boldsymbol{\phi}_{k+1/3}^e(\epsilon)\right) - \left(\boldsymbol{\phi}_{k+1/3}^e(-\epsilon)\right)^\top\left(\boldsymbol{\phi}_{k+1/3}^e(-\epsilon)\right)\right] \quad (40)
\end{aligned}
$$

$$
\begin{aligned}
\Delta_{\boldsymbol{B}}^{RHEL}(k,\epsilon) &= -\frac{\delta}{2\epsilon}\left(\nabla_{\boldsymbol{B}}H^{1/2}[\boldsymbol{\Phi}_k^e(\epsilon),\boldsymbol{\theta},\boldsymbol{u}_{-(k+1)}] - \nabla_{\boldsymbol{B}}H^{1/2}[\boldsymbol{\Phi}_k^e(-\epsilon),\boldsymbol{\theta},\boldsymbol{u}_{-(k+1)}]\right)\\
&= \frac{\delta}{4\epsilon}\left[\left(\boldsymbol{\phi}_{k+1/3}^e(\epsilon) + \boldsymbol{\phi}_{k+2/3}^e(\epsilon)\right)\boldsymbol{u}_{-(k+1)}^\top\right.\\
&\qquad\left. - \left(\boldsymbol{\phi}_{k+1/3}^e(-\epsilon) + \boldsymbol{\phi}_{k+2/3}^e(-\epsilon)\right)\boldsymbol{u}_{-(k+1)}^\top\right]\\
&= \frac{\delta}{2\epsilon}\left[\boldsymbol{\phi}_{k+1/3}^e(\epsilon)\boldsymbol{u}_{-(k+1)}^\top - \boldsymbol{\phi}_{k+1/3}^e(-\epsilon)\boldsymbol{u}_{-(k+1)}^\top\right] \quad (41)
\end{aligned}
$$

The gradient estimator with respect to the input of the recurrent transformation is:

$$
\begin{aligned}
\Delta_{\boldsymbol{u}}^{RHEL}(k,\epsilon) &= -\frac{\delta}{2\epsilon}\left(\nabla_{\boldsymbol{u}}H^{1/2}[\boldsymbol{\Phi}_k^e(\epsilon),\boldsymbol{\theta},\boldsymbol{u}_{-(k+1)}] - \nabla_{\boldsymbol{u}}H^{1/2}[\boldsymbol{\Phi}_k^e(-\epsilon),\boldsymbol{\theta},\boldsymbol{u}_{-(k+1)}]\right)\\
&= \frac{\delta}{4\epsilon}\left[\boldsymbol{B}^\top\left(\boldsymbol{\phi}_{k+1/3}^e(\epsilon) + \boldsymbol{\phi}_{k+2/3}^e(\epsilon)\right)\right.\\
&\qquad\left. - \boldsymbol{B}^\top\left(\boldsymbol{\phi}_{k+1/3}^e(-\epsilon) + \boldsymbol{\phi}_{k+2/3}^e(-\epsilon)\right)\right]\\
&= \frac{\delta}{2\epsilon}\left[\boldsymbol{B}^\top\boldsymbol{\phi}_{k+1/3}^e(\epsilon) - \boldsymbol{B}^\top\boldsymbol{\phi}_{k+1/3}^e(-\epsilon)\right] \quad (42)
\end{aligned}
$$

**Parallel Scan.** Similarly to the LinOSS model [29], we can compute the recurrence of Linear HRU with a parallel scan [5] to reduce the computational time. To implement the parallel scan, we need to put the discretized dynamics in a form that can be computed in parallel. For this we vectorize the equation 38:

$$
\begin{aligned}
\boldsymbol{\Phi}_{k+1} &= \begin{bmatrix}\boldsymbol{I} & \frac{\delta}{2}\boldsymbol{I}\\ \boldsymbol{0} & \frac{\delta}{2}\boldsymbol{I}\end{bmatrix}\cdot\boldsymbol{\Phi}_{k+2/3}\\
&= \begin{bmatrix}\boldsymbol{I} & \frac{\delta}{2}\boldsymbol{I}\\ \boldsymbol{0} & \boldsymbol{I}\end{bmatrix}\cdot\left(\begin{bmatrix}\boldsymbol{I} & \boldsymbol{0}\\ -\delta\boldsymbol{A} & \boldsymbol{I}\end{bmatrix}\cdot\boldsymbol{\Phi}_{k+1/3} + \begin{bmatrix}\boldsymbol{0}\\ \delta\boldsymbol{B}\cdot\boldsymbol{u}_k\end{bmatrix}\right)\\
&= \begin{bmatrix}\boldsymbol{I} & \frac{\delta}{2}\boldsymbol{I}\\ \boldsymbol{0} & \boldsymbol{I}\end{bmatrix}\cdot\begin{bmatrix}\boldsymbol{I} & \boldsymbol{0}\\ -\delta\boldsymbol{A} & \boldsymbol{I}\end{bmatrix}\cdot\begin{bmatrix}\boldsymbol{I} & \frac{\delta}{2}\boldsymbol{I}\\ \boldsymbol{0} & \boldsymbol{I}\end{bmatrix}\cdot\boldsymbol{\Phi}_k + \begin{bmatrix}\frac{\delta^2}{2}\boldsymbol{B}\cdot\boldsymbol{u}_k\\ \delta\boldsymbol{B}\cdot\boldsymbol{u}_k\end{bmatrix}
\end{aligned}
$$

which gives us the discrete dynamics in matrix form:

$$
\boldsymbol{\Phi}_{k+1} = M\boldsymbol{\Phi}_k + F_k \quad (43)
$$

with:

$$
M = \begin{bmatrix}I - \frac{\delta^2}{2}\boldsymbol{A} & \delta[I - \frac{\delta^2}{4}\boldsymbol{A}]\\ -\delta\boldsymbol{A} & I - \frac{\delta^2}{2}\boldsymbol{A}\end{bmatrix}, \quad F_k = \begin{bmatrix}\frac{\delta^2}{2}\boldsymbol{B}\cdot\boldsymbol{u}_k\\ \delta\boldsymbol{B}\cdot\boldsymbol{u}_k\end{bmatrix} \quad (44)
$$

To compute the echo passes $\boldsymbol{\Phi}_k^e, k \in \{1,\cdots,K\}$, we only need to adapt $F_k$ to get for the positive nudging $+\epsilon$:

$$
F_k^e = \begin{bmatrix}\frac{\delta^2}{2}\boldsymbol{B}\cdot\boldsymbol{u}_{-(k+1)} + \epsilon\nabla_{\boldsymbol{\pi}}\ell[\boldsymbol{\Phi}_{-(k+1)}]\\ \delta\boldsymbol{B}\cdot\boldsymbol{u}_{-(k+1)} + \epsilon\nabla_{\boldsymbol{\phi}}\ell[\boldsymbol{\Phi}_{-(k+1)}]\end{bmatrix} \quad (45)
$$

### A.4.4 Relation between our Linear HRU and the one of LinOSS

In this section, we expand on the relationship between our Linear HRU and the one used in the LinOSS model [29]. We first recall that we are using the base continuous-time dynamics as in the LinoSS model (equations 17).

**RHEL requires a non-dissipative integrator.** RHEL in continuous time requires non-dissipative systems (see ). Hence, to extend RHEL to discrete time, we sought an integrator that preserves this property and had to rule out the first integrator used in the LinOSS model (the Implicit time integration (IM), equations 3 and 4 of the [29]). We note that this RHEL requirement can be a weakness for some Machine Learning tasks. As highlighted in the empirical results of [29], performance depends on the data distribution. On long-sequence tasks like Worms (17,984 timesteps), dissipative systems have an advantage because dissipation enables forgetting of past information. In contrast, non-dissipative Hamiltonian systems must preserve the entire trajectory history and may require larger models for equivalent performance. On the contrary, on data distributions coming from non-dissipative Hamiltonian systems, non-dissipative integrators are better models ([29], Appendix C.2).

**Our adaptation of the IMEX integrator of LinOSS.** In the LinOSS model, a non-dissipative integrator was also proposed, the Implicit-explicit time integrator (IMEX). Adapted to our notation (from equations, the equations of the integrator IMEX are:

$$\begin{cases} \boldsymbol{\pi}_{k+1} = \boldsymbol{\pi}_k + \delta\left(-\boldsymbol{A}\boldsymbol{\phi}_k + \boldsymbol{B}\boldsymbol{u}_{k+1}\right) \\ \boldsymbol{\phi}_{k+1} = \boldsymbol{\phi}_k + \delta\,\boldsymbol{\pi}_{k+1} \end{cases}$$

Re-using the Euler integrators defined in A.3, the IMEX integrator consists of doing an Euler step to change the momentum, followed by an Euler step to change the position:

$$\mathcal{M}_{H,\delta}^{\text{IMEX}} := \mathcal{M}_{T,\delta} \circ \mathcal{M}_{V,\delta}$$

In compact matrix form the IMEX integrator writes as:

$$\boldsymbol{\Phi}_{k+1} = M^{\text{IMEX}}\,\boldsymbol{\Phi}_k + F_{k+1}^{\text{IMEX}}, \tag{46}$$

$$\text{where} \quad M^{\text{IMEX}} = \begin{bmatrix} I - \delta^2\boldsymbol{A} & \delta\,I \\ -\delta\,\boldsymbol{A} & I \end{bmatrix}, \qquad F_{k+1}^{\text{IMEX}} = \begin{bmatrix} \delta^2\,\boldsymbol{B}\boldsymbol{u}_{k+1} \\ \delta\,\boldsymbol{B}\boldsymbol{u}_{k+1} \end{bmatrix}. \tag{47}$$

This shows that IMEX is still different from the integrator we used in our linear HRU (Eq. 44). A quick calculation shows that:

$$\left(\mathcal{M}_{H,\delta}^{\text{IMEX}}\right)^{-1} = \left(\mathcal{M}_{T,\delta} \circ \mathcal{M}_{V,\delta}\right)^{-1} = \mathcal{M}_{V,-\delta} \circ \mathcal{M}_{T,-\delta} \ \neq \ \mathcal{M}_{T,-\delta} \circ \mathcal{M}_{V,-\delta} = \mathcal{M}_{H,-\delta}^{\text{IMEX}},$$

which violates the discrete-time reversibility required to prove RHEL (Lemma A.4). On the contrary, the leapfrog integrator is the simple rearrangement of update steps:

$$\mathcal{M}_{H,\delta} \ = \ \mathcal{M}_{T,\delta/2} \circ \mathcal{M}_{V,\delta} \circ \mathcal{M}_{T,\delta/2},$$

which has the desired properties, as proven in Lemma A.4.

### A.4.5 Nonlinear HRU

**Hamiltonian of the recurrence.** The nonlinear HRU block is the composition of a nonlinear spatial transformation and a nonlinear recurrent transformation (see Eq. 18). The Hamiltonian of the nonlinear HRU block is defined by the following equations:

$$H[\boldsymbol{\Phi}, \boldsymbol{\theta}, \boldsymbol{u}] = T[\boldsymbol{\pi}, \boldsymbol{\theta}, \boldsymbol{u}] + V[\boldsymbol{\phi}, \boldsymbol{\theta}, \boldsymbol{u}]$$
$$= \frac{1}{2}\|\boldsymbol{\pi}\|^2 + \frac{\alpha}{2}\|\boldsymbol{\phi}\|^2 + \left(\boldsymbol{A}^{-\top} \cdot \log\left(\cosh\left(\boldsymbol{A} \cdot \boldsymbol{\phi} + \boldsymbol{B} \cdot \boldsymbol{u} + \boldsymbol{b}\right)\right)\right) \tag{48}$$

**Dynamics.** The dynamics of the nonlinear HRU block are defined by the following equations:

$$\begin{cases} \dot{\phi} = \pi \\ \dot{\pi} = -\left( \tanh\left(A\phi + Bu + b\right) + \alpha\phi \right) \end{cases} \tag{49}$$

Which, after time-discretization with the integrator defined in A.2.1, gives the following equations:

$$\begin{cases} \phi_{k+1/3} = \phi_k + \dfrac{\delta}{2}\nabla_{\pi}T[\pi_k, \theta, u_k] \\ \qquad\quad = \phi_k + \dfrac{\delta}{2}\pi_k \\ \pi_{k+1/3} = \pi_k \\ \phi_{k+2/3} = \phi_{k+1/3} \\ \pi_{k+2/3} = \pi_{k+1/3} + \delta\nabla_{\phi}V[\phi_{k+1/3}, \theta, u_k] \\ \qquad\quad = \pi_{k+1/3} - \delta(\tanh\left(A\phi_{k+1/3} + Bu_k + b\right) + \alpha\phi_{k+1/3}) \\ \phi_{k+1} = \phi_{k+2/3} + \dfrac{\delta}{2}\nabla_{\pi}T[\pi_{k+2/3}, \theta, u_k] \\ \qquad\quad = \phi_{k+2/3} + \dfrac{\delta}{2}\pi_{k+2/3} \\ \pi_{k+1} = \pi_{k+2/3} \end{cases} \tag{50}$$

### A.4.6 Differentiating the time-discretization

In training HRUs, one can also train the multidimensional time-discretization $\delta \in \mathbb{R}^{d_\Phi \times d_\Phi}$ that is assumed to be diagonal. For this, it suffices to reparametrize the integrator and the Hamiltonian, so that the time discretization becomes a parameter of the Hamiltonian. Hence the HRU equation 9 becomes:

$$\Phi_{k+1} = \mathcal{M}_{\hat{H},1}[\Phi_k, \theta, u_k, \delta] \quad \forall k = -K \cdots -1, \tag{51}$$

with $\hat{H}$ is a reparametrization of the Hamiltonian $H$ such that $\mathcal{M}_{\hat{H},1}[\Phi_k, \theta, u_k, \delta] = \mathcal{M}_{H,\delta}[\Phi_k, \theta, u_k]$.

For instance, for the linear HRU, we can reparametrize the Hamiltonian as:

$$\hat{H}[\Phi, \theta, u, \delta] = \frac{1}{2}\pi^\top \delta\pi + \left(\frac{1}{2}\phi^\top(\delta A)\phi - \phi^\top(\delta B)u\right) \tag{52}$$

And for the nonlinear HRU, we can reparametrize the Hamiltonian as:

$$\hat{H}[\Phi, \theta, u, \delta] = \frac{1}{2}\pi^\top \delta\pi + \frac{\alpha}{2}\phi^\top \delta\phi + \frac{1}{2}\left(\delta A^{-\top} \cdot \log\left(\cosh\left(A \cdot \phi + B \cdot u + b\right)\right)\right) \tag{53}$$

$$\tag{54}$$

Note that to match with previous implementations of the nonlinear HRU [30], we used the following parametrization for the discretization step:

$$\hat{H}[\Phi, \theta, u, \delta] = \frac{1}{2}\pi^\top \sigma(\delta)\pi + \frac{\alpha}{2}\phi^\top \sigma(\delta)\phi + \frac{1}{2}\left(\sigma(\delta)A^{-\top} \cdot \log\left(\cosh\left(A \cdot \phi + B \cdot u + b\right)\right)\right) \tag{55}$$

where $\sigma(\delta)$ is a diagonal matrix with $\sigma(\delta)_{ii} = 0.5 + 0.5\tanh(\delta_{ii}/2)$.

### A.4.7 Echo passes with automatic differentiation

As mentioned in Remark 8, learning in HSSMs with RHEL involves chaining RHEL gradient estimators with Automatic Differentiation (AD). This design makes RHEL implementation readily compatible with modern automatic differentiation frameworks such as PyTorch or JAX. These libraries provide the capability to create custom functions that, when invoked within a computational

---
**Algorithm 3** Custom Reverse Mode Automatic Differentiation for an arbitrary HRU
---
1: @custom_autodiff
2: **function** HRU($\boldsymbol{\theta}_{HRU}, \overline{\boldsymbol{u}}$)
3:     $\overline{\overline{\boldsymbol{\Phi}}} \leftarrow$ HRU-HELPER($\mathbf{0}, \boldsymbol{\theta}_{HRU}, \overline{\boldsymbol{u}}, \overline{\mathbf{0}}$)
4:     **return** $\overline{\overline{\boldsymbol{\Phi}}}$
5: **end function**

6: **function** FORWARDHRU($\boldsymbol{\theta}_{HRU}, \overline{\boldsymbol{u}}$)
7:     $\overline{\overline{\boldsymbol{\Phi}}} \leftarrow$ HRU-HELPER($\mathbf{0}, \boldsymbol{\theta}_{HRU}, \overline{\boldsymbol{u}}, \overline{\mathbf{0}}$)
8:     **return** $\overline{\overline{\boldsymbol{\Phi}}}, \boldsymbol{\Phi}_K$
9: **end function**

10: **function** BACKWARDHRU($\boldsymbol{r}, \overline{\boldsymbol{g}}$, input_forward)
11:     $\boldsymbol{\theta}_{HRU}, \overline{\boldsymbol{u}} \leftarrow$ input_forward
12:     $\tilde{\boldsymbol{u}} \leftarrow$ REVERSE($\overline{\boldsymbol{u}}$)                            ▷ Reverse the input sequence
13:     $\boldsymbol{\Phi}_K \leftarrow \boldsymbol{r}$
14:     $\boldsymbol{\Phi}_{0,\pm\epsilon}^e \leftarrow \boldsymbol{\Sigma}_x \boldsymbol{\Phi}_K + \epsilon \boldsymbol{\Sigma}_x \boldsymbol{g}_K$
15:     $\overline{\boldsymbol{\Phi}}_{\pm\epsilon}^e \leftarrow$ HRU-HELPER($\boldsymbol{\Phi}_{0,\pm\epsilon}^e, \boldsymbol{\theta}_{HRU}, \overline{\boldsymbol{u}}, \overline{\boldsymbol{g}}$)
16:     $\Delta_{\boldsymbol{\theta}_{HRU}}, \Delta_{\overline{\boldsymbol{u}}} \leftarrow$ LEARNINGHRU($\overline{\boldsymbol{\Phi}}_{\pm\epsilon}^e$)
17:     **return** $\Delta_{\boldsymbol{\theta}_{HRU}}, \Delta_{\overline{\boldsymbol{u}}}$    ▷ Loss gradient with respect to the input of FORWARDHRU($\cdot, \cdot$)
18: **end function**

19: HRU.define_custom_autodiff(ForwardHRU, BackwardHRU)
---

---
**Algorithm 4** Helper functions for custom Automatic Differentiation
---
1: **function** HRU-HELPER($\boldsymbol{\Phi}_0, \boldsymbol{\theta}_{HRU}, \overline{\boldsymbol{u}}, \overline{\boldsymbol{n}}$)    ▷ See Dynamics. in App. A.4.3 and A.4.5 for concrete examples
2:     **Input:** Initial State $\boldsymbol{\Phi}_0$, Parameters of the recurrence $\boldsymbol{\theta}_{HRU}$, Inputs $\overline{\boldsymbol{u}}$, Nudging $\overline{\boldsymbol{n}}$
3:     **Output:** State sequence $\overline{\overline{\boldsymbol{\Phi}}}$
4: **end function**

5: **function** LEARNINGHRU($\overline{\boldsymbol{\Phi}}_{\pm\epsilon}^e, \overline{\boldsymbol{u}}$) ▷ See RHEL gradient estimator in App. A.4.3 for a concrete example
6:     **Input:** Echo passes $\overline{\boldsymbol{\Phi}}_{\pm\epsilon}^e$, input sequence time-reversed $\tilde{\boldsymbol{u}}$
7:     **Output:** Loss gradient w.r.t to the input of FORWARDHRU($\cdot, \cdot$): $\Delta_{\boldsymbol{\theta}_{HRU}}, \Delta_{\overline{\boldsymbol{u}}}$
8: **end function**
---

graph, are automatically differentiated. Algorithm 3 demonstrates how to implement RHEL/AD chaining for the recurrent component of an HRU block.

To implement a custom autodiff function HRU, we must define two additional functions required by AD: ForwardHRU and BackwardHRU. The ForwardHRU function is invoked when the HRU function is called within a computational graph to be differentiated. The ForwardHRU function returns two elements: its output for computing the remainder of the computational graph $\overline{\overline{\boldsymbol{\Phi}}}$, and a *residual* to be stored for the backward pass. For the HRU, the residual consists only of the final state $\boldsymbol{\Phi}_K$ from the forward pass. The BackwardHRU function is called during graph backpropagation. It receives as input the loss gradient from the upstream portion of the computational graph $\overline{g}$, the input from the forward pass $\overline{u}$, and the residual $\boldsymbol{\Phi}_K$. The BackwardHRU function computes the gradient of the loss with respect to the HRU block parameters, which is subsequently used to update the HRU parameters. It also computes the gradient of the loss with respect to the forward pass input $\overline{u}$, which is then used to propagate the loss gradient backward through the computational graph.

These three functions are then registered with the autodiff library using the HRU.define_custom_autodiff function, enabling the library to automatically differentiate the HRU function when it is called within a computational graph.

The three functions utilize two helper functions: `HRU-HELPER` and `LearningHRU`. The `HRU-HELPER` function implements the HRU dynamics and is employed in both the forward and backward passes. The `LearningHRU` function implements the RHEL gradient estimator and is used in the backward pass to compute the gradient of the loss with respect to the HRU block parameters.s

### A.5 Experimental details

**Summary.** This last section provides additional experimental details. More precisely:

- We provide details about the datasets at use, the devices we used, as well as detailed hyperparameters.
- Importantly, we detail **how RHEL can be subject to numerical *underflow* and how we mitigated this issue** – see Fig. 5–6 for details.

#### A.5.1 Datasets

This series of tasks was recently introduced as a subset of the University of East Anglia (UEA) datasets with the longest sequences for increased difficulty and recently used to benchmark the linear HSSM previously introduced [29]

The classification datasets are drawn from a recently introduced benchmark [34] that selects a subset of the University of East Anglia (UEA) datasets [35], specifically choosing those with the longest sequences to increase difficulty, and which has been recently employed to evaluate the linear HSSM model [29]. These datasets include EigenWorms (17,984 sequence length, 5 classes), SelfRegulationSCP1 (896 length, 2 classes), SelfRegulationSCP2 (1,152 length, 2 classes), EthanolConcentration (1,751 length, 4 classes), Heartbeat (405 length, 2 classes), and MotorImagery (3,000 length, 2 classes).

Additionally, we evaluate our HSSMs on the PPG-DaLiA dataset [36], a multivariate time series regression dataset designed for heart rate prediction using data collected from a wrist-worn device. It includes recordings from fifteen individuals, each with approximately 150 minutes of data sampled at a maximum rate of 128 Hz. The dataset consists of six channels: blood volume pulse, electrodermal activity, body temperature, and three-axis acceleration. After splitting the data, a sliding window of length 49920 and step size 4992 is applied, representing a challenging very long-range interaction task.

#### A.5.2 Simulation details and ressource consumption

**Simulation details.** The code to run the experiments is implemented using the JAX auto-differentiation framework [74]. All experiments were run on Nvidia V100 GPUs, except for the PPG experiments, which were run on Nvidia Tesla A100 GPUs due to larger memory demands.

**Computational complexity.** We do *not* claim any superiority of RHEL over BPTT on conventional *digital* hardware in terms of runtime or memory consumption. As mentioned in the "Future work" paragraph in the main text, turning RHEL into a compelling alternative to BPTT on digital accelerators with respect to these metrics is a research direction of its own. In its current form, RHEL exhibits the *same* computational and memory complexity as BPTT.

- **Single Hamiltonian Recurrent Unit (HRU).** For a HRU with hidden dimension $d$ and $T$ timesteps, both RHEL and BPTT have a time complexity of $\mathcal{O}(Td^2)$ under naive recurrence, corresponding to $d^2$ multiply–accumulate operations per step. For *linear* HRUs that support the use of *parallel scan* [1], this cost can be reduced to $\mathcal{O}(\log(T)d^2)$. In both cases, the memory complexity of BPTT and RHEL is $\mathcal{O}(Td)$ when storing all activations and $\mathcal{O}(d)$ when recomputing them backward from the final state, leveraging the *time-reversibility* of the symplectic integrator inside HRUs. This memory efficiency is a *model* feature shared by both algorithms, not a key differentiator of RHEL.
- **Hamiltonian State-Space Model (HSSM).** For a HSSM composed of multiple HRUs, the above computational cost scales linearly with the number of HRUs for both RHEL and BPTT. For a given HRU inside a HSSM, there is an additional $\mathcal{O}(Td)$ memory cost for storing the input sequence fed into that HRU.

**Empirical runtime and memory usage.** We report below the average runtime and peak GPU memory consumption of RHEL and BPTT across representative datasets. As expected, the two algorithms exhibit comparable performance profiles. In some cases, RHEL yields moderate speedups for nonlinear models—likely stemming from implementation details rather than fundamental algorithmic differences.

Table 3: Runtime and memory comparison between RHEL and BPTT.

| Dataset | Model | RHEL Time (s) | BPTT Time (s) | RHEL Mem (GB) | BPTT Mem (GB) |
|---|---|---|---|---|---|
| PPG | Linear | 11.50±4.72 | 10.42±6.86 | 11.387±0.001 | 11.387±0.002 |
| PPG | Nonlinear | 1150.32±26.78 | 1609.05±24.74 | 11.728±0.013 | 11.731±0.013 |
| EigenWorms | Linear | 3.16±5.38 | 3.21±6.24 | 0.716±0.001 | 0.716±0.001 |
| EigenWorms | Nonlinear | 65.06±2.33 | 117.66±2.43 | 0.759±0.004 | 0.760±0.003 |
| SCP2 | Linear | 15.59±15.53 | 13.02±8.87 | 0.104±0.000 | 0.103±0.000 |
| SCP2 | Nonlinear | 27.96±42.50 | 41.28±29.39 | 0.107±0.000 | 0.108±0.001 |

**Observations.**

- **Memory usage** is s comparable between RHEL and BPTT as expected.

- **Runtime** is generally comparable as expected too, with RHEL showing significant speedup for nonlinear models (e.g., 2× faster for EigenWorms with UniCORNN). We hypothesize that this speed-up is implementation-dependent and would require further analysis to be conclusive.

### A.5.3 Hyperparameters

We adopted the hyperparameters of [29] without modification, as their experiments utilized the IMEX integrator, which, like our approach, derives from the LeapFrog integrator. The hyperparameters are: learning rate (lr), number of layers (#blocks), number of hidden neurons (hidden dim), state-space dimension (state dim), and whether the time dimension is sent as input (include time). These hyperparameters were found by grid search and are presented in Table 4.

We note that the implementation of the linear HSSM models is based on the code of [29] and uses complex numbers for implementation, which corresponds to doubling the state-space dimension mentioned in Table 4. We found that this dimensional doubling was necessary to recover the performance reported in the original paper, so we maintained this approach. We did not apply the same doubling of parameters for the nonlinear HSSM.

We reused the optimization scheme from [29], using the ADAM optimizer with default parameters and an early-stopping procedure based on the validation loss.

For the $RHEL$ algorithm, we have two additional hyperparameters: the nudging strength $\epsilon$ and the scaling factor $\gamma$ (see Appx. A.5.4). The nudging strength $\epsilon$ was set to $10^{-1}$ without prior tunning. For the scaling factor $\gamma$ we did a grid search over the values $\{10^0, 10^1, 10^2, 10^4\}$ for the regression task (PPG-DaLiA) and found that the best performing parameter was $10^4$. For the classification tasks, we performed a grid search over the values $\{10^0, 10^4, 10^8, 10^{12}\}$ and found that the best-performing scaling was $10^4$ based on the averaged score.

Table 4: Hyperparameters for the linear and nonlinear HSSM model

| Dataset | lr | hidden dim | state dim | #blocks | include time |
|---|---|---|---|---|---|
| Worms | 0.0001 | 64 | 16 | 2 | False |
| SCP1 | 0.0001 | 64 | 256 | 6 | False |
| SCP2 | 0.00001 | 64 | 256 | 6 | True |
| Ethanol | 0.00001 | 16 | 256 | 4 | False |
| Heartbeat | 0.00001 | 64 | 16 | 2 | True |
| Motor | 0.0001 | 16 | 256 | 6 | True |
| PPG | 0.0001 | 64 | 16 | 2 | True |

We employ different initialization schemes for linear and nonlinear HSSM variants. For the linear HSSM model, we follow the same initialization scheme as [29]:

- $A \sim \text{Uniform}(0, 1)$

- $B \sim \text{Uniform}\left(-\frac{1}{\text{hidden\_dim}}, \frac{1}{\text{hidden\_dim}}\right)$

- $C \sim \text{Uniform}\left(-\frac{1}{\text{state\_dim}}, \frac{1}{\text{state\_dim}}\right)$
- $D \sim \mathcal{N}(0,1)$
- $\delta \sim \text{Uniform}(0,1)$

For the nonlinear HSSM model, we adopt the initialization scheme from [30]:

- $A \sim \text{Uniform}(0.5,1)$
- $B \sim \text{Uniform}\left(-\frac{1}{\text{hidden\_dim}}, \frac{1}{\text{hidden\_dim}}\right)$
- $C = 0$
- $D \sim \mathcal{N}(0,1)$
- $b \sim \mathcal{N}(0,1)$
- $\alpha \sim \text{Uniform}(0.1, 1.0)$
- $\delta \sim \text{Uniform}(-1,1)$

### A.5.4 Gradient re-scaling for dealing with numerical instabilities

**Analysis.** From the theoretical results on RHEL, the nudging strength $\epsilon$ should be chosen as small as possible to accurately estimate the gradient of the loss function. However, naive numerical implementation can encounter underflow issues. In RHEL, gradient information is encoded in small perturbations to the state $\boldsymbol{\Phi}$ that generate the echo trajectory $\boldsymbol{\Phi}^e$. This presents numerical challenges when the perturbations and state values differ by several orders of magnitude.

In practice, when using finite precision representations (such as floating-point numbers), the perturbations may be lost due to discretization error, where small values cannot be accurately represented or distinguished from zero in finite precision arithmetic.

To address this numerical challenge, we employ a simple gradient rescaling method. We multiplicatively scale the output loss $\ell[\cdot,\cdot]$ by a constant $\gamma > 1$ during the echo dynamics computation. This amplification ensures that the perturbation-induced changes in the loss remain within the representable range of floating-point arithmetic. Subsequently, we divide the RHEL parameter gradient estimate $\left(\Delta_{\boldsymbol{\theta}}^{\text{RHEL}}(k, \epsilon)\right)$ by the same scaling factor $\gamma$ to recover the unbiased gradient.

We now demonstrate the effectiveness of this gradient rescaling method on both Linear and Nonlinear HSSM. To evaluate the effect of gradient scaling, we compare the gradients of RHEL and BPTT. Given a HSSM model, we randomly sample a tuple $(\boldsymbol{u}_i, \boldsymbol{y}_i) \sim \mathcal{D}$ from the SCP1 dataset (see App. A.5.1) and compute the gradients of the loss function with respect to the parameters of the recurrence of the HRUs in Fig. 5A for the linear HSSM and Fig. 6A for the nonlinear HSSM.

To gain a more fine-grained understanding, we also compute the gradients with respect to the inputs of the third layer of the HSSM (see Eq. 17 and 18) in Fig. 5B and Fig. 6B. Compared to the parameter gradients, these input gradients are not time-averaged and hence reveal more detail about the underflow issues.

**Gradient rescaling recovers the underflow issues.** As a general pattern across both architectures, we observe that the unscaled ($\gamma = 10^0$) parameter and input gradients tend to be biased: they have a norm ratio above 1.0 and cosine similarity below 1.0. We also note that for both the linear and nonlinear cases, there exists a scaling factor ($10^6$ for linear and $10^4$ for nonlinear) that recovers a nearly perfect match between RHEL and BPTT.

Additionally, for the input gradients that are not time-averaged (first column of Fig. 5B and Fig. 6B), there is a large amount of noise in the unscaled gradients, which is eliminated for the best-performing scaling factor ($10^6$ for linear and $10^4$ for nonlinear). We conjecture that this noise is due to the underflow effects described above.

**Linear HSSM: scaling improves gradient matching.** For the linear analysis, we observe a general pattern: the more we scale the RHEL gradient, the better it matches BPTT, both for input gradients and parameter gradients (Fig. 5A and B).

**Nonlinear HSSM: optimal scaling balances underflow and linearization errors.** For the nonlinear analysis, we also observe that scaling the RHEL gradient improves the match between RHEL and BPTT gradients. However, for higher values of the scaling factor ($\gamma = 10^6$), we observe that both input and parameter gradients show a drop in matching quality. We conjecture that this is due to linearization errors. If the loss gradient is too large, the nudging it drives will be large, and the finite difference method used for the learning rule will no longer be valid.

**Solution Implemented.** For the experiments of the main text, we conducted a grid search over the scaling parameter $\gamma$. In our initial implementation, we only applied gradient scaling without the corresponding downscaling step, effectively amplifying the gradient magnitude throughout training. This provided valuable insights into the sensitivity of RHEL dynamics to gradient scaling and improved performance. The complete rescaling procedure (including both upscaling and downscaling) used in the current section is also implemented in our code base to provide a more comprehensive analysis of the numerical stabilization approach.

### A.5.5 Vanishing and Exploding gradients of the Nonlinear HRU

In this section we validate empirically the claim made in section 4 that the Nonlinear HRU mitigate by design the problem of vanishing and exploding gradients. We emphasize that i) this claim has been demonstrated in a prior work [30], as such it is not one of ours, ii) this claim is a property of the model, not of the gradient computation algorithms, therefore it equally applies to BPTT and RHEL.

**Theoretical claims.** The gradient stability properties stem from the underlying Hamiltonian structure with symplectic integration. Prior work [30] rigorously demonstrated that the recurrent component of our nonlinear HRU block (corresponding to the Hamiltonian in Equation 18) possesses:

- Non-exploding gradients (Proposition 3.1 in [30]): the total loss gradient (i.e. aggregating loss "sensitivities" across all timesteps) is upper bounded.

- Non-vanishing gradients (Proposition 3.2 in [30]): when looking far enough back in time, the loss "sensitivities" (i.e the additive time-wise contributions to the loss gradient) are strictly non-zero and independent of the timestep.

**Experimental setup.** We randomly sampled $(u_i, y_i) \sim \mathcal{D}$ from the SCP1 dataset and computed time-dependent sensitivities for both RHEL ($\Delta_A^{\text{RHEL}}(k)$) and BPTT ($g_A(k)$) across the sequence length. We then performed linear regression analysis:

| Analysis | Correlation $r$ | Slope | Interpretation |
|----------|-----------------|-------|----------------|
| $\|\Delta_A^{\text{RHEL}}(k)\|$ vs $\|g_A(k)\|$ | 0.99998 | 1.0007 | RHEL $\approx$ BPTT gradients |
| $\|\Delta_A^{\text{RHEL}}(k)\|$ vs $k$ | 0.11068 | $2.82 \times 10^{-15}$ | No vanishing/exploding gradient |
| $\|g_A(k)\|$ vs $k$ | 0.11680 | $2.98 \times 10^{-15}$ | No vanishing/exploding gradient |

Table 5: Linear regression analysis between RHEL and BPTT gradient norms and timestep index $k$.

**Key findings.**

- **Row 1** confirms RHEL's gradient fidelity ($r \approx 1$, slope $\approx 1$).

- **Rows 2–3** demonstrate gradient stability: slopes $\approx 0$ indicate neither exponential growth (exploding) nor decay (vanishing) over time.

This empirical evidence supports our theoretical claims and demonstrates that RHEL on our proposed nonlinear HRU preserves the favorable gradient properties of the underlying Hamiltonian architecture. We note, however, that this analysis only holds at the level of a single HRU; vanishing gradients can still arise across layers in deeper architectures (i.e., within the feedforward transformations coupling successive HRUs). Both RHEL and BPTT are subject to these issues, as they compute mathematically equivalent gradients, and both benefit equally from standard architectural remedies such as skip connections (see Appendix C.4–C.5 in [30]).

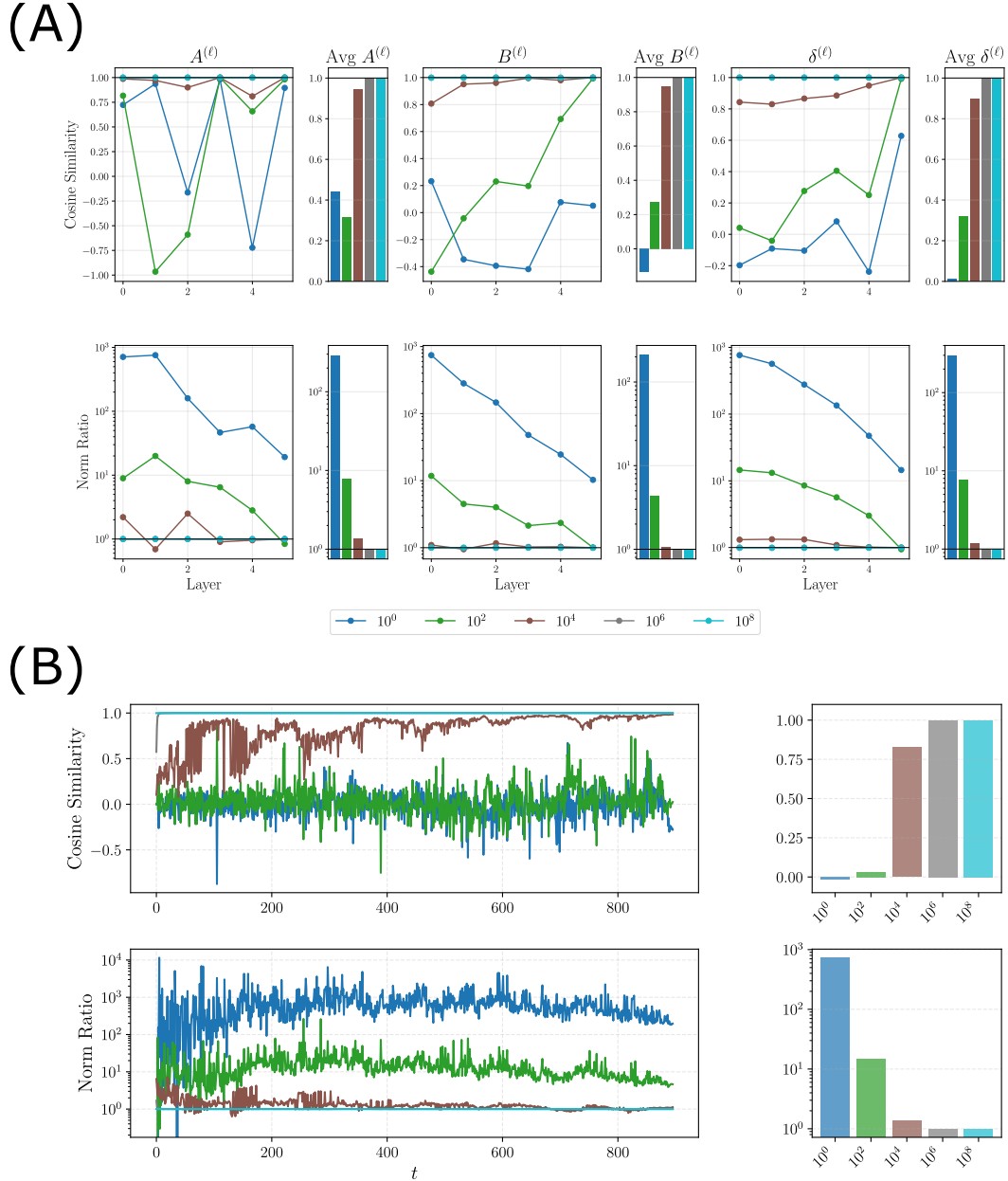

Figure 5: **(A): Parameters gradients comparison between RHEL and BPTT for HSSM.** Given some $(\boldsymbol{u}_i, \boldsymbol{y}_i) \sim \mathcal{D}$, we perform BPTT and RHEL on six blocks-deep linear HSSMs for different scaling factor $\gamma$ of RHEL (different colors). We measure, per layer (line plot) and when averaged across layers (bar-plot), the cosine similarity (top panels) and norm ratio (bottom panels) between RHEL and BPTT parameters gradients of a linear HSSM (Eq. (17)). **(B): Inputs gradient comparison between RHEL and BPTT for HSSM.** Same setting as (A) but we focus on the gradients with respect to the inputs of the third layer ($\boldsymbol{u}^{(3)}$, see Eq. 17). We measure, per time steps (line plot) and when averaged across time (bar-plot), the cosine similarity (top panels) and norm ratio (bottom panels) between RHEL and BPTT inputs gradients.

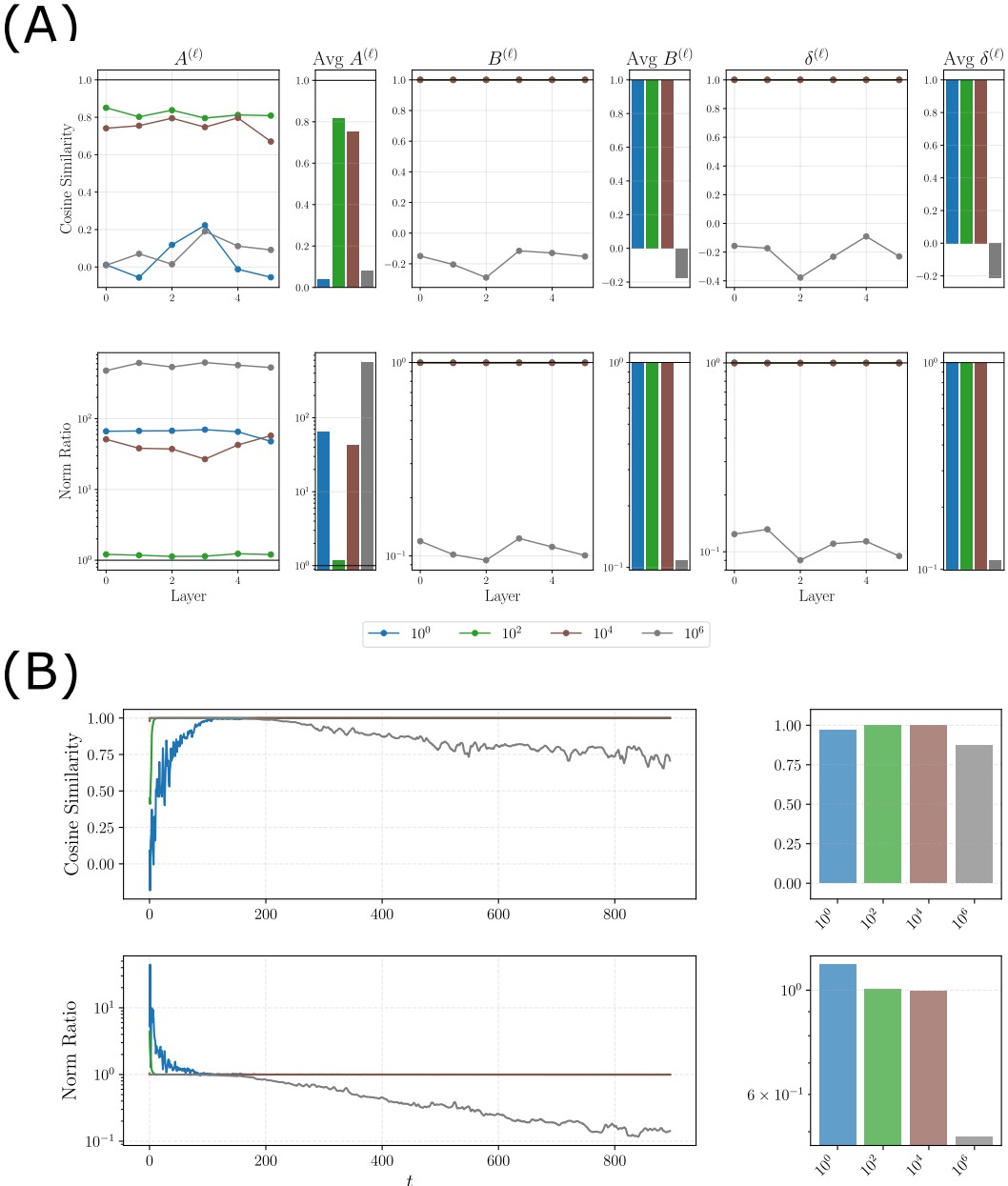

Figure 6: **(A): Parameters gradients comparison between RHEL and BPTT for HSSM.** Given some $(\boldsymbol{u}_i, \boldsymbol{y}_i) \sim \mathcal{D}$, we perform BPTT and RHEL on six blocks-deep nonlinear HSSMs for different scaling factor $\gamma$ of RHEL (different colors). We measure, per layer (line plot) and when averaged across layers (bar-plot), the cosine similarity (top panels) and norm ratio (bottom panels) between RHEL and BPTT parameters gradients of a nonlinear HSSM (Eq. (18)). **(B): Inputs gradient comparison between RHEL and BPTT for HSSM.** Same setting as (A) but we focus on the gradients with respect to the inputs of the third layer ($\boldsymbol{u}^{(3)}$, see Eq. 18). We measure, per time steps (line plot) and when averaged across time (bar-plot), the cosine similarity (top panels) and norm ratio (bottom panels) between RHEL and BPTT inputs gradients. For (A) and (B), $\gamma = 10^8$ was also tested but produced numerical instabilities leading to NaN values and is therefore omitted from the results.

