# OpenReview forum: "Learning long range dependencies through time reversal symmetry breaking"
_NeurIPS.cc/2025/Conference — NeurIPS 2025 oral_

### Official Review · Reviewer_k16k · 2025-07-01

**Clarity:** 3
**Significance:** 3
**Originality:** 3
**Rating:** 5
**Confidence:** 3

**Summary:**

This paper extends and generalizes Hamiltonian Echo Backprop (HEB) into the proposed Recurrent Hamiltonian Echo Learning (RHEL). The method is shown to be formally equivalent to the continuous-time adjoint state method (ASM) and Backpropagation Through Time (BPTT) in discrete time. The authors introduce Hamiltonian Recurrent Units (HRUs) and stack them into Hamiltonian State Space Models (HSSMs) to build deep sequence models that are compatible with RHEL.

**Questions:**

- Can you discuss the computational cost of running RHEL vs BPTT in a conventional setting?
- Is there a recommended heuristic for setting epsilon in practice?

**Ethical Concerns:**

["NO or VERY MINOR ethics concerns only"]

**Limitations:**

yes.

**Quality:**

4

**Strengths And Weaknesses:**

Strengths:
-This paper is well written with precise exposition of the RHEL method and supporting results.
-The experiments on long-range tasks demonstrate that RHEL matches the performance of BPTT while being useful for continuous learning or alternate hardware implementations, with potential memory and computation savings.
- The theoretical equivalences to both BPTT and ASM are well developed in the limit of small perturbations and supported with clear empirical results.

Weaknesses:
- As presented this method is only applicable to reversible Hamiltonian  systems which can be a restrictive class, and relies on symplectic
integration.
- The theoretical equivalence between RHEL and BPTT and ASM is  achieved as the perturbation strength, epsilon, goes to zero. However accurate gradient computations rely on epsilon > 0, particularly in lower precision, which may result in practical implementation being sensitive. It would be useful to see some consideration of continuity / similarities for a non zero perturbation.
-The claim of eliminating / reducing exploding and vanishing gradients would be strengthened by a supporting result. There is no
demonstrated improvement over BPTT (which is expected and fine) but could be the case when BPTT experience gradient maladies.
-Some of the phrasing in the introduction and conclusion, motivating this method through connections with hardware advancements, is a
little odd and does not seem to fully compliment the method begin presented.

Some of the language, particularly in sentence one of the abstract, could be toned down. Other examples: ""This paper takes an extreme view in the world of ai hardware...". The clarity score would be higher if these were addressed.

---

> ### Author Rebuttal · Authors · 2025-07-30
>
> We thank Reviewer k16k for their thoughtful evaluation. We appreciate their constructive feedback on several important aspects of RHEL. Below we address their specific concerns and provide detailed empirical analysis to strengthen our claims as well as **two proposals** to incorporate in our paper in case of acceptance.
>
> >As presented this method is only applicable to reversible Hamiltonian systems [...].
>
> **Answer 1.**
> While we fully acknowledge that extending RHEL to dissipative systems represents an important avenue for future work, *the restriction to Hamiltonian systems is less limiting than it may seem at first glance*.
>
> From a physical viewpoint, strictly reversible dynamics can be relaxed into *a suitable interplay between the time constants involved*. Namely: the timescale of dissipation within the self-learning device must be much larger than the timescale of interaction between the input and self-learning system. This timescale separation allows the system to behave *approximately* as a reversible Hamiltonian system during the computation window. As elaborated in the original HEB work [1], many existing physical systems can satisfy this condition: optical systems with Kerr nonlinearity, cold atom clouds in optical lattices, spin wave systems, and superconducting circuits. The key requirement is that dissipation timescales are much longer than the computation timescales, which is achievable in many well-isolated physical systems.
>
> Alternatively, we also investigate in details the possibility of extending RHEL to dissipative systems in our answer to Reviewer WcdN (Answer 1, Proposition 1).
>
> **Proposition 1.** We propose to incorporate the above discussion on the physical feasibility of reversibility constraints to implement RHEL on real systems.
>
> >The theoretical equivalence between RHEL and BPTT and ASM is achieved as [...] epsilon, goes to zero. However accurate gradient computations rely on epsilon > 0, particularly in lower precision [...] Is there a recommended heuristic for setting epsilon in practice?
>
> **Answer 2.** Both reviewers k16k and WcdN correctly identified that our theoretical results assume infinitesimally small nudging strength $\epsilon$, which requires careful consideration in finite precision implementations. We conducted extensive empirical analysis in **Appendix A.4.4 to inform our selection of $\epsilon$**, revealing a fundamental trade-off:
>
> *Numerical precision challenges*. In practice, gradient information in RHEL is encoded in small perturbations to the state $\Phi$ that generate the echo trajectory $\Phi^e$ (Theorem 3.1 of our submission). When perturbations and state values differ by several orders of magnitude, finite precision arithmetic suffers from *numerical underflow* issues whereby small perturbation values are flushed to zero. To address this, we employ a *gradient rescaling method* where we multiplicatively scale the output loss $\ell[\cdot,\cdot]$ by a constant $\gamma > 1$ during echo dynamics computation. Note that: i) it is equivalent to scaling the nudging strength, ii) gradient rescaling has become standard and integrated inside machine learning libraries. However, some subtleties arise with RHEL.
>
> *Accurate gradient estimation trade-offs*:
> - **Linear case**: We observed that increasing the scaling factor consistently improves gradient matching between RHEL and BPTT. This is because linear systems maintain linear response regardless of $\epsilon$ magnitude, so larger scaling only helps with numerical precision without introducing approximation errors. Note that vanilla backprop, and more generally the adjoint method, fall in this category, as error propagation is *linear* with respect to the adjoint variables.
> - **Nonlinear case**: There exists an optimal scaling factor that balances two competing error sources:
>   - *Underflow errors* ($\epsilon$ too small): Perturbations lost to finite precision
>   - *Linearization errors* ($\epsilon$ too large): RHEL uses finite difference approximation $\nabla f \approx [f(x + \epsilon) - f(x - \epsilon)]/(2\epsilon)$, which assumes linear response to perturbations. In nonlinear systems, large nudging causes the response to deviate significantly from linear behavior, making the finite difference approximation inaccurate. A solution to this (which we did not explore in our initial submission) would be to use *higher order* estimators, for instance: $\nabla f \approx (1/12\epsilon)[-f(x+2\epsilon) + 8 f(x + \epsilon) - 8 f(x - \epsilon) + f(x - 2\epsilon)]$
>
> *Implementation*: Through grid search, we found optimal scaling factors for both the Linear and Nonlinear case.
>
> This analysis demonstrates that while theoretical equivalence requires $\epsilon \to 0$, practical implementation with appropriate scaling maintains gradient accuracy while avoiding both numerical instabilities and linearization errors.
>
> > -The claim of eliminating / reducing exploding and vanishing gradients would be strengthened by a supporting result
>
> **Answer 3**.
> *Clarification of our claims*. We understand the need for such evidence and realize that our wording in the initial submission was not clear enough. When we wrote that "it has been shown that these HRUs mitigate by design vanishing and exploding gradients" (L.248-249) we meant that: i) this claim has been demonstrated in a *prior work* [2], as such it is not one of ours, ii) this claim is a property of the *model*, not of the gradient computation algorithms, therefore it *equally* applies to BPTT and RHEL. This being said, we appreciate the value of providing details about the theoretical grounding of these claims as well an empirical validation thereof for self-containedness of our work.
>
> *Theoretical guarantees*. The gradient stability properties stem from the underlying Hamiltonian structure with symplectic integration. Prior work [2] rigorously demonstrated that the recurrent component of our nonlinear HRU block (corresponding to the Hamiltonian in Equation 18) possesses:
> - **Non-exploding gradients** (Proposition 3.1 in [2]): the *total* loss gradient (*i.e.* aggregating loss "sensitivities" across all timesteps) is upper bounded.
> - **Non-vanishing gradients** (Proposition 3.2 in [2]): when looking far enough back in time, the loss "sensitivities" (*i.e* the additive time-wise contributions to the loss gradient) are strictly non-zero and independent of the timestep.
>
> *Empirical validation*: Since RHEL provides accurate BPTT gradient approximations (Figure 4 of our submission), it must also inherit these stability properties. To demonstrate this empirically, we analyzed gradient magnitudes over time for the recurrence matrix $A$ of the nonlinear HRUs.
>
> *Experimental setup*: We randomly sampled $(u_i, y_i) \sim \mathcal{D}$ from the SCP1 dataset and computed time-dependent sensitivities for both RHEL ($\Delta_A^{\text{RHEL}}(k)$) and BPTT ($g_A(k)$) across the sequence length. We then performed linear regression analysis:
>
>
> | Analysis | Correlation \(r\) | Slope | Interpretation |
> |-----------|----------------|-------|----------------|
> | $\|\Delta_A^{\text{RHEL}}(k)\|$ vs $\|g_A(k)\|$ | 0.99998 | 1.0007 | RHEL ≈ BPTT gradients |
> | $\|\Delta_A^{\text{RHEL}}(k)\|$ vs $k$ | 0.11068 | 2.82×10⁻¹⁵ | No vanishing/exploding gradient |
> | $\|g_A(k)\|$ vs $k$ | 0.11680 | 2.98×10⁻¹⁵ | No vanishing/exploding gradient |
>
> **Key findings**:
> - **Row 1** confirms RHEL's gradient fidelity (r ≈ 1, slope ≈ 1)
> - **Rows 2-3** demonstrate gradient stability: slopes ≈ 0 indicate neither exponential growth (exploding) nor decay (vanishing) over time.
>
> This empirical evidence supports our theoretical claims, those of [2] and demonstrates that RHEL maintains the beneficial gradient properties of the underlying Hamiltonian architecture.
>
> > There is no demonstrated improvement over BPTT (which is expected and fine) but could be the case when BPTT experience gradient maladies.
>
> This is an interesting suggestion. However, we do not expect RHEL to improve over BPTT regarding gradient pathologies, as these are inherent properties *of the model architecture* rather than the learning algorithm as mentioned earlier. At the single HRU level, both methods inherit well-behaved gradients (non-exploding/non-vanishing) from the Hamiltonian dynamics. While vanishing gradients can occur *across* layers in deep architectures (i.e. inside feedforward transformations binding HRUs), both RHEL and BPTT face identical challenges since they compute mathematically equivalent gradients. Both methods benefit equally from standard architectural solutions like skip connections -- see Appendix C.4-C.5 in [2].
>
> **Proposion 2.** We propose to add the table presented in answer 2 to validate the non-vanishing/exploding gradient properties of the nonlinear HRU.
>
> > Some of the phrasing in the introduction and conclusion [...] is a little odd [...] Some of the language, particularly in sentence one of the abstract, could be toned down.
>
> We fully acknowledge this point. As phrased in details in our answer to Reviewer AkMK (Answers 2-3, Propositions 2-3), we will eliminate these sentences and replace them with a more detailed background on physical computing and contrast this literature with standard digital design with a softened tone. We will also include detailed discussions about possible physical implementations of RHEL (Reviewer UVRA, Answer 1, Proposition 2) as well as a more physically realistic version of RHEL which does not require the exact knowledge of the Hamiltonian to perform the weight update (Reviewer WcdN, Answer 3, Proposition 2).
>
> > Can you discuss the computational cost of running RHEL vs BPTT in a conventional setting?
>
> We carry out this comparison in details in our answer to Reviewer AkMK (Answer 1, Proposition 1).
>
> [1] Lopez-Pastor & Marquardt. "Self-learning machines based on Hamiltonian echo backpropagation. 2023.
>
> [2] Rusch & Mishra. Unicornn: A recurrent model for learning very long time dependencies. 2021.

---

### Official Review · Reviewer_WcdN · 2025-07-03

**Clarity:** 4
**Significance:** 3
**Originality:** 4
**Rating:** 6
**Confidence:** 4

**Summary:**

This paper introduces Recurrent Hamiltonian Echo Learning (RHEL), a novel and elegant algorithm for training a class of recurrent models known as Hamiltonian State Space Models (HSSMs). The core idea is to leverage the physical principle of time-reversal symmetry, which is inherent to conservative Hamiltonian systems. Instead of using backpropagation, RHEL computes gradients through a clever, physics-inspired procedure: it runs the model forward, applies a small "nudge" to the final state in the direction of the loss gradient, and then lets the system evolve backward in time (an "echo"). The deviation of this echo trajectory from the true time-reversed path provides a direct, unbiased estimate of the loss gradients.

The authors make several key contributions:
1.  They formalize RHEL first in continuous time, proving its equivalence to the continuous adjoint state method.
2.  They derive a practical discrete-time version for a specific class of models they call Hamiltonian Recurrent Units (HRUs), proving its equivalence to Backpropagation Through Time (BPTT).
3.  They generalize the algorithm to deep, stacked models (HSSMs) via a procedure called RHEL chaining.
4.  They provide compelling empirical evidence that RHEL perfectly matches the gradients computed by BPTT and achieves equivalent training performance on challenging long-range sequence modeling tasks.

Ultimately, the work provides a powerful new perspective on credit assignment, framing it as a physical process and paving the way for the design of novel, energy-efficient "self-learning" physical hardware.

**Questions:**

1.  **Extending to Dissipative Systems:** The restriction to conservative Hamiltonian systems is the most significant limitation. Could the authors elaborate on the fundamental challenges of extending this framework to dissipative systems where time-reversal symmetry is broken? Does the echo mechanism break down completely, or could it be adapted, perhaps by having the echo system model the dissipation in reverse (i.e., by actively pumping energy back into the system)?
2.  **Sensitivity to the Nudging Parameter `ε`:** The algorithm introduces a new hyperparameter, the nudging strength `ε`. Could you provide more intuition on its role and sensitivity? How was it selected in your experiments? Is there a principled way to set it, perhaps by adapting it based on the magnitude of the loss or the learning rate?
3.  **Path to a "True" Physical Implementation:** The vision of a self-learning physical system is compelling. However, Algorithm 1 still requires explicitly computing the gradient of the Hamiltonian with respect to the parameters (`∇_θ H`), which may be non-trivial for a physical system to compute. Could you expand on the feasibility of achieving a truly "black-box" physical implementation, perhaps by elaborating on the "homeostatic control" idea mentioned in the discussion? What would this look like in practice?

**Ethical Concerns:**

["NO or VERY MINOR ethics concerns only"]

**Final Justification:**

I’m maintaining my Strong Accept (6) rating. The authors have done an outstanding job addressing all of my questions, and this is by far the best rebuttal response I’ve seen.

**Limitations:**

Yes. The authors have done an excellent job of discussing the limitations of their work in the discussion section. They are upfront about the restriction to non-dissipative systems, the offline nature of the algorithm, the computational cost on digital hardware, and the potential bias from the finite nudging strength. Their discussion is honest, thorough, and points clearly to important directions for future research.

**Paper Formatting Concerns:**

No formatting issues were identified.

**Quality:**

4

**Strengths And Weaknesses:**

**Strengths:**

1.  **Exceptional Originality and Significance:** The paper's central contribution is not an incremental improvement but a paradigm-shifting idea. By elegantly connecting the mathematical formalism of backpropagation to the physical principle of time-reversal symmetry, it opens a genuinely new avenue for research in AI. The ultimate vision of building physical systems that learn *in-situ* using their own dynamics is profound and could have a major long-term impact on the field of energy-efficient AI and neuromorphic computing.

2.  **High Technical Quality and Rigor:** The paper is built on a solid theoretical foundation. The authors provide clear, formal theorems (Theorems 3.1, 3.2, 3.3) that establish the equivalence of RHEL with standard gradient-based methods (Adjoint State Method and BPTT). The derivation is logically sound, progressing from the continuous-time idealization to a practical, discrete-time algorithm and finally to deep hierarchical models.
3.  **Outstanding Clarity and Presentation:** The paper is exceptionally well-written and easy to follow. The core concept of RHEL is introduced with clear intuition, supported by excellent illustrations (Fig. 1 and 2). The logical flow from problem statement to continuous theory, discrete algorithm, and stacked models makes the complex material highly accessible. The authors do a great job of situating their work with respect to prior art like HEB while clearly articulating their novel contributions.

**Weaknesses:**

(These are minor and mostly represent avenues for future work, which the authors acknowledge.)

1.  **Limited Model Class:** The primary limitation is that RHEL is applicable only to conservative Hamiltonian systems that exhibit time-reversal symmetry. This is a restrictive subset of dynamical systems, and many real-world processes (and potentially more expressive models) are dissipative. While the authors suggest workarounds, extending the core theory to non-conservative systems remains an open challenge.

2.  **Practicality on Conventional Hardware:** On current digital hardware (GPUs/TPUs), RHEL requires three "forward-like" passes (one free evolution and two parallel echo passes) to compute gradients. This may not offer a significant speed or memory advantage over a standard BPTT implementation (one forward, one backward pass). The true promise of the algorithm lies in its potential implementation on novel analog or physical hardware, a point the authors rightly emphasize.

3.  **Offline Nature:** The current formulation of RHEL is offline, as it requires processing the entire sequence forward before initiating the backward-in-time echo pass. This makes it unsuitable for online or real-time learning scenarios.

---

> ### Author Rebuttal · Authors · 2025-07-30
>
> We thank Reviewer WcdN for the time they spent to carefully review our submission, raise important questions and we are very happy that they liked our paper. We make **two propositions** of modifications of our paper in case of acceptance along the lines suggested by Reviewer WcdN.
>
> > Extending to Dissipative Systems: [...] fundamental challenges of extending this framework to dissipative systems where time-reversal symmetry is broken? Does the echo mechanism break down completely, or could it be adapted, perhaps by having the echo system model the dissipation in reverse [...]?
>
> **Answer 1.**
> This is an excellent question! To figure it out, let us model dissipation in the Hamiltonian formalism and follow the exact same steps as in our submission.
>
> - **Modelling dissipation in the Hamiltonian formalism.** One common way to model dissipation in the Hamiltonian formalism is to modify Eq. 1 of into [1]:
>
> $$
>    \forall t \in [-T, 0]: \ \partial_t \Phi(t) = (J -R)\cdot \nabla_{\Phi} H[\Phi(t), \theta, u(t)],
> $$
>
> where $R$ must be *symmetric* and *positive definite*. For simplicity of the derivation, we choose $R=I_\Phi$ (the identity matrix).
>
> - **Time-reversed dynamics.** With this new model, the time-reversed dynamics, *i.e.* the dynamics of $\widetilde{\Phi}^\star(t):=\Sigma_z\cdot\Phi(-t)$ read:
>
> $$
>     \partial_t \widetilde{\Phi}^\star(t) =(J + I)\cdot  \nabla_{\Phi^\star}H[\widetilde{\Phi}^\star(t), \theta, u(-t)]
> $$
>
> As the reviewer rightfully noted: time-reversal symmetry is broken since the dissipation ($-R\cdot \nabla_{\Phi} H$) yields, in *reverse*, "pumping energy back into the system"($+R\cdot \nabla_{\Phi^\star} H$). Yet, one could wonder whether RHEL could be extended such that deviations from this time-reversed trajectory, *even if it does not match the forward-time trajectory*, capture the error signals carried by the continuous Adjoint State Method (ASM).
>
> - **ASM with dissipative dynamics**. To answer this question, we must first derive the ASM on the dissipative dynamics introduced above which yields, with $\lambda(0)=0$:
>
> $$
> \partial_t \lambda(t) = \nabla^2_{\Phi}H[\Phi(-t), \theta, u(-t)] \cdot J^\top \cdot \lambda(t) + \nabla_{\Phi}\ell[-t, \Phi(-t), \theta]
> -\nabla^2_{\Phi}H[\Phi(-t), \theta, u(-t)] \cdot \lambda(t)
> $$
>
> Dissipation introduces the new last term in the above equation. We will therefore pay particular attention to this term in the last step.
>
> - **Echo dynamics with pumping.** As suggested by the reviewer, we now define the echo dynamics as, taking again $\Phi^e(0) = \Phi^\star(0)$:
>
> $$
>     \partial_t \Phi^e(t) =(J + I)\cdot  \nabla_{\Phi^e}H[\Phi^e(t), \theta, u(-t)] - \epsilon J \cdot \nabla_{\Phi^e}\ell[-t, \Phi^e(t)],
> $$
>
> where we actively pump energy back into the system. These echo dynamics are designed such that when $\epsilon=0$, we recover the time-reversed dynamics derived above. Proceeding exactly as we did in the proof of Theorem 3.1 (p. 27 in the Appendix) the derivative of the above echo dynamics with respect to $\epsilon$ evaluated at $\epsilon=0$ yields the following equation, denoting $\Delta_\Phi(t) := \Sigma_x \cdot \partial_\epsilon(\Phi(t,\epsilon))|_{\epsilon=0}$:
>
> $$
> \partial_t \Delta_\Phi(t)= \nabla^2_{\Phi}H[\Phi(-t), \theta, u(-t)] \cdot J^\top \cdot \Delta_\Phi(t) + \nabla_{\Phi}\ell[-t, \Phi(-t), \theta]
> $$
>
> $$
> -J\cdot\nabla^2_{\Phi}H[\Phi(-t), \theta, u(-t)] \cdot J \cdot \Delta_\Phi(t)
> $$
>
> Unfortunately, we observe that $\Delta_\Phi(t)$ no longer satisfies the adjoint dynamics because the last term of the above equation ($-J\cdot\nabla^2_{\Phi}H \cdot J \cdot \Delta_\Phi$) does not match the last term of the adjoint dynamics ($-\nabla^2_{\Phi}H \cdot \lambda$).
>
> **Conclusion.** Using our methodology based on the equivalence with the continuous ASM, the echo dynamics cannot be seamlessly adapted to the case where the system is subject to dissipation, even when "pumping energy back into the system". However, this does *not* mean that developing forward-only, perturbation-based methods for temporal credit assignment is impossible in general when systems exhibit dissipation. In a very recent work released shortly after our submission, it was shown that a class of *linear* dissipative systems could be described with a Lagrangian formalism (which is closely related to our Hamiltonian formalism) and gradients estimated as finite differences of two *dissipative* trajectories, provided that the system is subject to periodic boundary conditions [2].
>
>
> **Proposition 1**. We propose to mention this important discussion in the main of the paper, and elaborate in greater details with these equations in Appendix.
>
>
> >Sensitivity to the Nudging Parameter ε: [...] Is there a principled way to set it [...]?
>
> **Answer 2.** As reviewer k16k raised the same question, we refer to our comprehensive response in section k16k (Answer 2).
>
>
> >Path to a "True" Physical Implementation [...] Algorithm 1 still requires explicitly computing the gradient of the Hamiltonian with respect to the parameters (∇_θ H) [...] Could you expand on the feasibility of [...] the "homeostatic control" idea mentioned in the discussion?
>
> **Answer 3.**
> This is also a very important question. We elaborate below how an *agnostic* version of RHEL ("aRHEL") would look like in practice by closely following the methodology proposed in the *agnostic Equilibrium Propagation* paper [3]. While we sketch these ideas in the rebuttal to provide some intuitions, *we acknowledge upfront that they would require a more rigorous treatment in a separate paper*.
>
> - **Definition of the joint neuron and parameter dynamics**. We define the *joint* dynamics of $\theta$ and $\Phi$ over $t\in[-T, 0]$ as:
>
> $$
> \partial_t \Phi(t) = J\cdot \nabla_{\Phi} H[\Phi(t), \theta(t), u(t)],
> $$
> $$
>    \partial_t \theta(t) = - \alpha  \nabla_{\theta}\left[H[\Phi(t), \theta(t), u(t)] + \frac{1}{2} \|c(t) - \theta(t)\|^2\right]
> $$
>
> where $c(t)$ is a *control* variable acting on the parameters and $\alpha$ a learning rate. Note that: i) the dynamics of $\theta$ are purely dissipative, ii) we do not model the *momentum* of $\theta$ but only its position. We acknowledge that this modeling choice is far from obvious and could be subject to further discussions, but we adopt this one as this is *the most simple choice* to readily adapt the agnostic mechanism from [3] to RHEL.
>
> - **First phase of the agnostic procedure.** Assuming that $\theta(-T):=\theta_\star$, we adjust the control variable $c$ during the free phase such that $\theta$ remains at $\theta_\star$ - for instance using a simple proportional controller as in [3]. Namely, $\forall t \in [-T, 0]$:
>
> $$
>  \partial_t \Phi(t) = J\cdot \nabla_{\Phi} H[\Phi(t), \theta(t), u(t)],
> $$
>
> $$
>    \partial_t \theta(t) = - \alpha  \nabla_{\theta}\left[H[\Phi(t), \theta(t), u(t)] + \frac{1}{2} \|c(t) - \theta(t)\|^2\right] \approx 0 \\
> $$
>
> $$
>   \Rightarrow \forall t \in [-T, 0]: \ c(t)\approx \theta_\star + \nabla_{\theta}H[\Phi(t), \theta_\star, u(t)]
> $$
>
> We denote $\{c_\star(t)\}_{t\in[-T, 0]}$ the resulting control trajectory.
>
> - **Second phase of the agnostic procedure.** During the echo phase, the control trajectory $c_\star$ is played in reverse, yield the following dynamics $\forall t \in [0, T]$:
>
> $$
> \partial_t \Phi^e(t) = J\cdot \nabla_{\Phi^e} H[\Phi^e(t), \theta^e(t), u(-t)] - \epsilon J\nabla_{\Phi^e}\ell[-t, \Phi^e(t)], \\
> $$
>
> $$
> \partial_t \theta^e(t) = - \alpha  \nabla_{\theta}\left[H[\Phi^e(t), \theta^e(t), u(-t)] + \frac{1}{2} \|c_\star(-t) - \theta^e(t)\|^2\right]
> $$
>
> Taking a closer look at the dynamics satisfied by $\theta^e$ and using the definition of $c_0$ above:
>
> $$
> \partial_t \theta^e(t) = -\alpha \left[\nabla_{\theta}H[\Phi^e(t), \theta^e(t), u(-t)] - c_\star(-t) + \theta^e(t)\right]
> $$
>
> $$
> \approx -\alpha \left[\nabla_{\theta}H[\Phi^e(t), \theta^e(t), u(-t)] - \nabla_{\theta}H[\Phi(-t), \theta_\star, u(-t)] - \theta_\star + \theta^e(t)\right]
> $$
>
> $$
> = \alpha \epsilon \Delta_{\theta}^{\rm aRHEL}(t, \epsilon, \alpha) - \alpha \nabla_{\theta}\left[\frac{1}{2} \|\theta^e(t) - \theta_\star \|^2\right]
> $$
>
> with $\Delta_{\theta}^{\rm aRHEL}(t, \epsilon, \alpha):=\epsilon^{-1}(\nabla_{\theta}H[\Phi^e(t), \theta^e(t), u(-t)] - \nabla_{\theta}H[\Phi(-t), \theta_\star, u(-t)])$. Note that $\Delta_{\theta}^{\rm aRHEL}$ would recover $\Delta_{\theta}^{\rm RHEL}:=\epsilon^{-1}(\nabla_{\theta}H[\Phi^e(t), \theta_\star, u(-t)] - \nabla_{\theta}H[\Phi(-t), \theta_\star, u(-t)])$ if $\theta^e(t)$ was "close enough" to $\theta_\star$, which the second term of the above dynamics naturally biases towards.
> Therefore, the proposed agnostic mechanism approximates the desired weight update $\Delta \theta^{\rm RHEL} = \lim_{\epsilon, \alpha \to 0}\int_0^T \Delta_{\theta}^{\rm aRHEL}(t, \epsilon, \alpha)$, without requiring the exact knowledge of $H$ to compute the learning rule.
>
> **Proposition 2**. We propose to include this analysis in the appendix of the paper.
>
> [1] Smith et al. "Learning dissipative Hamiltonian dynamics with reproducing kernel Hilbert spaces and random Fourier features." 2024.
>
> [2] Berneman et al. "Equilibrium Propagation for Periodic Dynamics." 2025.
>
> [3] Scellier et al. "Agnostic physics-driven deep learning." 2022.

---

> > ### Comment · Reviewer_WcdN · 2025-08-06
> > **Maintaining Strong Accept (6): Outstanding Rebuttal Response**
> >
> > I’m maintaining my Strong Accept (6) rating. The authors have done an outstanding job addressing all of my questions, and this is by far the best rebuttal response I’ve seen.

---

### Official Review · Reviewer_UVRA · 2025-07-03

**Clarity:** 4
**Significance:** 4
**Originality:** 3
**Rating:** 5
**Confidence:** 3

**Summary:**

This paper proposes a novel algorithm for learning continuous-time sequence models on non-dissipative physical device (Hamiltonian systems). In addition, it introduces a discrete-time version called Hamiltonian Recurrent Units (HRUs), providing a means for simulation. Furthermore, by proposing Hamiltonian State Space Models (HSSMs), which scale and extend HRUs in a hierarchical manner, the paper demonstrates that the proposed model can be expanded into sequence models such as State Space Models (SSMs), which have recently gained attention in ML. It also shows that HSSMs can learn very long sequence tasks through simulation.

**Questions:**

- In line 173, for the time derivative of $\pi_1$ and $\pi_4$ in Equation 8, it would be helpful to add explanations about $u$ (external force) and $y$ (target). As shown in Figure 2B, explicitly specifying the external force ($u$) and target ($y$) for the object would make it easier to understand.
- In line 48, the text mentions three conditions: i) use “forward passes” only (i.e., no backward passes), ii) do not use explicit state-Jacobian (i.e., Jacobian of the system’s dynamics). However, only two conditions are described earlier. This seems to be a typo.
- Regarding HSSMs with multiple layers, it is stated that activation patterns need to be stored to compute gradients. Does this mean physical implementation is impossible? Or is this limitation only applicable to the discretized time model, while the continuous-time model does not face such physical constraints?
- The title, "Learning long-range dependencies through time reversal symmetry breaking," seems misleading as there is no mention of time-reversal symmetry breaking in the main text. This could confuse readers. According to Lopez-Pastor & Marquardt (PRX 2023), "*The self-learning device obeys time-reversal symmetry.*" The title appears contradictory. Was the authors' intention to highlight that the system being studied has time-reversal symmetry, and that external driving forces applied when starting the backward trajectory create symmetry breaking?

**Ethical Concerns:**

["NO or VERY MINOR ethics concerns only"]

**Limitations:**

Yes

**Quality:**

4

**Strengths And Weaknesses:**

**Strengths**
- The paper is very clearly written and rigourous proof for each theorem.
- The paper also compare their algorithm with the previous algorithm in detail and address the important limitation of the previous algorithm.
- This paper is an excellent contribution as it develops a real on-chip self-learning system that reduces the high energy cost of GPUs while demonstrating scalability, such as with deep SSMs.

---

> ### Author Rebuttal · Authors · 2025-07-29
>
> We appreciate the reviewer's time for providing a thorough evaluation of our work, making constructive suggestions and spotting typos. In the following, we respond to their specific points and outline **three proposed modifications** to strengthen the paper in case of acceptance.
>
> > In line 173, for the time derivative of $\pi_1$
>  and $\pi_4$
>  in Equation 8, it would be helpful to add explanations about
>  $u$ (external force) and
>  $y$(target). As shown in Figure 2B, explicitly specifying the external force ($u$) and target ($y$) for the object would make it easier to understand.
>
> **Answer 1**. Thank you for this valuable suggestion to improve readability and strengthen the connection between the mathematical formulation and Figure 2B. In the main text we indeed only presented the Hamiltonian and the dynamics of the raw physical system without its coupling with the input and nudging output, which makes it difficult to understand how the external force and target relate to the visual representation. We will present a condensed version of the full equations from the appendix (Equations 24-25) to make this connection explicit.
>
>
> **Hamiltonian with input**:
>
> $$
> H[\Phi, \theta, u] = \sum_i^6\frac{\pi_i^2}{2m_i} + \frac{1}{2}\sum_{i}^6k_i \phi_i^2 + \frac{1}{2}\sum_{i}^6\sum_{j,j>i}^6k_{ij} (\phi_j-\phi_i)^2 + u\phi^1
> $$
>
> **Complete Dynamics Equations:**
>
> $$
> \partial_t\phi_i = \pi_i /m_i, \quad \partial_t \pi_i = -k_i \phi_i + \sum_i^6 \sum_{j, j\neq i}^6k_{ij}(\phi_j-\phi_i) - \delta_{i1}\mathbf{u}\phi^1 + \delta_{i4}\delta_e\epsilon(\phi^4-y)
> $$
>
> where $\mathbf{u}$ is the external driving force applied to oscillator 1, $y$ is the target trajectory, $\delta_e$ is the indicator function of the echo pass (equal to 1 during the echo pass and 0 otherwise), and $\delta_{ij}$ is the Kronecker delta.
>
> **Proposition 1.** We propose to replace the equations of the toy example by the equation mentioned in answer 1 to mention the coupling with input/output of the system.
>
> > Regarding HSSMs with multiple layers, it is stated that activation patterns need to be stored to compute gradients. Does this mean physical implementation is impossible? Or is this limitation only applicable to the discretized time model, while the continuous-time model does not face such physical constraints?
>
> **Answer 2.**
> This is an excellent question which raises several points:
>
> - this limitation is *solely* tied to the choice of the **Hamiltonian State Space Model** (HSSM) architecture introduced in Eq.16 of our submission. Whether HSSMs are simulated in discrete time *or* in continuous time, or physically implemented (see next bullet), they would *always* require to store (or somehow recompute) the input sequences that are fed into each HSSM in order to estimate gradients using our RHEL chaining algorithm.
> - Whether HSSMs are amenable to a physical implementation, regardless of the above limitation, is an interesting and *independent* question of its own. We could imagine a physical implementation of a HSSM as follows. As the forward pass of a HSSM reads a composition of recurrent and feedforward units, the corresponding "RHEL chaining" algorithm (Alg. 1 inside section 3.3 of our submission) reads as an *implicit* backward chaining of backprop and RHEL respectively. Namely: error signals are backpropagated through feedforward modules and "RHEL-propagated" through recurrent modules -- see Remark 8, p. 40 inside our appendix where we highlight this aspect. Therefore, an intuitive implementation of a HSSM would be to map feedforward parts onto *digital circuits* and recurrent parts onto *analog circuits*, with backprop and RHEL running on these digital and analog circuits respectively. This mapping is strongly reminiscent of the recently proposed backprop-Equilibrium Propagation hybridization [1]. Yet, this type of hybrid approach potentially comes at the cost of expensive ADC-DAC conversions across layers [2].
> - While we used HSSMs to provide of scalability proof-of-concept of RHEL and leverage most recent SSM architectures, there exists an alternative architecture which may not exhibit this limitation and may be more amenable to a physical implementation. If HSSMs can be regarded as model of hybrid systems comprising digital and analog parts being composed hierarchically, a model corresponding to a **fully analog** Hamiltonian system could be defined as a sum-separable Hamiltonian of the form:
>
> $$
> H = \sum_{\ell = 1}^L \frac{1}{2} \|\pi^{(l)}\|^2 + \frac{1}{2} \phi^{(\ell)^\top}\cdot A^{(\ell)} \cdot \phi^{(\ell)} - \phi^{(\ell)^\top}\cdot B^{(\ell)} \cdot \phi^{(\ell - 1)}, \quad \phi^0:=u
> $$
>
>  - Applying Hamiltonian dynamics (Eq. 1 in our submission) with this choice of Hamiltonian yields:
>  $$
>  \dot{\phi}^{(\ell)}= \pi^{(\ell)}, \quad \dot{\pi}^{(\ell)}=-A^{(\ell)}\cdot \phi^{(\ell)} + B^{(\ell)}\cdot\phi^{(\ell - 1)} + B^{(\ell + 1)^\top}\cdot \phi^{(\ell + 1)}
>  $$
>
>  - Note that the resulting equations are very different from that of a HSSM: instead executing $L$ trajectories hierarchically, a *single* trajectory updating all layers *simultaneously* at each time step is executed. As such, the only sequence which would need to be stored is just the *data sequence* $\phi^{(0)}:=u$ itself.
> - Note also we could seamlessly introduce nonlinearities inside the proposed Hamiltonian above in the same way as it is done in the nonlinear HRU introduced in our submission (Eq. 19 of our submission).
>
> **Proposition 2.** We propose to add a detailed discussion around physical implementations of these various Hamiltonian models and RHEL in a dedicated paragraph.
>
> > The title [...] Was the authors' intention to highlight that the system being studied has time-reversal symmetry, and that external driving forces applied when starting the backward trajectory create symmetry breaking?
>
> Yes, this was *exactly* our intention. And this is also consistent with the requirement that the system must obey time-reversal symmetry, as highlighted in the paragraph "Assumptions" of our submission (starting at L.123), and therefore also consistent with the seminal HEB work [3]. This being said, we acknowledge that the term "symmetry breaking" is never mentioned again in the rest of the paper which may hinder clarity.
>
> **Proposition 3.** We propose to make an explicit connection to "symmetry breaking" when introducing the RHEL algorithm.
>
>
> [1] Nest et al. "Towards training digitally-tied analog blocks via hybrid gradient computation." 2024.
>
> [2] Li et al. "Merging the interface: Power, area and accuracy
> co-optimization for rram crossbar-based mixed-signal computing system". 2015.
>
> [3] Lopez-Pastor & Marquardt. "Self-learning machines based on Hamiltonian echo backpropagation". 2023.

---

### Official Review · Reviewer_AkMK · 2025-07-03

**Clarity:** 2
**Significance:** 3
**Originality:** 3
**Rating:** 5
**Confidence:** 3

**Summary:**

This paper proposes Recurrent Hamiltonian Echo Learning (RHEL), a forward-only gradient-estimation algorithm for non-dissipative Hamiltonian systems. RHEL works by perturbing a time-reversed copy of the forward trajectory with a small ``nudge'', and shows that the resulting finite differences converge to the continuous adjoint-state gradients. A discrete variant is derived, and this paper shows its equivalence with BPTT. Empirical validation occurs on six UEA long-sequence classification datasets and the PPG-DaLiA regression task. HRUs trained with RHEL match BPTT (although mean performance is a little lower, this is typically not statistically significant).

**Questions:**

- Why is it surprising/need to be posited that "some SSMs could inherently be mapped onto dynamical systems"?
- Theorem 3.1, if you take $\epsilon$ to 0, wouldn't both methods give the echo trajectory simply be the forward trajectory in reverse for both cases?

Minor:
- Page 2, Line 48: three -> two? I think I only saw two conditions.
- Page 3, Line 127: let -> left
- Page 6, Line 219: rewrite -> can be rewritten
- Page 7, Line 221: spurious "reading"?
- Page 9, Line 296: Remove "Yet"
- It is a bit misleading to bold font all the averaged numbers in Table 1.
- Recommend to use hyperref so that the refs and cross-refs are clickable.

**Ethical Concerns:**

["NO or VERY MINOR ethics concerns only"]

**Final Justification:**

The authors clarified some misunderstandings and provided additional interesting analysis. The suggested changes would strengthen the paper and I am raising my score from 4 to 5.

**Limitations:**

Yes, the authors are upfront about the limitations of the method and that it slightly underperforms BPTT.

**Quality:**

3

**Strengths And Weaknesses:**

Strengths:
- The method proposed offers a hardware-friendly alternative to BPTT and only uses three forward passes,
- Theorems are also supported with proofs (some sketched in text, others in Appendix).
- The diagrams in the paper are good, very useful for illustrating the method.

Weaknesses:
- There are no results/values of runtime, and this reduces the selling point of the paper being "forward-only".
- The writing is not very clear at times and sometimes seem a bit overdramatic. For example, "the physical realization of our algorithm" in the abstract is not very clear. and "this paper embraces an extreme view" is quite dramatic.
- The structure and value of the Supplementary materials need to be made clearer. The informal versions of the theorems are stated in the paper and the formal versions are not signposted.
- Many systems in reality are not conservative, which may reduce the impact.

---

> ### Author Rebuttal · Authors · 2025-07-29
>
> We thank the reviewer for their valuable time to give constructive feedback and spot typos. We address below the specific concerns they raised and accordingly provide **four propositions** of modifications of the paper in case of acceptance.
>
> > There are no results/values of runtime [...].
>
> **Answer 1**.
> We do not claim any superiority of RHEL over BPTT on conventional *digital* hardware in terms of run time or memory consumption. As mentioned in the "Future work" paragraph L.306-308 of our submission: turning RHEL into a compelling alternative to BPTT on digital accelerators with respect to these metrics is a direction of research of its own.
>
> In its current form, **RHEL has the *same* memory and computational cost as BPTT**:
>
> - on a **single Hamiltonian Recurrent Unit** (HRU) using $T$ timesteps and with a hidden layer dimension $d$: when using "naive" recurrent computation, the time complexity of BPTT and RHEL is $O(Td^2)$, *i.e.* $d^2$ MACs per step. For *linear* HRUs which allow for the use of parallel scan [1], this cost can be reduced to $O(log(T)d^2)$. In both cases, the memory complexity of BPTT and RHEL is $O(Td)$ when naively storing activations of the forward trajectory, and $O(d)$ when re-computing them backwards from the final state leveraging the time-reversibility of the symplectic integrator at use inside HRUs. As mentioned L.150-152 of our submission, this memory efficiency is a *model* feature that equally applies to BPTT and RHEL, not a key differentiator of RHEL.
>
> - On a **Hamiltonian State Space Model** (HSSM): the above computational cost scales linearly with the number of HRUs for both BPTT and RHEL. For the memory cost on a given HRU taken inside a HSSM, there is an extra memory cost of $O(Td)$ for both BPTT and RHEL to store the input sequence which is fed into a given HRU.
>
> This being said, we acknowledge the value of providing these metrics in general. Below are the runtime and memory comparisons:
>
> | Dataset | Model | RHEL Time (s) | BPTT Time (s) | RHEL Mem (GB) | BPTT Mem (GB) |
> |---------|-------|---------------|---------------|---------------|---------------|
> | PPG | Lin | 11.50±4.72 | 10.42±6.86 | 11.387±0.001 | 11.387±0.002 |
> | PPG | Nonlin | 1150.32±26.78 | 1609.05±24.74 | 11.728±0.013 | 11.731±0.013 |
> | EigenWorms | Lin | 3.16±5.38 | 3.21±6.24 | 0.716±0.001 | 0.716±0.001 |
> | EigenWorms | Nonlin | 65.06±2.33 | 117.66±2.43 | 0.759±0.004 | 0.760±0.003 |
> | SCP2 | Lin | 15.59±15.53 | 13.02±8.87 | 0.104±0.000 | 0.103±0.000 |
> | SCP2 | Nonlin | 27.96±42.50 | 41.28±29.39 | 0.107±0.000 | 0.108±0.001 |
>
> **Key observations:**
>
> * **Memory usage** is comparable between RHEL and BPTT as expected.
> * **Runtime** is generally comparable as expected too, with RHEL showing significant speedup for nonlinear models (e.g., 2× faster for EigenWorms with UniCORNN). We hypothesize that this speed-up is implementation-dependent and would require further analysis to be conclusive. Yet again, as stated above, the goal of our work is not to outperform BPTT in terms of these metrics.
>
> **Proposition 1**. We propose to include both the table runtimes and memory consumptions, as well as the complexity analysis of RHEL and BPTT.
>
> > [...] and this reduces the selling point of the paper being "forward-only".
>
> **Answer 2**.
> We also want to clarify the position of our algorithm being "forward-only" with respect to *zeroth-order optimization* (ZO)[2] which the reviewer may have in mind. In the literature of digital hardware-friendly learning, ZO techniques are pursued for their **memory efficiency** compared to vanilla backprop with gradient checkpointing [3], which can be further enhanced with appropriate quantization schemes [4]. The relatively recent exploration of ZO techniques on Large Language Models (LLMs) finetuning [5] has spurred a revival of "forward-only" techniques for memory-cheap gradient computation in LLMs on a variety of applications.
>
> However, the motivation for the "forward-only" feature of RHEL is *not* memory efficiency. Though we make use of the same terminology, "forward-only" gradient computation algorithms are very differently motivated in the *physical computing* literature [6] which RHEL belongs to -- see next section of our rebuttal. In our context, "forward-only" means that both the forward *and* backward passes obey the *same* physical principles, for instance obeying to Hamiltonian dynamics as required by RHEL. This means that in principle, any piece of hardware that can sustain inference can also sustain gradient computation, out of the same physics, and therefore learn "by itself".
>
> **Proposition 2**. We propose to add an extra related work paragraph dedicated to forward-only algorithms and explain how their motivation contrast with ours.
>
> > The writing is not very clear at times and sometimes seem a bit overdramatic. For example, "the physical realization of our algorithm" in the abstract is not very clear. and "this paper embraces an extreme view" is quite dramatic.
>
> **Answer 3.**
> We want to acknowledge the lack of clarity of our wording revolving around the motivation for physical computing.
>
> Every computation, be it on a digital accelerator or any alternative substrate, is sustained by a physical process carried by wires and transistors which creates entropy [7]. Therefore the very notion of "physical computing" may sound pleonastic. However, digital hardware designs abstract core physics away to enable idealized computations relying upon statelessness, unidirectionality, determinism and synchronization [6]. Compilation pipelines are then typically designed to be as hardware-agnostic and as general-purpose as possible. In this paradigm, "hardware-software" codesign often amounts for instance to write bespoke low-level kernels for specific workloads to mitigate their compute, memory costs and off-chip memory accesses [8]. Still, these approaches remain largely physics-agnostic.
>
> "Physics-based" computing significantly deviate from this framework in two ways. First, the designs may target Application-Specific Integrated Circuits (ASICs) rather than general-purpose machines [6]. Second, the aforementioned digital design constraints may be relaxed and the resulting *analog* physics leveraged. For instance, supply voltages may be decreased and resulting circuits become non-deterministic, so that stochastic algorithms can be inherently mapped to such systems [9]. Uni-directionality may be relaxed such that vector-matrix multiplication can be natively implemented by Kichoff laws of currents [10]. Optimization problems may be embodied into a physical system and subsequently solved as the system settles to equilibrium [11]. As a particular case, physical "self-learning machines" are physical embodiments of neural networks whose inference and gradient computation are, *for instance*, solved by relaxing to equilibrium [12, 13]. This is how "physical realization of our algorithm" should be understood as.
>
> **Proposition 3**. We will remove the sentences which may sound "overdramatic" (L. 9 and L.29 ), replace them with clearer and softened statements based on the comments above and add a related work paragraph around physical computing.
>
> > Why is it surprising/need to be posited that "some SSMs could inherently be mapped onto dynamical systems"?
>
> **Answer 4.**
> We acknowledge this is not surprising, as SSMs are themselves derived as discretization of continuous-time dynamics [14]. However, the primary motivation of SSMs is not the design of "physics-based" models but a means to overcome the quadratic cost in sequence length of transformer-based training on digital hardware [8].
>
> > Many systems in reality are not conservative, which may reduce the impact.
>
> **Answer 5**.
> We acknowledge this limitation. See our answers to Reviewers  WcdN (Answer 1, Proposition 1) and k16k (Answer 1, Proposition 1) for greater details.
>
> > The informal versions of the theorems are stated in the paper and the formal versions are not signposted.
>
> **Proposition 4.** Acknowledged, this will be fixed in case of acceptance.
> > Theorem 3.1, if you take  to 0, wouldn't both methods give the echo trajectory simply be the forward trajectory in reverse for both cases?
>
> **Answer 6**.
> We assume that the reviewer implies that both $\Phi^e(t, \epsilon)$ and $\Phi^e(t, -\epsilon)$ converge to the time-reversed trajectory $\Phi^\star(-t)$ as $\epsilon \to 0$ and therefore the $\Phi^e(t, \epsilon) - \Phi^e(t, -\epsilon)$ vanishes. Fortunately, the error signal is carried by $\epsilon^{-1}(\Phi^e(t, \epsilon) - \Phi^e(t, -\epsilon)) \to \partial_\epsilon(\Phi^e(t, \epsilon))|_{\epsilon=0}$ and converges to a *finite* value as $\epsilon\to 0$. Our proof of Theorem 3.1 heavily relies on this fact.
>
> [1] Guy E Blelloch. “Prefix Sums and Their Applications”. In: (1990).
>
> [2] Spall. "Multivariate stochastic approximation using a simultaneous perturbation gradient
> approximation". 1992.
>
> [3] Chen et al. "Training deep nets with sublinear
> memory cost". 2016.
>
> [4] Zhou et al. "QuZO: Quantized zeroth-order fine-tuning for large language models." 2025.
>
> [5] Malladi et al. "Fine-tuning language models with just forward passes." 2023.
>
> [6] Aifer, Maxwell, et al. "Solving the compute crisis with physics-based ASICs." 2025.
>
> [7] Wolpert, David H. "The stochastic thermodynamics of computation." 2019.
>
> [8] Gu & Dao. "Mamba: Linear-time sequence modeling with selective state spaces." 2023.
>
> [9] Coles et al, "Thermodynamic AI and the fluctuation frontier". 2023.
>
> [10] Burr et al. "Neuromorphic computing using non-volatile memory". 2017.
>
> [11] Vadlamani et al , "Physics successfully implements lagrange multiplier optimization". 2020.
>
> [12] Stern et al. "Supervised learning in physical networks: From machine learning to learning machines." 2021.
>
> [13] Kendall et al. "Training end-to-end analog neural networks with equilibrium propagation." 2020.
>
> [14] Gu et al. "Efficiently modeling long sequences with structured state spaces." 2021.

---

> > ### Comment · Reviewer_AkMK · 2025-08-05
> >
> > Thank you to the authors for the detailed response, and interesting analysis in Q1 response of WcdN and k16k. I think Propositions 1 to 4 will improve the quality of the paper and I am happy to raise my score.
> >
> > My final comment though is that SSMs as used here in the paper refers only to deep SSMs. But SSMs in general existed in dynamical systems and control theory much earlier than the introduction of the structured SSMs for sub-quadratic costs as an alternative to transformers (to at least Kalman and possibly earlier), we shouldn't forget about classical theory! Hence why I remarked that "some SSMs could inherently be mapped onto dynamical systems" should not be surprising.

---

> > > ### Author Response · Authors · 2025-08-06
> > >
> > > We thank the reviewer for their response and we are happy they acknowledged our proposed improvements by raising their score.
> > >
> > > We acknowledge that State Space Models have deep roots in dynamical systems and control theory, dating back to classical works like Kalman filtering, which makes it natural to think of them as physical systems being controlled. Building upon these works, our contribution is demonstrating that by assuming a reversible Hamiltonian structure, these physical systems can also be used for learning their own parameters to minimize a cost function, creating self-learning machines.

---

### Comment · Area_Chair_ZvqB · 2025-08-05
**Rebuttal Phase Closing Soon**

Dear Reviewers,

As the author rebuttal phase will close in less than 48 hours, we kindly ask you to make use of the remaining time to engage with the authors if there are any outstanding questions or concerns. At a minimum, please acknowledge that you have read the rebuttal. If you have already done so, we sincerely appreciate your efforts.

Thank you in advance for your continued contributions.

Best,

AC

---

### Comment · Area_Chair_ZvqB · 2025-08-08
**Question about baseline results in Table 1**

I had a quick question regarding the results presented in Table 1. It seems that some strong baselines are missing. While I do not necessarily expect your method to outperform all existing SSMs, including such baselines would help put your performance into perspective. For example, [1] reports about 95 for Worms, whereas your baseline is 78.3.

Could you clarify whether this gap reflects a limitation of your current method and setup, or whether you expect performance to improve with refinements (e.g., more compute, larger models, better tuning)?

[1] https://openreview.net/pdf?id=GRMfXcAAFh

Best regards,

AC

---

> ### Author Response · Authors · 2025-08-09
>
> Thank you for this important clarification question regarding the performance of baselines observed in Table 1, particularly for the Worms dataset.
>
> **The performance gap is primarily due to integrator choice—RHEL requires a non-dissipative integrator.** The baseline you reference achieves 95% accuracy using the IM integrator, which is dissipative, while our approach requires a non-dissipative integrator to maintain the time-reversal symmetry essential for RHEL.
>
> **We used an adaptation of their non-dissipative IMEX integrator and achieved similar results to the original implementation (80.0% vs. our 78.3%).** This gap appears consistently for both RHEL and BPTT under symplectic integration, confirming the difference stems from the integrator choice, not our learning algorithm.
>
> **Dissipative and non-dissipative integration methods have different strengths depending on the task.** As highlighted in [1], performance depends on data distribution characteristics. On long-sequence tasks like Worms (17,984 timesteps), dissipative systems may have an advantage because dissipation enables forgetting of past information, while non-dissipative Hamiltonian systems must preserve the entire trajectory history and may require larger models for equivalent performance.
>
> **We propose to address this in our revision by:**
> - Adding an appendix section explaining the relationship between our integrator and those used in [1]. We use an adaptation of their IMEX integrator, but cannot use their dissipative IM integrator due to RHEL's requirement for reversible dynamics.
> - Discussing the performance trade-offs highlighted in [1] between dissipative and non-dissipative integration, depending on data characteristics.
>
> **We expect this gap could be reduced with larger models.** Our focus was on demonstrating that RHEL matches BPTT performance on models where RHEL can be applied (L235-L237). This performance gap underscores the importance of our discussions with reviewers (especially reviewer WcdN) about adapting RHEL to dissipative systems—a development that could potentially capture the performance benefits of dissipation while maintaining RHEL's forward-only learning advantages.
>
> [1] https://openreview.net/pdf?id=GRMfXcAAFh

---

### Decision · Program_Chairs · 2025-09-17

**Decision:**

Accept (oral)

**Comment:**

This paper introduces a new approach for training recurrent models without relying on Backpropagation Through Time (BPTT). Specifically, it proposes Recurrent Hamiltonian Echo Learning (RHEL), an algorithm that computes loss gradients as finite differences of trajectories of non-dissipative Hamiltonian systems.

A key downside of the proposed approach is that RHEL requires the underlying system to be non-dissipative. This is a somewhat restrictive assumption that can limit applicability and hurt performance. As a result, models trained with BPTT currently tend to outperform RHEL in terms of accuracy. Nevertheless, the fact that only three forward passes are required for learning is an exciting property. As mentioned by the authors, its full potential will likely depend on the development of improved analog hardware, and it remains to be seen whether such hardware will become available. Despite these limitations, the work provides a fresh perspective and opens an exciting new research direction motivated by dynamical systems theory.

The reviews are overall very positive and confirm that the underlying mathematical framework is sound. Open questions and concerns have been effectively addressed during the rebuttal and discussion phases. I agree with some of the comments that the work remains somewhat limited in its current form, but this does not diminish its potential impact. I would encourage the authors to more explicitly acknowledge current limitations and to expand the discussion of promising directions for future work. Additionally, I suggest moderating the use of strong adverbs (e.g., “radically,” “tremendous”) to maintain a more neutral academic tone.

In light of these considerations, I strongly recommend this paper for acceptance.